# A smart multiantenna gene theranostic system based on the programmed assembly of hypoxia-related siRNAs

Xue Gong[1], Haizhou Wang[2], Ruomeng Li[1], Kaiyue Tan[1], Jie Wei[1], Jing Wang[1], Chen Hong[1], Jinhua Shang[1], Xiaoqing Liu [1✉], Jing Liu[2] & Fuan Wang [1✉]

The systemic therapeutic utilisation of RNA interference (RNAi) is limited by the non-specific off-target effects, which can have severe adverse impacts in clinical applications. The accurate use of RNAi requires tumour-specific on-demand conditional activation to eliminate the off-target effects of RNAi, for which conventional RNAi systems cannot be used. Herein, a tumourous biomarker-activated RNAi platform is achieved through the careful design of RNAi prodrugs in extracellular vesicles (EVs) with cancer-specific recognition/activation features. These RNAi prodrugs are assembled by splitting and reconstituting the principal siRNAs into a hybridisation chain reaction (HCR) amplification machine. EVs facilitate the specific and efficient internalisation of RNAi prodrugs into target tumour cells, where endogenous microRNAs (miRNAs) promote immediate and autonomous HCR-amplified RNAi activation to simultaneously silence multiantenna hypoxia-related genes. With multiple guaranteed cancer recognition and synergistic therapy features, the miRNA-initiated HCR-promoted RNAi cascade holds great promise for personalised theranostics that enable reliable diagnosis and programmable on-demand therapy.

---

[1] Key Laboratory of Analytical Chemistry for Biology and Medicine (Ministry of Education), College of Chemistry and Molecular Sciences, Wuhan University, Wuhan, P. R. China. [2] Department of Gastroenterology, Zhongnan Hospital, Wuhan University, Wuhan, P. R. China. ✉email: xiaoqingliu@whu.edu.cn; fuanwang@whu.edu.cn

RNA interference (RNAi) has been recognised as a new versatile gene-silencing tool for posttranscription-based gene therapy, in which small interfering RNAs (siRNAs) are utilised as effector molecules to guide potent mRNA cleavage[1–3]. In addition to the efficient siRNA delivery, non-specific off-target effects are significant hurdles because they can have severe adverse impacts in clinical applications[4,5]. Therefore, the design of a conditionally activated RNAi system is highly appealing for maximising therapeutic efficacy while minimising side effects[6–10]. In particular, tumour-specific stimulation by RNAi might hold great promise for the development of new intelligent theranostic systems. A smart RNAi theranostic system requires efficient RNAi delivery/release and programmed activation/expression by appropriate endogenous biomarker stimuli[11–14]. Among these different biomarkers, microRNAs (miRNAs) have attracted a large amount of attention because variations in their expression are closely associated with different disease types and stages[15,16]. Moreover, efficient RNAi relies only on double-stranded siRNAs (duplex siRNAs) that consist of a sense strand and a complement antisense strand to suppress the expression of the target gene. Thus, efficient integration of the miRNA stimulus and duplex siRNA assembly is needed to develop the on-demand RNAi-based smart therapies. Considering the low abundance of miRNAs in earlier stage of disease, efficient RNAi activation requires in situ sustained duplex siRNA assembly through DNA amplification devices or circuits. The simultaneous stimulation of multiple RNAi pathways could also contribute to the promoted therapeutic performance[17–19]. Fortunately, the hybridisation chain reaction (HCR) is considered an ideal and promising machine for enhancing the miRNA recognition with multiple RNAi executions[20–22]. Analyte-triggered HCR mediates the promoted assembly of long dsDNA copolymers from the autonomous and successive cross-opening of hairpin reactants[23–25], which could also be utilised as versatile RNAi nanocarriers and real-time status trackers. The encoding of these functional sense and antisense RNAs into their respective HCR hairpins leads to inert RNAi for the active duplex siRNA, which is prevented by these caged hairpin partitions. Hence, HCR could be used as a diagnosis-guided smart therapy through the specific miRNA-amplified assembly of multiple duplex siRNAs with different RNAi functions.

DNA nanocarriers can protect siRNAs from undesired digestion, yet the utility of siRNAs is still limited by inefficient systemic delivery and poor stability under harsh in vivo conditions in clinical trials. Although various nanomaterials have been introduced for delivering siRNAs, including polymers, liposomes and metal-organic framework nanoparticles[26–29], they are serious concerns regarding the cytotoxicity and immunogenicity of these materials. Extracellular vesicles (EVs), nanoscale plasma membrane-encased vehicles, have been recently recognised as attractive nanocarriers for precise delivery of therapeutic/diagnostic cargos[30–33]. Their naturally inherited biochemical compositions endow EVs with fascinating parent cell-targeting features, high biocompatibility and favourable immunogenicity. With an ideal "invisibility cloak", these allogenic EV shuttles exhibit tremendously enhanced stability by diminishing the clearance of the mononuclear phagocyte system during systemic circulation[34–37]. Thus, the tumour cell-derived EV-delivered RNAi nanoplatform is expected to sustain personalised theranostic schemes by integrating diagnosis-guided RNAi activation with real-time tracking of pathological tissues.

In this work, we construct an endogenous miRNA-guided HCR-promoted RNAi theranostic system that is easily delivered by specific tumour-targeting Trojan EVs. Compact RNAi is reconstituted by splitting the active duplex siRNA into inactive sense and antisense RNAs that are hybridised with different HCR reactants, resulting in RNA/DNA hybrid assembly without RNAi activity. These integrated RNAi prodrugs are loaded into EV nanovectors to facilitate their tumour-specific accumulation and efficient endocytosis by cancer cells. Then, endogenous miR-21-activated HCR mediates the autonomous cross-opening of these functional RNAi prodrugs to simultaneously produce two different duplex siRNAs with restored RNAi functions. The present miRNA-stimulated bis-siRNA theranostic nanoplatform achieves the multiantenna-guided cascade gene silencing, and its therapeutic performance is synergistically enhanced. Thanks to the protecting EV nanocarriers and HCR amplifiers, compared to regular siRNAs, the sophisticated RNAi prodrugs exhibits substantially enhanced stability and tumour-targeting properties, thus guaranteeing accurate and efficient gene silencing. Under sequential and multiple guaranteed RNAi administration and immediate activation via miRNA guidance, intelligent HCR-stimulated multiple siRNAs achieve in vivo programmable anticancer theranostics in both subcutaneous and metastatic tumours.

## Results

**Principle of miRNA-responsive RNAi.** The principle of our stimulus-responsive RNAi cascade is schematically illustrated in Fig. 1. The active duplex siRNAs were split into inactive sense RNA(s) and antisense RNA(s) that were hybridised with the partition DNAs, leading to the formation of two RNA/DNA hybrids without RNAi capacity (part I, Fig. 1). The higher thermodynamic stability of RNA/DNA hybrids than the RNA duplex results in a prolonged circulation duration in the bloodstream and thus enhanced RNAi efficiency[38,39]. In the presence of trigger (T), the exposed sequences x* and y* allowed the proximity-induced mutual recognition/hybridisation of RNA/DNA hybrids that led to energetically favoured toehold-mediated branch migration and strand displacement, resulting in the activation of the RNAi system. The as-assembled stimulus-responsive RNAi strategy could convert target recognition into siRNA generation, thus offering the facile control of siRNA activity on demand without external interference. To further sensitise RNAi stimulation of limited molecular recognition, the HCR amplifier was introduced to successively activate the RNAi procedure by an endogenous biomarker (part II, Fig. 1). The present miR-21-activated HCR-promoted bis-RNAi nanosystem was based on two pairs of split siRNAs (RNAi-1 and RNAi-2) that comprised inactive sense and antisense RNAs. These RNAs were hybridised with hairpins $H_1$, $H_2$, $H_3$ and $H_4$, resulting in RNA/DNA hybrid assembly without their corresponding RNAi activity. In our design, $H_1$ was elongated with domain $s_1*$, which was complementary to the sense RNAi-1($s_1$), while $H_3$ was grafted with domain $as_1*$, which was complementary to antisense RNAi-1 ($as_1$). Meanwhile, sense RNAi-2 ($s_2$) and antisense RNAi-2 ($as_2$) were hybridised with the 5′-elongated domain $s_2*$ of $H_2$ and the 3′-elongated domain $as_2*$ of $H_4$, respectively (for details, see Supplementary Fig. 1). Each of the RNA/DNA hybrids was designed and optimised with NUPACK to avoid undesired RNAi leakage. The driving force for miR-21-initiated HCR-guided RNAi operation could be revealed by the substantial free energy difference between the initial and final states ($\Delta\Delta G = -41.89$ kcal mol$^{-1}$; for details, see supporting information), suggesting the feasibility of our proposed strategy. To trace the miR-21-amplified assembly of multiple siRNAs, all of these sense RNAs were labelled at their 5′-end with a fluorescence donor (Cy3), while the antisense RNAs were labelled at their 3′-end with a fluorescence acceptor (Cy5). The target miR-21 opened $H_1$ by hybridising with domain a*-b*. The as-exposed domain b-c of $H_1$ immediately hybridised with domain c*-b* of $H_2$. The as-

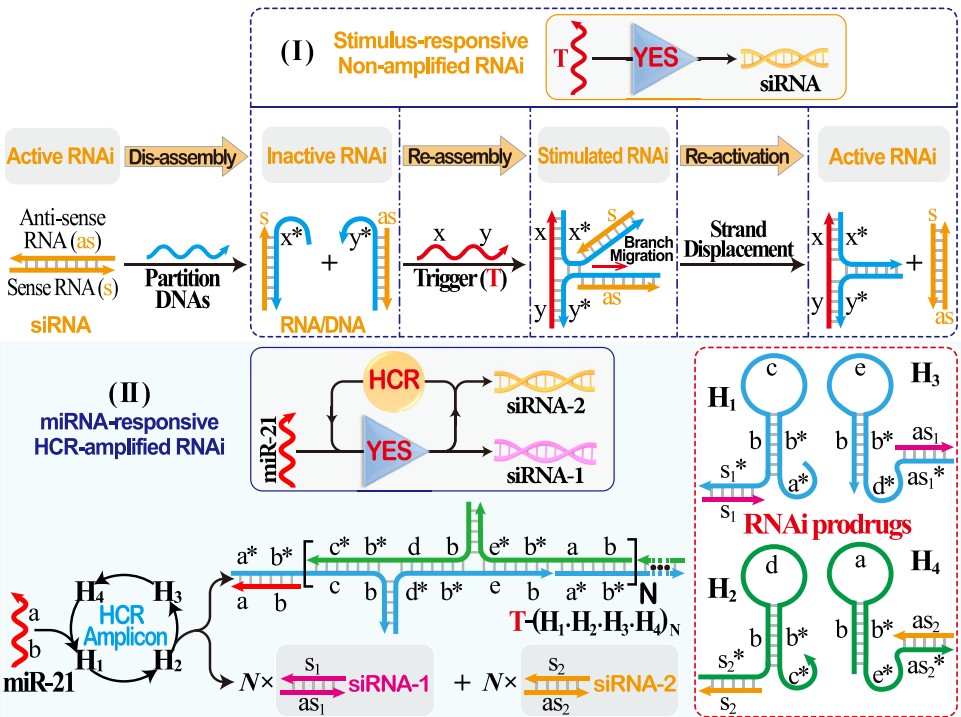

**Fig. 1 The stimulus-responsive RNAi theranostic system.** Reconstitution of the RNAi system from active siRNA (consisting of sense RNA and antisense RNA) into inactive RNA/DNA hybrids for assembling stimulated RNAi through siRNA reassociation. The RNA/DNA hybrids pave the way for stimulus-responsive non-amplified RNAi (part I) and miRNA-stimulated HCR-amplified RNAi (part II) systems.

unfolded $H_2$ then exposed the sequence d-b, which was complementary to the sequence d*-b* of $H_3$. As a result, the opened $H_3$ hybridises with the sequence e*-b* of $H_4$ to release the target sequence a-b that could unfold $H_1$ again, leading to the continuous and successive cross-hybridisation cycle. Simultaneously, the unlocked $H_3$ included grafted sequence $as_1$* that could bring it close to the complementary sequence $s_1$* of $H_1$, resulting in the proximity-induced toehold-mediated branch migration and strand displacement. This led to the cyclic reassembly of $s_1/as_1$ to efficiently activate the corresponding RNAi-1 system by an HCR amplifier. At the same time, unfolded $H_2$ and $H_4$ brought the strands $s_2$* and $as_2$* in close proximity to activate another RNAi-2 pathway via HCR amplification. Upon the reassembly and activation of bis-siRNAs, the fluorophore donor (Cy3) of sense RNAs and the fluorescence acceptor (Cy5) of antisense RNAs were brought into close proximity, resulting in an efficient Förster resonance energy transfer (FRET) readout. This functional HCR system enabled a highly sensitive and selective assay of miR-21, even in diluted 10% serum solution (Supplementary Figs. 2 and 3). The reassociation of the bis-siRNA system was evident, as revealed by native polyacrylamide gel electrophoresis (PAGE) experiments (Supplementary Fig. 4). Then, the morphological features of HCR-assembled dsDNA products were characterised by atomic force microscopy (AFM, Supplementary Fig. 5). Thus, the HCR-amplified activation of bis-RNAi occurs only in the presence of miR-21, indicating that the RNA/DNA hybrids could be utilised as programmable RNAi prodrugs for simultaneously stimulating cancer-specific diagnosis and on-demand therapy.

**Characterisation of the RNAi prodrug-packaged EVs.** To efficiently deliver the designed RNAi prodrugs, tumour cell-derived EVs were harvested from specific miR-21 inhibitor-pre-treated MDA-MB-231 cells by using ultracentrifugation (for details of the synthesis and characterisation of the small molecule inhibitor, see Supplementary Figs. 6–8; for details of the preparation of EVs, see

Supplementary Fig. 9)[32,40]. Then, the undesired miR-21-induced HCR reaction could be fully expelled in EVs before their internalisation into target MDA-MB-231 cells. Dynamic light scattering (DLS) analysis revealed that EVs exhibited a characteristic size-distribution profile (132 nm, zeta potential of −17.4 mV, Supplementary Fig. 10a), and transmission electron microscopy (TEM) analysis showed that EVs showed a typical cup-like morphology (Supplementary Fig. 10b). Furthermore, western blot analysis showed an enrichment of common EV biomarkers, such as TSG101, flotillin 2 and tetraspanins CD63, along with a decrease in the expression of organelle marker calnexin in EVs (Supplementary Fig. 10c). As expected, endogenous miR-21 was almost completely eliminated from EVs (Supplementary Fig. 10d), and miR-21-lacking EVs were used as versatile nanocarriers for all subsequent experiments. Our EVs were sufficiently stable and did not exhibit obvious size changes even after three freeze-thaw treatments (Supplementary Fig. 11). Then, the EVs were transfected with RNAi prodrugs via electroporation (Fig. 2a), after which they displayed a slightly increased size (~146 nm) without noticeable morphological changes (Fig. 2b, c) or obvious aggregation of RNAi prodrugs (Supplementary Fig. 12). The corresponding loading efficiency was calculated to be 0.14 μg of RNAi prodrug per μg of protein (Fig. 2d, e), which was comparable or even higher than that of some of the existing siRNA delivery carriers (Supplementary Table 3). Moreover, the EVs could prevent the nuclease degradation of the encapsulated RNAi prodrugs (Supplementary Fig. 13) and undesired HCR leakage (Supplementary Fig. 14). In addition, the high biocompatibility of our EVs was demonstrated by a cytotoxicity assay, haemolysis test and systemic immune response analysis (Supplementary Fig. 15), as well as long-term haematology studies (Supplementary Fig. 16).

Subsequently, we examined the homologous uptake of EVs by recipient MDA-MB-231 cells and found that clathrin/caveolae-mediated endocytosis played a dominate role (Supplementary Fig. 17).

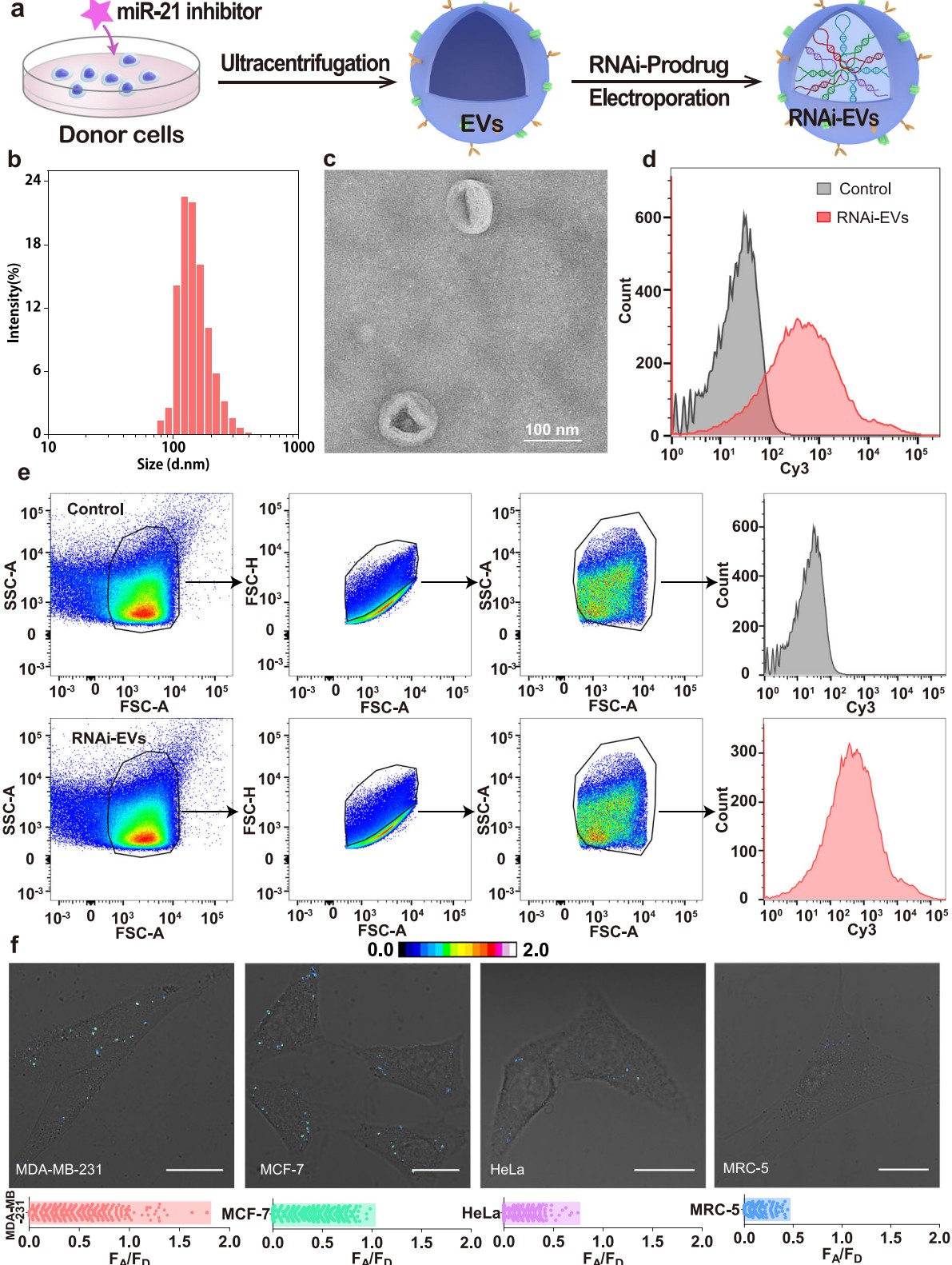

**Fig. 2 Characterisation of the RNAi prodrug-packaged EVs. a** Schematic illustration of RNAi prodrug-encapsulated EVs (RNAi-EVs). **b** Hydrodynamic diameter measurement, **c** TEM image and **d**, **e** flow cytometric analysis of the RNAi prodrug-loaded EVs. The characteristic results (DLS, TEM, flow cytometric analysis) are representative of three independent experiments. **f** Diagnostic EV-sustained HCR-amplified FRET analysis of intracellular miR-21 (in the form of $F_A/F_D$) and corresponding FRET signal distributions in different cell lines. The fluorescence images are representative of ten independent experiments. All scale bars are 20 μm. Source data are provided as a Source Data file.

Then, the EV-packaged RNAi prodrugs escaped from lysosomes (Supplementary Fig. 18), which might have been attributed to the back fusion of EVs with the endosomal membrane[41]. Thus, the RNAi prodrugs could be efficiently delivered into the cytosolic compartment of live cells. In addition, preferential tropism of EVs towards their progenitor cells was explored in MDA-MB-231, MCF-7, HeLa and MRC-5 cells (Supplementary Fig. 19). Clearly, homologous MDA-MB-231 cells exhibited the higher uptake of EVs than nonhomologous cells, which was in good accordance with previous reports[36,37]. In addition, the bioactivity of the EV membrane was retained after electroporation, as indicated by the high cellular internalisation of the EVs (Supplementary Fig. 20). Moreover, the MAD-MB-231 cells showed a significantly higher cellular uptake of homologous EVs than heterologous EVs and even the commercial transfection reagents (Supplementary Fig. 21). Taken together, the results suggest that biocompatible EVs enabled specific targeting to homotypic cells with significantly enhanced payload transduction.

To trace the tumour-specific stimulation of RNAi in the intracellular environment, we assessed miR-21 expression in different living cells by confocal laser scanning microscopy (CLSM). To avoid undesired interference from the surrounding complex intracellular environment, the fluorescence emission ratio of the acceptor to donor ($F_A/F_D$) of the HCR amplifier was adapted as the readout signal to ensure a more effective and accurate diagnosis and to immediately guide therapeutic operation at the single cell level. HCR-motivated RNAi dominated the overall cellular biotransformations for the initial 4 h (Supplementary Fig. 22), after which the RNase H-mediated degradation of RNAi prodrugs could eliminate the residual RNAi prodrugs without side effects. As displayed in Fig. 2f, the most intense FRET imaging signal was observed in MDA-MB-231 and MCF-7 cells, followed by HeLa cells, while a minimal FRET imaging signal was observed in noncancerous MRC-5 cells (for details, see Supplementary Fig. 23). These results are consistent with literature reports[42–44], in which miR-21 was shown to be highly expressed in tumour cells, but not in normal cells. Obviously, the integrated HCR nanosystem could discriminate the varied expression of miR-21 in different cells, which was in good accordance with the quantitative reverse transcription-PCR (qRT-PCR) data (Supplementary Fig. 24). We found that the FRET signal of the bis-siRNA-loaded system was ~1.5 times (0.51, sample a in Fig. 2f) higher than that of one siRNA-loaded system alone (0.33, sample a in Supplementary Fig. 25), which was attributed to the enhanced assembly of siRNA from the bis-RNAi HCR nanoplatform by the miR-21 analyte. Furthermore, the FRET signal was ~0.23 upon incubation with the $H_4$-excluded system (sample b in Supplementary Fig. 25), which was ascribed to the terminated HCR transduction that leads to a tiny fraction of siRNA activation. Almost no FRET signal was obtained when the miR-21 expression was knocked down by introducing an anti-miR-21 inhibitor oligonucleotide into MDA-MB-231 cells (sample c in Supplementary Fig. 25), indicating that endogenous miR-21 activates the efficient assembly of RNAi prodrugs on demand without interference. In addition, the FRET efficiency was estimated to be 0.67 based on a conventional acceptor-photobleaching technique (Supplementary Fig. 23c). The miRNA-catalysed HCR-mediated cascade drug release/activation system may offer new possibilities for early cancer diagnosis and immediate therapeutic intervention.

**In vitro cytotoxicity assay.** The mechanism of miR-21-activated HCR-involved multiple RNAi operation was further explored by assessing cytotoxicity under hypoxic conditions. To investigate the generality of the split siRNA strategy, two pairs of RNA/DNA hybrids were designed as model therapeutic agents against twist

and hypoxia-inducible factor-1α (HIF-1α), and were integrated into the same HCR amplifier. HIF-1α is induced by intratumoural hypoxia, which plays a critical role in tumour progression and metastasis, leading to treatment failure and even mortality. Recent studies have suggested that twist-related proteins are downstream targets of HIF-1α and have important roles in metastatic phenotypes induced by hypoxia or HIF-1α overexpression[19]. Simultaneous inhibition of the key hypoxia-involved signalling pathway may lead to enhanced anticancerous efficacy and inhibited tumour progression. Here, the therapeutic performance of RNAi prodrug-packaged EVs was investigated in MDA-MB-231, MRC-5, HeLa and MCF-7 cells. Among these different cells, MDA-MB-231 cells were the most sensitive to functional EVs (Fig. 3a), which was attributed to the homotypic targeting capacity of EVs and the subsequent bis-siRNA activation via the HCR amplifier (Supplementary Fig. 26). Then, the HCR hairpin was modified with the photosensitizer indocyanine green (ICG) to achieve auxiliary photothermal therapy (PTT) (for detailed characterisation, see Supplementary Fig. 27). The EV-encapsulated ICG-RNAi prodrugs showed slightly better photothermal performance than ICG-RNAi prodrugs only (Supplementary Fig. 28), which was attributed to the high condensation of ICG with lower heat dissipation in EVs[45]. Note that photoirradiation had no effect on the therapeutic performance of RNAi prodrugs (Supplementary Fig. 29), indicating that ICG- labelled RNAi prodrugs could be utilised for combined gene therapy and phototherapy. The synergistic therapeutic effect of our multiantenna gene regulation and phototherapy was further demonstrated by using the MTT assay (Supplementary Fig. 30).

The combination index (CI) was used to evaluate the inherent relation between hypoxia-related gene silencing and auxiliary PTT. As displayed in Fig. 3b, the miRNA-stimulated twist and HIF-1α siRNAs showed an obvious synergistic effect (0 < CI < 0.4) on MDA-MB-231 cells, suggesting that the EV-delivered bis-siRNAs could indeed enhance gene silencing. In addition, an obvious synergism (0 < CI < 0.44) was observed for the photo-irradiated miRNA-activated bis-RNAi system, indicating a cooperative therapeutic effect between gene therapy and PTT. Moreover, the motility/invasiveness of MDA-MB-231 cells subjected to different treatments investigated by using the transwell migration and Matrigel invasion assays (Fig. 3c, d and Supplementary Fig. 31). The synchronised activation of twist and HIF-1α siRNAs led to decreased motility/invasiveness (69% and 82%, respectively), which were much higher than those of the twist or HIF-1α siRNA alone. In contrast, HCR-assembled scrambled siRNAs (negative control) had little effect on the motility/invasiveness of cells, which was because for these deficient siRNAs have no RNAi activity. In addition, the miR-21-activated bis-RNAi system led to ~1.3-fold reduction in motility/invasiveness compared with the straightforward bis-RNAi approach, indicating the enhanced therapeutic effect of our miRNA-activated RNAi approach. Under photoirradiation, the efficiency of the inhibitory effect of EV-delivered bis-RNAi on motility/invasiveness was ~1.1-fold higher than that of the liposome system, demonstrating the homotypic cell targeting of EVs. Photoirradiated bis-RNAi abrogated the motility/invasiveness of cells, indicating the enhanced therapeutic performance of the combined bis-gene silencing and PTT, which was consistent with the live/dead cell assay (Fig. 3e).

**In vivo systemic biodistribution.** The satisfactory in vitro diagnosis-guided therapeutic performance of the HCR-stimulated RNAi prodrug system encouraged us to assess the simultaneous miRNA-stimulated bis-RNAi platform in vivo (Fig. 4a). To guarantee a more developed hypoxic tumour microenvironment,

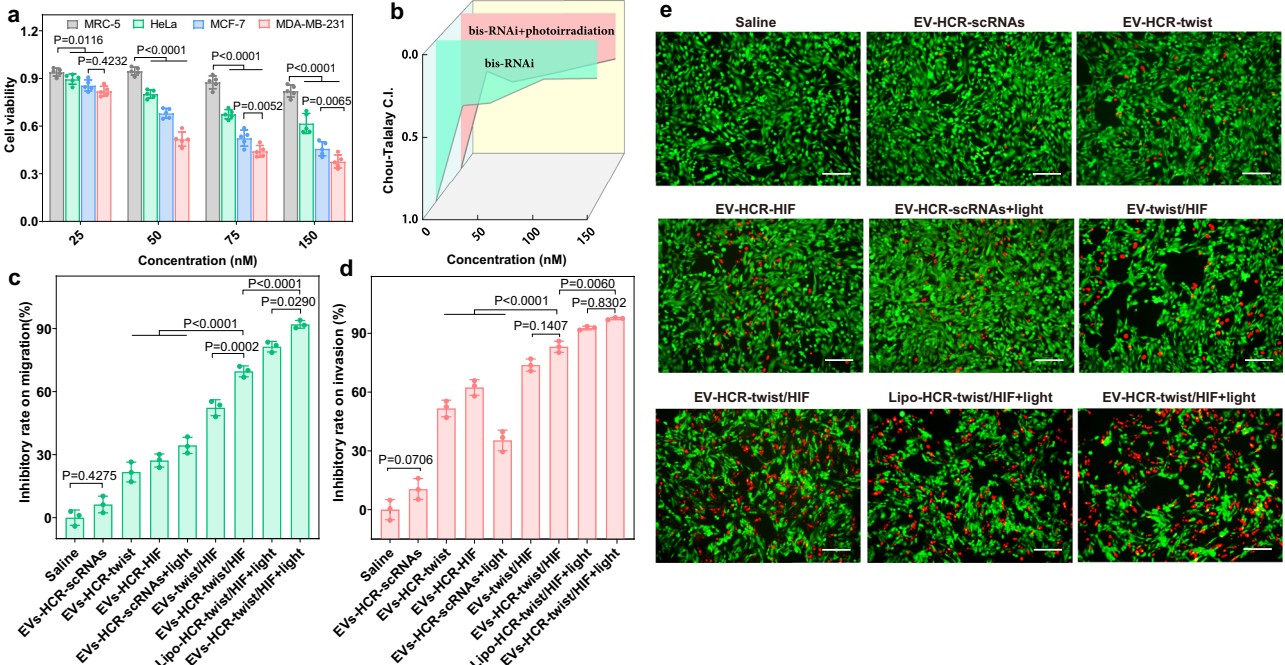

**Fig. 3 Cell cytotoxicity evaluation of RNAi prodrug-packaged EVs under hypoxic conditions. a** Cell viability of MRC-5, HeLa, MCF-7 and MDA-MB-231 cells after incubation with functional EVs containing with different concentrations of RNAi prodrugs (25, 50, 75 and 150 nM) for 48 h. Data represent the mean ± s.d. from five independent replicates. Statistical significance is calculated by two-way ANOVA followed by Bonferroni's multiple comparisons test. **b** Combination index (CI) values of the RNAi prodrug-packed EVs with and without photoirradiation. Results of **c** the transwell migration assay, **d** the Matrigel invasion assay and **e** live/dead cell analysis in MDA-MB-231 cells subjected to different treatments. All photoirradiation was carried out with an 808 nm laser (0.7 W/cm²) for 5 min. Data represent the mean ± s.d. from either five (**a**) or three (**c**, **d**) independent replicates. The images are representative of three independent experiments. Statistical significance is calculated by one-way ANOVA followed by Tukey's post hoc test for **c** and **d**. All scale bars are 200 μm. Source data are provided as a Source Data file.

mice bearing ~300 mm³ tumours were given tail vein injections of free ICG-RNAi prodrugs, ICG-RNAi prodrug-loaded liposomes and ICG-RNAi prodrug-packaged EVs. Then, these intravenously injected mice were examined by using an in vivo optical imaging system at various time points thereafter. As shown in Fig. 4b, the intravenously injected free RNAi prodrugs were eliminated without obviously accumulating in tumour site. Unlike the liposomes, the EVs progressively accumulated in tumour tissue and reached maximal accumulation after 12 h. In addition, the endothelial cells of tumour vessels were stained for the biomarker CD31 to evaluate ability of EVs to penetrate deep tumour tissue. Compared with the free RNAi prodrugs and liposome-delivered RNAi prodrugs, the EV-delivered RNAi prodrugs showed much less colocalization with tumour vessels (Fig. 4c), implying that the EV carriers can effectively extravasate from tumour vessels to penetrate into tumour tissue.

We also evaluated their time-dependent distribution in the main organs. After normalising the radiation efficiency by organ weight, the EVs displayed a higher ratio of tumour to liver or kidney (Supplementary Fig. 32). These ex vivo results were consistent with the whole-body imaging observations, thus demonstrating the favourable tumour accumulation of EVs. The inherent photothermal ICG facilitated in vivo photothermal imaging (Supplementary Fig. 33). The temperature of saline and free ICG-RNAi prodrugs (35.6 °C and 38.2 °C, respectively) was not sufficient to destroy the tumour tissue, while ICG-RNAi prodrug-loaded EV-treated tumours (44.7 °C) had already reached the damage threshold of tumour ablation (45 °C). In contrast, even though the mice were injected with two-fold ICG-RNAi prodrugs when they were packaged in liposomes than

when they were delivered alone, the local temperature increased to 41.5 °C only, indicating that EV-mediated tumour accumulation contributes to their superior photothermal performance.

Then, the tumour hypoxia-alleviating effect of the bis-RNAi nanosystem was evaluated in MDA-MB-231 tumour-bearing mice. To achieve a reliable comparison of gene-silencing performance, these different RNAi systems were intratumourally injected to guarantee comparable and constant siRNA administration (Supplementary Fig. 34). Compared with other groups, the HCR-twist/HIF-1α siRNA system achieved much more effective downregulation of twist/HIF-1α tumour gene and protein expression, demonstrating the cooperatively enhanced gene-silencing abilities of twist and HIF-1α siRNAs. Subsequently, the miR-21-activated bis-RNAi system was intravenously administered to evaluate the systemic hypoxia-alleviating effect. As expected, the miR-21-assembled siRNAs significantly down-regulated of twist/HIF-1α mRNA (Fig. 4d, e) and protein expression (Fig. 4f). In contrast, the miR-21-assembled scrambled siRNAs (negative control) had little effect on the expression of these mRNAs and proteins, demonstrating the effective gene-silencing ability of our RNAi prodrugs. Interestingly, the gene-silencing efficiency of the miRNA-initiated HCR-promoted RNAi strategy was significantly higher than that of the straightforward siRNA protocol, which might have been attributed to the tumour-specific cascade activation of bis-siRNAs with prolonged gene-silencing duration. Moreover, marked upregulation of PDCD4, PTEN and caspase-3 mRNA and protein expression was observed for the HCR-involved gene regulation systems (Fig. 4g and Supplementary Fig. 35), demonstrating the efficiency of the specific miR-21-initiated RNAi platform. These

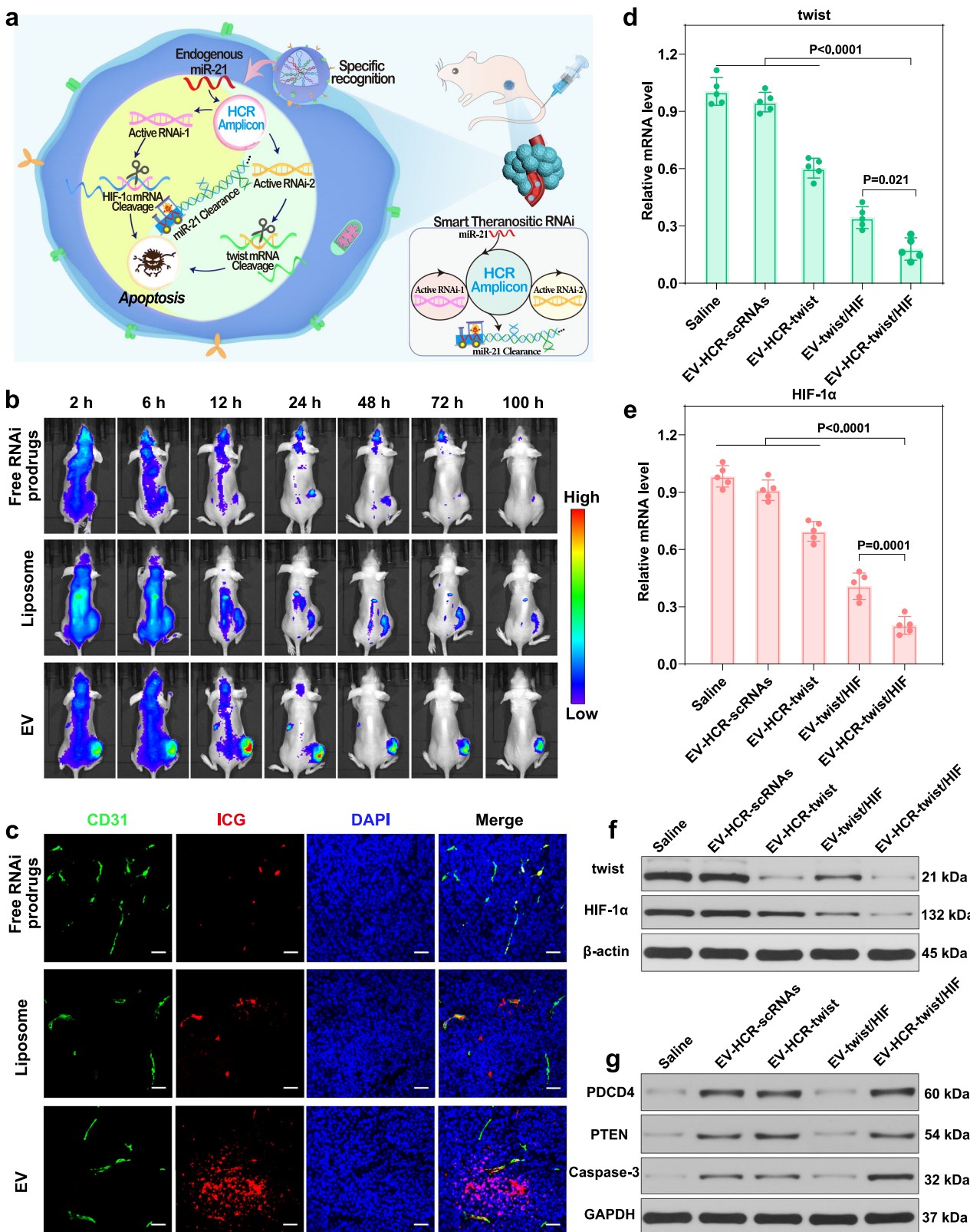

results convincingly demonstrated the specific and effective activation of our RNAi prodrug-packaged EVs in vivo.

**Demonstration of the in vivo therapeutic effect.** To extensively investigate the therapeutic performance of the EV-sustained responsive bis-RNAi nanosystem, we randomly divided MDA-

MB-231 tumour-bearing nude mice (with a tumour volume of ~300 mm³) into different groups and intravenously injected then with different RNAi systems. Based on the initial time-dependent therapeutic evaluation (Supplementary Fig. 36), the mice received systemic administration of various therapeutic agents according to the therapeutic protocol in Fig. 5a. Compared with saline and

**Fig. 4 In vivo biodistribution and gene-silencing efficiency of RNAi prodrug-packaged EVs in MDA-MB-231 tumour-bearing nude mice. a** Trojan EV-sustained programmable cascade activation of multiantenna gene regulation for anticancer theranostics. **b** In vivo distribution of free RNAi prodrugs, liposome-packaged RNAi prodrugs (liposome) or EV-packaged RNAi prodrugs (EV) in tumour-bearing mice under varied administration durations. **c** Colocalization of RNAi prodrugs (red) with CD31-labelled endothelial cells (green) in tumour sections after intravenous injection of free RNAi prodrugs, liposome-packaged RNAi prodrugs (liposome) or EV-packaged RNAi prodrugs (EV) for 12 h. The scale bars are 100 μm. **d**, **e** qRT-PCR analysis and **f** western blot analysis of twist/HIF-1α mRNA and protein expression in mice subjected to different treatments on day five. **g** Western blot analysis of PDCD4, PTEN and caspase-3 protein expression in mice subjected to different treatments as described above. Data represent the mean ± s.d. from five (**d**, **e**) independent replicates. The images (**b**, **c**) and western blot analysis (**f**, **g**) are representative of three independent experiments. Statistical significance (**d**, **e**) is calculated by one-way ANOVA followed by Tukey's post hoc test. Source data are provided as a Source Data file.

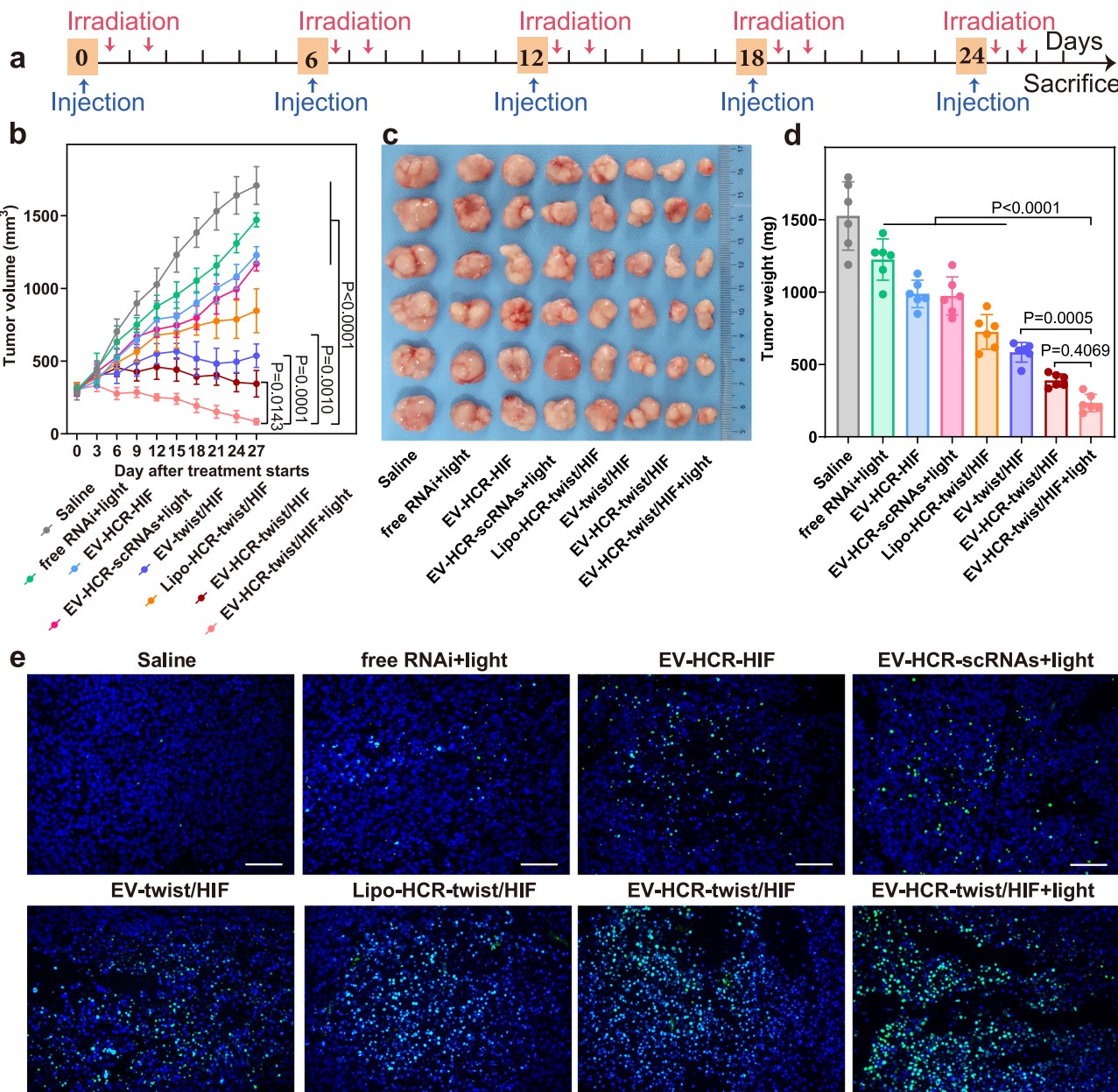

**Fig. 5 In vivo combined bis-gene therapy and PTT for MDA-MB-231 tumour eradication. a** Schematic of the therapeutic protocol used in tumour-bearing mice. **b** Time-dependent tumour growth profiles of mice that were treated with saline, free RNAi prodrugs with photoirradiation, EV-sustained HCR-stimulated HIF-1α, EV-sustained HCR-scrambled-siRNAs with photoirradiation, liposome-sustained HCR-stimulated twist/HIF-1α, EV-sustained direct twist/HIF-1α, EV-sustained HCR-stimulated twist/HIF-1α and EV-sustained HCR-stimulated twist/HIF-1α with photoirradiation. All photoirradiation was carried out with an 808 nm laser (0.7 W/cm²) for 5 min. **c** Representative tumour images and **d** the average tumour weights of mice subjected to the different treatments. **e** TUNEL analysis of the tumours of mice subjected to different treatments. All scale bars are 200 μm. Data represent the mean ± s.d. from six (**b**, **d**) independent replicates. The images are representative of three independent animals. Statistical significance (**b**, **d**) is calculated by two-way ANOVA followed by Bonferroni's multiple comparisons test. Source data are provided as a Source Data file.

photoirradiated free RNAi prodrugs, the bis-RNAi/liposome platform moderately halted tumour progression (Fig. 5b). The bis-RNAi/EV platform exhibited a stronger inhibitory effect than the bis-RNAi/liposome group, which was attributed to the tumour-specific accumulation of EVs with homotypic targeting properties. In addition, the bis-RNAi/EVs system exhibited more obvious tumour suppression than the HIF-1α/EVs system, indicating the enhanced therapeutic performance of our multiantenna gene regulation system. In particular, the bis-RNAi/EV strategy was observed to more efficiently inhibit tumour progression than the straightforward bis-siRNA administration, which might have been attributed to the improved stability of HCR-involved RNAi therapeutic agents. The photoirradiated HCR-assembled scrambled siRNA (photoirradiation control) moderately suppressed the growth of tumours, while the photoirradiated bis-RNAi/EVs showed significant anticancer activity, demonstrating the enhanced therapeutic performance of the combined bis-RNAi and PTT. The therapeutic outcome was also confirmed by photographing and weighing of tumours harvested at day 27 post injection (Fig. 5c, d). The therapeutic performance of these different systems was further validated by TUNEL staining (Fig. 5e), the results of which were in good agreement with these aforementioned observations. Furthermore, no obvious inflammation or disorganisation was observed in the main organs of the mice after administration of our RNAi system (Supplementary Fig. 37), indicating satisfactory systemic biocompatibility of EVs.

**In vivo therapeutic effect on lung metastasis.** Finally, the in vivo antimetastatic efficacy of the EV-sustained miRNA-activated bis-RNAi platform was evaluated in a lung metastatic model established by intravenously injecting MDA-MB-231 cells into mice (Fig. 6a). Saline and HCR-scrambled siRNAs failed to suppress lung metastasis, as revealed by widespread metastatic nodules (Fig. 6b). As expected, the miR-21-triggered HCR-promoted bis-RNAi system resulted in much fewer metastatic foci than the straightforward bis-siRNA system and the bis-RNAi/liposome system. This might have been attributed to the accurate active targeting of EVs and the immediate release/activation of RNAi prodrugs that could achieve more efficient gene silencing upon arriving at metastatic niches. The inhibition rate of different formulations on lung metastasis is shown in Fig. 6c, validating the promoted therapeutic performance of our multiantenna gene-silencing strategy. Moreover, haematoxylin and eosin (H&E) staining assays confirmed that the miRNA-assembled bis-siRNAs were the most effective in inhibiting breast cancer metastasis (Fig. 6d).

## Discussion
HCR was initially explored for amplified intracellular imaging with the assistance of liposomes and inorganic nanocarriers[46–48], although its therapeutic utilisation is still in its infancy. To date, only aptamer-tethered and pH-responsive HCR nanostructures have been utilised for intracellular imaging and as-guided drug delivery[49,50]. A triggered gene-silencing system was recently realised on a long linear DNA scaffold[51], yet it was delivered by a significant amount of synthetic polycationic nanocarriers that may cause undesirable immunogenicity or cytotoxicity. Naturally, secreted EVs have recently emerged as attractive nanocarriers due to their high biocompatibility and stability during systemic administration. In particular, EVs display cell-specific recognition and uptake capabilities by virtue of their intrinsic membrane proteins.

In this study, we realised smart EV-sustained cascade activation of multiantenna siRNA-mediated RNAi therapy by designing

biocompatible EV-sheathed RNAi prodrugs encoded by programmable RNAi administration/activation features. The conditional activation of RNAi was implemented by sophisticated reconstitution of the active siRNA into RNAi-inactive RNA/DNA hybrids. By virtue of the programmed HCR amplifier, the temporally inactive RNAi system could be specifically activated by endogenous miRNA to greatly restore intrinsic RNAi activity. Via hybridisation with DNA, these RNAs could be effectively protected by HCR hairpins that could be further encapsulated into Trojan EVs with more robust tumour-targeting features. After intravenous injection, these EVs exhibit enhanced tumour-specific accumulation and penetration via homotypic targeting, and endogenous miR-21-induced HCR leads to effective tumour recognition. More importantly, the HCR amplifier directs the immediate generation of therapeutic bis-siRNAs for simultaneously silencing the multiantenna hypoxia gene with synergistically promoted therapeutic performance. The photosensitizer ICG effectively facilitates the NIR-irradiated generation of heat in situ for auxiliary phototherapy. Compared to the straightforward siRNA system in vivo, multiple guaranteed tumour-specific stimulated RNAi exhibits enhanced stability and prolonged gene-silencing duration. Based on an RNAi disassembly/reassembly strategy, biomimetic camouflage EV-sustained HCR-amplified cascade activation of multiantenna RNAi is anticipated to pave the way for more versatile and smarter theranostic systems with clinical value.

## Methods
**Materials and reagents.** All oligonucleotides were synthesised and HPLC-purified by Sangon Biotechnology Co., Ltd. (Shanghai, China), and the sequences are listed in Supplementary Table 1. Recombinant murine interleukin 4 (IL-4) and recombinant murine granulocyte-macrophage colony-stimulating factor (GM-CSF) were both obtained from Peprotech (USA). Matrigel was obtained from Corning (356234, USA). GelRed was purchased from Bio-Rad (USA). Lipofectamine 3000, Dulbecco's modified Eagle's medium (DMEM), RPMI-1640, foetal bovine serum (FBS), trypsin, penicillin–streptomycin, Dulbecco's phosphate buffered saline (PBS), calcein-AM/PI double-staining kit and Hoechst 33342 were all obtained from Thermo Fisher. The ICG-NHS ester was obtained from Xi'an ruixi Biological Technology Co., Ltd (China) and ICG-labelled DNA was synthesised from Takara (Japan). The Bulge-Loop$^{TM}$ miRNA qRT-PCR Starter Kit was purchased from RIBOBIO (China). Mir-X miRNA First-Strand Synthesis Kit and 200 bp DNA Ladder (Dye Plus) were acquired from Takara (Japan). Crystal violet (CV) and BCA protein assay kit were obtained from Beyotime (China). Anti-PDCD4 (catalogue number: ab80590), anti-PTEN (catalogue number: ab32199), anti-twist (catalogue number: ab50887), anti-caspase-3 (catalogue number: ab32351) and goat anti-rabbit IgG (catalogue number: ab6721) were acquired from Abcam Biosciences (USA). Anti-HIF-1α (catalogue number: sc-13515), anti-CD63 (catalogue number: sc-5275) and anti-TSG101 (catalogue number: sc-7964) were obtained from Santa Cruz Biotechnology. Anti-calnexin (catalogue number: AC019) was purchased from Beyotime (China). Anti-flotillin 2 (catalogue number: 610383) was obtained from BD Bioscience. Anti-GAPDH (catalogue number: D16H11) and anti-β-actin (catalogue number: 8H10D10) were purchased from Cell Signaling Technology (USA). Other reagents were of analytical grade and were used without further purification.

**Generation of bone marrow-derived dendritic cells (BMDCs).** C57BL/6 (male, 8–10 weeks) were sacrificed and disinfected in 75% ethanol for 10 min. Femurs and tibiae were obtained under sterile conditions. Bone marrow was rinsed out with cold sterile PBS containing 1% FBS. The cells were collected by centrifugation at $2000 \times g$ for 5 min. Then, red blood cell lysis buffer was added to the cell pellet for 5 min to remove red blood cells. The cell suspension was further centrifuged at $2000 \times g$ for 5 min to collect monocytes.

The collected monocytes were filtered through a 40 μm cell strainer (corning, 252340) and cultured in RPMI-1640 medium, supplemented with 10% FBS. The culture medium was supplemented with IL-4 ($10 \text{ ng mL}^{-1}$) and GM-CSF (20 ng $\text{mL}^{-1}$). After 6 days, the loosely adherent BMDCs were harvested by gently pipetting the medium against the flask for further use.

**Cell culture and EVs purification.** Human breast cancer cells (MCF-7), cervical cancer cells (HeLa), OSCC cell (Cal 27), human normal lung fibroblast (MRC-5) and MDA-MB-231 cells were grown in DMEM (MEM for MRC-5) containing 10% FBS and 1% penicillin/streptomycin at 37 °C in humidified 5% $CO_2$ atmosphere.

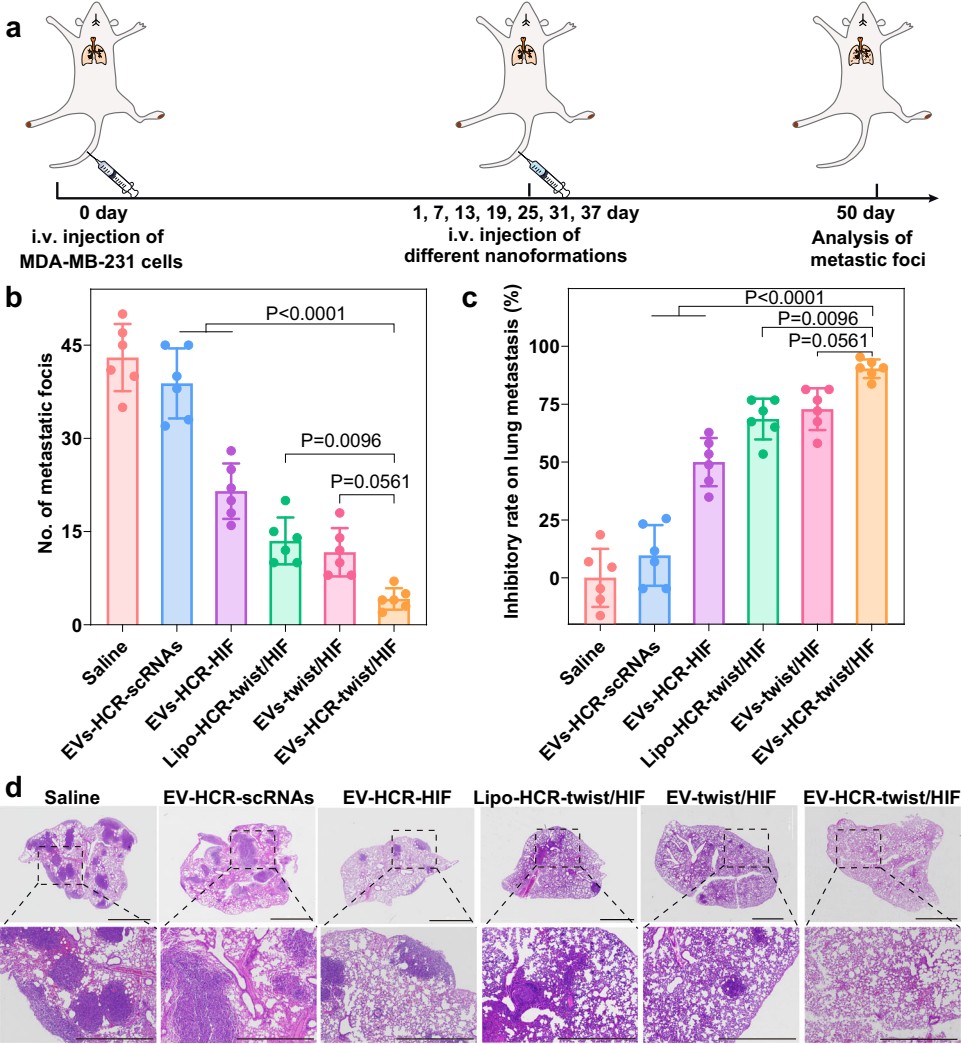

**Fig. 6 Therapeutic effects of RNAi prodrug-packaged nanovectors on lung metastases of breast cancer. a** Therapeutic schedule for EV-sustained RNAi prodrug-mediated inhibition of lung metastases. **b** Quantification of pulmonary metastasis nodes in lungs treated with different formulations. **c** The relative inhibition rate of different formulations compared to saline. **d** Histological examinations of lung tissues collected on day 50 by H&E staining. The scale bars of top panel are 200 μm, while the scale bars of downstream panel are 100 μm. Data represent the mean ± s.d. from six (**b**, **c**) independent replicates. The images are representative of three independent animals. Statistical significance (**b**, **c**) is calculated by one-way ANOVA followed by Tukey's post hoc test. Source data are provided as a Source Data file.

EVs were purified from the supernatants of starved cells by differential centrifugation process[32]. In brief, MDA-MB-231 cells were pre-treated with the miR-21 inhibitor (10 μM) for 48 h, followed by washing thoroughly with PBS and then incubating in serum-free medium for another 48 h for releasing the miR-21-eliminated EVs. The supernatant of cell culture was collected and centrifuged at $2000 \times g$ for 20 min at 4 °C to remove dead cells, followed by spinning at 10,000 g for 30 min at 4 °C to remove cell fragments and microvesicles. EVs were pelleted via ultracentrifugation with a Type 70 rotor (Beckman Coulter, USA) at $130,000 \times g$ for 90 min and washed once in cold sterile PBS. The obtained EVs were re-suspended in sterile PBS and stored at −80 °C for further use.

**Characterisation of EVs**. Total protein content of the obtained EVs was quantified by using BCA protein assay kit (Beyotime, China). The concentration of EVs was quantified by using a nanoparticle tracking analysis (NTA) NS300 system (Malvern, UK). The average EVs yield was 69.2 μg from 30 mL ($2-5 \times 10^7$ cells) of culture supernatant ($n = 8$). The hydrodynamic size and zeta potential of the EVs were measured by a Zetasizer Nano ZS (Malvern, UK). For TEM characterisation, the EVs suspended in sterile PBS were dropped onto a carbon-coated copper grid for 10 min and negatively stained with 2% phosphotungstic acid for 10 s. TEM imaging (Hitachi H7500 TEM, Tokyo, Japan) was carried out to investigate the morphology of the EVs. Meanwhile, protein was extracted from the EVs using RIPA buffer, resolved by SDS-polyacrylamide gels and then transferred to PVDF membrane. The PVDF membrane was blocked with 5% BSA and then incubated overnight at 4 °C with primary antibody. The primary antibodies used included

anti-CD63 (1:1000), anti-TSG101 (1:500), anti-calnexin (1:500) and anti-flotillin 2 (1:500). Next, the PVDF membrane was washed three times with TBST (Thermo Fisher) for 5 min each time, followed by incubation at room temperature for 1 h with diluted secondary antibody (HRP-goat anti-rabbit: 1:2000) in TBST. Following additional washed three times with TBST, these membranes were then incubated with ECL substrate (Bio-Rad), and immediately detected using a ChemiDoc with Image Lab software (Bio-Rad).

**Evaluation of miR-21 in EVs and cells**. The number of EVs was quantified by NTA. The total RNAs were extracted from MDA-MB-231 cells or EVs via TRIzol Reagent Kit (Invitrogen). After these RNAs were quantified by nanodrop (Thermo Fisher), 2 ng of RNAs were introduced into a reverse transcription by using Mir-X miRNA First-Strand Synthesis Kit (Takara). Then, the quantitative PCR analysis was carried out by using the Hieff™ qPCR SYBR® Green Master Mix (Yeasen). The PCR amplification was performed on the CFX96TM Real-Time System where U6 was used as an endogenous normalisation control. The relative miR-21 expression was calculated using the $\Delta C_T$ method.

**RNAi prodrugs loading into EVs**. Electroporation of EVs was performed using the Bio-Rad Gene Pulser Xcell™ Electroporation System. For optimisation, 100 μg EVs and 20 μg RNAi prodrugs were mixed in 400 μL electroporation buffer (Bio-Rad) under the circumstance of 250 voltages, 350 μF capacitance in 0.4 cm cuvettes (Bio-Rad). After electroporation, the RNAi prodrugs-loaded EVs were pelleted via

ultracentrifugation as described above, and the unloaded prodrugs were collected from the supernatant. For in vivo experiments, electroporation was performed in 400 μL electroporation buffer and pooled together for ultracentrifugation. After ultracentrifugation, EVs were fully suspended in cold sterile PBS, kept on ice and conservatively injected into the mice immediately (100 μL per mice). The concentration of free RNAi prodrugs and total input RNAi prodrugs were calculated by UV–vis spectroscopy (Varian Inc.). RNAi prodrug-loaded EVs were analysed by flow cytometric (BD LSRFortessa™). The RNAi prodrugs loading efficiency was calculated through as follows:

$$\text{RNAi prodrugs loading efficiency} = (\text{Input RNAi prodrugs} - \text{free RNAi prodrugs})/(\text{Input RNAi prodrugs})$$

To quantify the RNA/DNA aggregates outside of EVs, the intact and RNase H-treated (10 U, 37 °C, 4 h) EVs pellet were separately isolated to collect the total RNAs. After these RNAs were spiked with varied concentrations of sense RNA ($s_1$), the reverse transcription was performed by using the Bulge-Loop™ miRNA qRT-PCR Starter Kit (RIBOBIO, China). Stability of RNAi prodrugs in EVs was evaluated by flow cytometric analysis (BD LSRFortessa™).

**Native PAGE.** The prepared different samples were loaded into 9% native PAGE and run at a constant voltage of 100 V for 3.5 h in 1×TBE buffer (89 mM Tris, 89 mM boric acid, 2.0 mM EDTA, pH 8.3). After staining with GelRed for 20 min, photos were obtained on a FluorChem FC3 (Protein Simple, USA) Imaging System with an excitation wavelength of 365 nm.

**Construction of miRNA-responsive RNAi prodrugs.** All of the RNA/DNA hybrids were separately mixed in HEPES buffer (10 mM HEPES, 1 M NaCl, 50 mM MgCl₂, pH 7.2) at 4 μM final concentrations and annealed at 95 °C for 5 min, followed by snap cooling down to 25 °C immediately. Then, different concentrations of miR-21 were mixed with the as-prepared RNA/DNA hybrids (100 nM) to initiate the HCR process. To assess the reassociation of RNA/DNA hybrids in vitro, fluorescence measurements were performed on a Cary Eclipse fluorescence spectrophotometer (Varian Inc.). The fluorescence emission spectra were recorded between 550 and 700 nm at an excitation wavelength of 540 nm in a 200 μL quartz cuvette.

For atomic force microscope (AFM) characterisation, sample (20 nM) was deposited on the freshly cleaved mica, and the sample allowed to absorb on the mica surface for 15 min. Then, the mica was rinsing with water for three times and drying under a stream of nitrogen. AFM imaging was performed at room temperature with a tapping mode on Multimode 8 Atomic Force Microscope with a NanoScope V Controller (Bruker Inc.). The silicon tips used for AFM analysis were SCANASYST-AIR (tip radius: ~2 nm; resonance frequency: ~70 kHz; spring constant: ~0.4 N/m; length: 115 μm; width: 25 μm).

**In vitro evaluation of homologous and heterologous uptake efficiency.** CLSM and flow cytometric assay were employed to investigate the cell uptake. For semi-quantitative CLSM observation, MDA-MB-231, MCF-7, HeLa and MRC-5 cells were seeded into confocal dishes and incubated overnight until adherent, respectively. Then, the medium was replaced with the fresh medium containing Cy3-RNAi prodrug-loaded EVs. Cells were washed three times with PBS after 3 h of incubation and stained with Hoechst 33342 for 15 min at 37 °C in humidified 5% CO₂ atmosphere. Cell imaging was performed on the Leica TCS SP8 confocal laser scanning microscope (with the excitation wavelength of 549 nm for Cy3 and 405 nm for Hoechst 33342). For quantitative analysis of the uptake of the EVs, cells were trypsinized and suspended in 500 μL PBS and analysed by flow cytometry (BD FACSVerse, CA, USA).

**Transfection of RNAi prodrugs.** The RNAi prodrugs was prepared in Opti-MEM (30 μL), and was then mixed with lipofectamine 3000 (3 μL) dispersed in Opti-MEM (30 μL) for 5 min. Subsequently, the prepared Opti-MEM transfection mixture was introduced into the plated cells for 3 h at 37 °C. For the in vivo assay, RNAi prodrug (0.5 mg) in Opti-MEM (60 μL) was mixed with lipofectamine 3000 (30 μL) in Opti-MEM (60 μL) for 5 min. Then, the above Opti-MEM transfection mixture was intravenously injected into mice immediately.

**CLSM analysis of endogenous miR-21-initiated HCR.** To assess the reassociation of RNA/DNA hybrids in living cells, measurements were performed using CLSM with 63.0 × 1.40 objective with water. A 549 nm laser accompanying emission ranging from 565 to 600 nm was used as the excitation source of the green channel of fluorophore (Cy3) donor. Acceptor (Cy5) fluorescence image was obtained in red channel with 640 nm excitation accompanying emission ranging from 655 to 700 nm. The FRET signal obtained with 549 nm excitation accompanying emission signal collection ranging from 655 to 700 nm. In order to achieve reliable quantitative FRET readout, the background FRET signal of solely acceptor ($F_A$) or donor ($F_D$) was subtracted from each of these samples. All FRET images of living cells were analysed and processed using Image J (64 bit).

**In vitro cytotoxicity and therapeutic study.** MDA-MB-231 cells were seeded into 96-well plate at the density of $1 \times 10^4$ cells per well and incubated overnight under hypoxia environment. Then, the different formats of the preprocessed EVs were incubated with MDA-MB-231 cells for 4 h. After rinsing three times with PBS to remove the redundant EVs, the cells of PTT group were irradiated with an 808 nm laser (0.7 W/cm², 5 min). After incubation for another 48 h, 100 μL of MTT solution (1 mg mL⁻¹ in PBS) was added to each well and further incubated for another 4 h at 37 °C. After that, the supernatant medium was carefully removed and the formazan crystals were dissolved in 150 μL DMSO with gently shaking for 15 min. The absorbance at the wavelength of 490 nm was measured with a microplate reader (Thermo Scientific). The cell viability was calculated as

$$\text{Cell Viability} = (\text{OD}_{\text{Treated}} - \text{OD}_{\text{Blank}}/\text{OD}_{\text{Control}} - \text{OD}_{\text{Blank}}) \times 100\%$$

The inherent relevancy among these different therapeutic strategies was assessed by CI through the Chou–Talalay analysis[52]. By plotting the cell viability vs the concentration of RNAi prodrugs, the CI value can be obtained in CalcuSyn software.

For live/dead assay, calcein-AM/PI double-staining were used to label cells for 15 min and washed three times with PBS. Then, the fluorescence images were performed on inverted fluorescence microscope (Leica DFC7000 T, Germany).

**In vitro migration and invasion assay.** For transwell migration assay, these differently treated MDA-MB-231 cells ($2.5 \times 10^4$, for 4 h) were plated in the top chamber with the non-coated membrane (24-well insert; pore size, 8 μm; Corning). For invasion assay, MDA-MB-231 cells ($5 \times 10^4$, for 4 h) were plated in the top chamber with the Matrigel-coated membrane (24-well insert; pore size, 8 μm; Corning). In both assays, MDA-MB-231 cells were re-suspended in DMEM without serum, and the 20% serum-supplemented DMEM was used as a chemoattractant in the lower chamber. These cells were incubated under hypoxic environment for 24 h, and then the residual non-migrated or non-invaded cells were removed by a cotton swab. After staining with CV, the lower membrane surface-fixed cells were counted and imaged through inverted microscope.

**Animal study.** All animal experimental protocols were approved by the Institutional Animal Care and Use Committee of the Animal Experiment Center of Wuhan University (Wuhan, China). All animal experimental procedures were performed in accordance with the Regulations for the Administration of Affairs Concerning Experimental Animals approved by the State Council of People's Republic of China. C57BL/6 mice (male, 6–8 weeks) and BALB/c nude mice (female, 4-6 weeks) were both purchased from Charles River Company and raised in a specific pathogen-free grade laboratory according to guidelines for laboratory animals established by the Wuhan University Center for Animal Experiment/A3-Lab (No. 00270444). All animals were housed with a 12 h light/dark cycle at 22 °C, 40% relative humidity and food and water ad libitum.

**Hemocompatibility assay.** Fresh blood was obtained from BALB/c mouse. RBCs were obtained by centrifuging at $800 \times g$ for 10 min. The RBCs were then washed several times with cold PBS until the supernatant was colourless. Then, 500 μL of 2% RBCs were mixed with 500 μL RNAi prodrug-packaging EVs with different concentrations. PBS and deionized water (DI) were used as negative and positive control, respectively. After incubation for 180 min, the solution was centrifuged at $800 \times g$ for 10 min and the absorbance spectra of the supernatant were measured at 570 nm. Haemolysis percentage was calculated as

$$\text{Hemolysis}(\%) = ((A_{\text{sample}} - A_{\text{negative}})/(A_{\text{positive}} - A_{\text{negative}})) \times 100\%$$

**SEM imaging of RBCs.** The 2% RBCs suspension (500 μL) was added to 500 μL of RNAi prodrugs-packaging EVs to a final concentration of 500 μg/mL and incubated 6 h at room temperature. Then, the RBCs were fixed by 2.5% glutaraldehyde for 30 min at 4 °C. The RBCs were dehydrated with ethanol (15, 30, 40, 50, 70, 80, 90, 100%) for 5 min orderly. Finally, the RBCs suspensions were dropped onto silicon slice and coated with Pt before viewing under a scanning electromicroscope (SEM, Hitachi, Japan).

**Biocompatibility evaluation of EVs in vivo.** Healthy C57BL/6 mice (male, 6–8 weeks) were injected intravenously with bare EVs (5 mg). Mice serum was then obtained by centrifuging whole blood taken by retro-orbital venous puncture at different time points. The serum interleukin-6 and tumour necrosis factor-α were evaluated by using the ELISA kits (Thermo Fisher). Meanwhile, the healthy female BALB/c nude mice were intravenously injected with saline or EVs (5 mg) every week for 5 times. After 6 weeks, the mice were sacrificed and the blood was collected for haematology and blood biochemistry analysis.

**In vivo multimodal imaging.** Mice were subcutaneously injected with MDA-MB-231 cells ($2 \times 10^6$ cells) in the right hind leg. Once tumours reached an approximate size of 300 mm³, mice were injected with ICG-labelled free RNAi prodrug, ICG-labelled RNAi prodrug-loaded liposome or ICG-labelled RNAi prodrug-packaged EV through a tail vein. At 2, 6, 12, 24, 48, 72 and 100 h post administration, the

whole-body distribution of RNAi prodrug was investigated via an IVIS imaging system. For in vivo photothermal imaging, the tumour sites were irradiated under 808 nm laser at a power density of 0.7 W/cm² for 5 min, and infra-red thermal images were recorded by an infra-red thermal camera.

**qRT-PCR analysis of miR-21**. The total RNAs were extracted in MDA-MB-231, MCF-7, HeLa and MRC-5 via TRIzol Reagent Kit (Invitrogen) by following the manufacturer's instruction. Meanwhile, the MDA-MB-231 tumour tissues were dissected out for total RNA extraction with TRIzol Reagent Kit. The cDNA were prepared by using Mir-X miRNA First-Strand Synthesis Kit (Takara) according to the indicated protocol. The 3′ primer for qPCR is the mRQ3′ Prime supplied with the kit and the sequences of 5′ primer are provided in Supplementary Table 2. Quantitative PCR analysis was carried out using the Hieff™ qPCR SYBR® Green Master Mix (Yeasen). PCR amplification was performed on the CFX96™ Real-Time System. U6 was used as an endogenous normalisation control.

**Detection of twist, HIF-1α, PDCD4, PTEN and caspase-3 at mRNA levels in vivo**. The MDA-MB-231 tumour region was carefully separated from mice and homogenised. For qRT-PCR assay, Trizol was added to extract total RNAs from tumour tissue. The cDNA was synthesised using a PrimeScript first-strand cDNA synthesis kit (Bio-Rad) by following the manufacturer's protocol. Quantitative PCR analysis was performed using the Hieff™ qPCR SYBR® Green Master Mix (Yeasen). PCR amplification was performed on the CFX96™ Real-Time System. The primers were purchased from Sangon Biotech. Co., Ltd. (Shanghai, China) and the sequences are listed in Supplementary Table 2.

**Detection of twist, HIF-1α, PDCD4, PTEN and caspase-3 at protein levels in vivo**. For western blot analysis, the tumour tissues were lysed using RIPA lysis buffer and centrifuged at 10,000 × g for 15 min at 4 °C. The supernatants were collected and quantified via BCA protein assay kit, separated by SDS-PAGE and then transferred to the PVDF membrane. After blocking with 5% BSA for 1 h, the blots were probed with the primary antibodies (rabbit-twist: 1:500; rabbit-HIF-1α: 1:500; rabbit-PTEN: 1:1000; rabbit-PDCD4: 1:1000; rabbit-caspase-3: 1:1000) in blocking buffer at 4 °C overnight. Next, the PVDF membrane was washed three times with TBST for 5 min each time, followed by incubation at room temperature for 1 h with diluted secondary antibody (HRP-goat anti-rabbit: 1:2000) in TBST. Following additional washed three times with TBST, these membranes were then incubated with ECL substrate (Bio-Rad) and immediately detected using a ChemiDoc with Image Lab software (Bio-Rad).

**Synergistic therapy in vivo**. BALB/c nude mice (female, 4–6 weeks) bearing subcutaneous MDA-MB-231 tumours (~300 mm³) were divided into eight groups (n = 6 per group) and were treated with saline, free RNAi prodrugs with photo-irradiation, EV-sustained HCR-stimulated HIF-1α, EV-sustained HCR-scrambled-siRNAs with photoirradiation, liposome-sustained HCR-stimulated twist/HIF-1α, EV-sustained direct twist/HIF-1α, EV-sustained HCR-stimulated twist/HIF-1α and EV-sustained HCR-stimulated twist/HIF-1α with photoirradiation. The mice received intravenous administration of various therapeutic agents (0.5 mg RNAi prodrugs per mice equivalent) for five times and the photoirradiation was conducted after 12 and 30 h of injection. At the 27 days, all mice were euthanized for evaluating the therapy efficiency.

**Anti-metastasis effect of multiantenna gene regulation**. At 24 h after MDA-MB-231 cells (1 × 10⁶ per mouse) were intravenously injected into BALB/c nude mice (female, 4–6 weeks), the mice were intravenously administrated with saline, EV-sustained HCR-scrambled-siRNAs, liposome-sustained HCR-stimulated twist/HIF-1α, EV-sustained direct twist/HIF-1α and EV-sustained HCR-stimulated twist/HIF-1α at dosage of 0.5 mg per mice equivalent (n = 6 per group). All mice groups were administrated five times at 6-day intervals. The body weight and tumour volume of the mice were measured every 3 days until the completion of treatment. At the 50 days, the mice were euthanized and the lungs were acquired. The numbers of tumour nodules on the surface of lungs were recorded. Lungs were then fixed with 4% paraformaldehyde, sectioned and examined by H&E staining.

**Statistical analysis**. The comparison between two groups was investigated by using unpaired two-tailed Student's t test, while the one-way ANOVA or two-way ANOVA was used for comparison of more than two groups. Values with p < 0.05 were considered as statistically significant. All of these data analyses were conducted using GraphPad 8.0 software without excluding any samples.

**Reporting summary**. Further information on research design is available in the Nature Research Reporting Summary linked to this article.

## Data availability

The source data underlying Figs. 2a, 3a, c, d, 4d–g, 5b, d and 6b, c and Supplementary Figs. 2–4, 10a, c, 11a–c, 12a, 13, 15a, d, e, 16, 17b, c, 18b, c, 19b, c, 20, 21b, c, 24, 28a, 29,

30, 31, 32b, c, 34 and 35 are provided as a Source Data file. All the relevant data are available from the authors upon reasonable request. A reporting summary for this article is available as a Supplementary Information file. Source data are provided with this paper.

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

## Acknowledgements

This work was supported by the National Natural Science Foundation of China (21874103 and 22074112) and Fundamental Research Funds for the Central Universities (2042019kf0206).

## Author contributions

F.W. and X.L. conceived and designed the experiments. X.G. and R.L. performed the experiments. K.T. synthesised miR-21 inhibitor. K.T. and C.H. collected EVs. H.W. and J.L. provided guidance for animal experiments. H.W. preformed western blot analysis in vivo. J.Wei. collected fluorescence spectra. C.H., J.Wang. and J.S. performed flow cytometry analysis. X.G. and F.W. wrote the manuscript. All authors have given approval to the final version of the manuscript.

## Competing interests

The authors declare no competing interests.
