## [Peer Review File · Nature Communications]

Reviewers' Comments:

Reviewer #1:

Remarks to the Author:

Gong et al. developed a tumorous biomarker-activated RNAi platform through the delicate design of RNAi prodrugs in EVs with cancer-specific recognition/activation features. These RNAi prodrugs were designed ingeniously. And obvious anti-tumor effects were also indicated. However, there are still some drawbacks needed to be promoted.

- 1) According to method, the prepared EVs were exosomes. But the average diameter is ~221 nm, which are almost 2 times than that in TEM images. In generally, diameters measured by DLS were little larger than that of TEM images. Was it possible that the large diameter was induced by additional microparticles (100-1000 nm) in author's samples.
- 2) The loading efficiency was 0.14 µg of RNAi prodrug per µg of protein. Could the author compare it with presented carriers in manuscript?
- 3) In figure 3D, cell cytotoxicity was evaluated. The results of group g and h were similar, and it is hard to declare more cytotoxicity were observed in group h.
- 4) For in vivo biodistribution, tumors of 50 mm³ were used. It's too small. And also, in the evaluation of hypoxia-alleviating efficiency, when tumors were only 50 mm³, the hypoxia microenvironment was negligible, which might lead to no effects of the RNAi system.
- 5) Why intratumoral injection was administrated to evaluate the extent of gene silencing? While intravenous injection was applied in other experiments.
- 6) In figure 5, could the authors provide some evidences for the therapeutic protocol, the laser was too frequency.
- 7) In figure 5, the initial tumor size was too small for tumor inhibition. According to the anti-tumor results (Fig 5B), contribution of siRNA to tumor inhibition was weak, photothermal therapy induced more obvious tumor eradication. Incomplete microenvironment of small tumors might be responsible. could the author provide anti-tumor results in big tumors (about 300 mm³) which possess more developed tumor microenvironment.
- 8) In figure S18, group h was missed.

Reviewer #2:

Remarks to the Author:

The manuscript by Gong and co-workers employed the extracellular vesicles to load the RNAi prodrug for gene therapy. The RNAi drug (siRNA) can be generated from the RNAi prodrug through HCR amplification initiated by intracellular miRNA. After the combination with PTT, the drug delivery system demonstrated efficient tumor therapy. Generally, it is an interesting work. There are some major concerns that should be addressed.

1. As shown in Figure 1, the siRNA was divided into two parts (guide strand and passenger strand). After the endocytosis, the siRNA will be recovered from the RNA-DNA hybrid (RNAi prodrug) through HCR amplification. However, as we know, the intracellular RNase H is responsible for the digestion of the RNA strand in the RNA-DNA hybrid, which is the principle of antisense therapy. The authors should investigate how much the siRNA strands are still retained in the intracellular environment. Will the PTT treatment damage the siRNA?
2. As shown in Figure 4A, the intracellular trafficking pathway of the delivery system assisted by extracellular vesicles should be studied in detail. Where does the HCR occur, in or outside the EVs? If in the EVs, how does the miRNA enter? If outside the EVs, how to protect the RNA strands from RNase H?
3. The size of EV and the amount of siRNA in each EV are distributed in a certain range. The authors should provide a detailed protocol for the quantification of the siRNA in each injection.
3. In Figure 4B, it is confused to find that much lower fluorescence intensity was observed in the group of EVs in the whole body than the other two controls at 2h. This low fluorescence may indicate that the nanocarriers have been almost metabolized during the short 2 hours. At the same time, the fluorescence intensity of EVs in the tumor region is also very low at 2h.

4. In Figure 4D, a noticeable higher MFI was found in the kidney for EVs compared with the control groups. The authors should discuss the possible side effects.

5. In Figure S18, group h is missing.

Reviewer #3:

Remarks to the Author:

General:

- In this paper the authors developed a biomarker-guided EV-based therapeutic system for cancer, through the incorporation of RNAi prodrugs, that are only activated the presence of miR21 (tumour biomarker). Additionally, to achieve photothermal therapy, the authors combined indocyanine green (ICG), which further improved the therapeutic efficacy of the developed system upon photoirradiation. The exceptionally positive results observed both in vitro and in vivo, as well as the potential to reduce the RNAi off-target effects, makes this paper important to others in the field.
- Although the manuscript is novel and interesting, it is very difficult to read. English needs to be significantly improved, and the text needs to be rearranged so that the message is clear and concise.
- A discussion of the results obtained in the context of previous literature is missing.
- Furthermore, I have several concerns, as described below.

Major points:

1. The authors should provide more background on the use of HCR amplification strategies towards therapeutics. Specifically, the authors should mention whether this kind of strategy was previously tested using other nanocarriers, advantages/disadvantages of their system compared to others (if applicable), whether this was tested in other disease contexts or not.
2. EVs need to be further characterized. According to the MISEV 2018 guidelines, it is recommended to detect in EVs – 1) the presence of specific proteins (at least 3, including transmembrane/ lipid-bound proteins and cytosolic markers) and 2) absence or significant reduction of negative markers (e.g. calnexin).
3. Electroporation can induce precipitation of siRNAs, and thus of RNA/DNA hybrids as well (please check this reference <https://pubmed.ncbi.nlm.nih.gov/23994516/>). This needs to be taken into consideration throughout the manuscript. An additional control for electroporation, repeating the protocol with RNA/DNA hybrids in the absence of EVs, should also be performed, so that the extent of RNA/DNA aggregate formation can be evaluated.
4. Nude mice lack a thymus, and are unable to produce T cells. Therefore, to be informative, the immunogenicity of the developed system needs to be tested in immunocompetent mice. Furthermore, the levels of cytokines should be evaluated a few hours after injection, as elevations in serum cytokines are transient and typically resolve within 12-24 hours after initial treatment.
5. A detailed description of the liposomes used in their studies, and how these were conjugated with HCR hairpins, is lacking. Furthermore, if using lipofectamine 3000, at least for in vivo studies, the authors cannot argue that EVs are more efficient, as there are several improved liposomal preparations that perform much better in vivo (e.g. stable nucleic acid lipid particles). This should be added to discussion.
6. The authors mention throughout the manuscript the specificity, absence of immunogenicity and lack of off-target effects of the developed system. These claims are not fully proven and therefore these should be suggestions rather than certainties.
7. The authors show that MDA-MB-231 cells exhibit higher uptake for MDA-MB-231 EVs than other cells e.g. MCF-7. In contrast, it was previously reported higher transfer of EV cargo from MDA-MB-231 EVs to different target cells, including MCF-7 (<https://www.sciencedirect.com/science/article/pii/S0092867415004985?via=ihub>). This is an important point for discussion.
8. In materials and methods the authors mention that all data were analysed using Student's t-test, but in most, if not all cases, there are more than two different groups of treatment. The authors should rectify the statistical analysis throughout the manuscript.
9. The authors use different names for the RNA/DNA hybrids throughout the manuscript (e.g. RNAi

prodrugs, HCR hairpins, siRNA/DNA hybrids, siRNA/DNA prodrugs, ...). Can the authors adopt one term while presenting the results and use it throughout the rest of the manuscript, so that the reader can follow the paper more easily?

Minor points:

1. Can the authors add a brief description of the scheme represented in Figure 1 to the legend?
2. Why are the isolated EVs so big (> 200 nm), considering that bigger EVs were eliminated by UC at 10,000 g? Any explanation?
3. The authors suggest that the RNAi prodrug is encapsulated in EVs (Page 10, Fig. S8) due to a tolerance for nuclease degradation. But exposure to DNase/RNases should be tested to prove their point. The explanation for the stability in 10% FBS at 37°C might be due to the higher thermodynamic stability of DNA/RNA hybrids, as the authors previously mentioned, and it does not mean that the RNAi prodrug is encapsulated in EVs.
4. The authors refer in page 10 that the EVs are safe and do not transfer large amounts of oncogenic miRNA. It would be important to evaluate the miR21 levels in EVs relative to the cells (MDA-MB-231).
5. In Fig. S9D, are the authors sure about the units for TNF- α levels? Shouldn't it be pg/ml?
6. In Fig. S12C, the transfection efficiency is relative to what? (considering that efficiency in MDA-MB-231 is less than 100%)
7. In Fig. S13 it seems the authors used ICG-pDNA for liposomes and Cy5-RNAi prodrug for EVs. How can they compare the signal if they used different dyes and different cargoes? (although they mention something different in Fig. S13A and C). Did they load pDNA or the RNAi prodrug in EVs?
8. The expression levels of miR21 in different cells are relative to what (Fig. S15)?
9. In Figure 3A the authors mention in the legend, that cells were incubated with DNA-prodrug packaging EVs, but they must be using RNA/DNA hybrids.
10. In Fig. S18 one group – group h – is missing.
11. The photothermal activity of ICG should be explained in more detail in page 15, before the obtained results are explained.
12. How were the RNA/DNA hybrids functionalized with ICG? There is no reference to this anywhere, and this protocol is very important to enable data reproducibility.
13. It is not clear if the RNA/DNA hybrids are functionalized with ICG in Fig. S18. Legends should reflect this information. An additional control using EVs HCR-scramble siRNAs subjected to photoirradiation should also be added, as it would be important to evaluate the effect of photoirradiation by itself.
14. A description how the combination index in Fig. 3B was calculated should be added to Materials and Methods.
15. Again, in Fig. S20, S21 and Fig. 3, an additional control using EV HCR scramble-siRNAs with photoirradiation is lacking. The presence of ICG in the respective groups, should be included in the figure legends.

Reviewer #1:

Gong et al. developed a tumorous biomarker-activated RNAi platform through the delicate design of RNAi prodrugs in EVs with cancer-specific recognition/activation features. These RNAi prodrugs were designed ingeniously. And obvious anti-tumor effects were also indicated. However, there are still some drawbacks needed to be promoted.

1) According to method, the prepared EVs were exosomes. But the average diameter is ~221 nm, which are almost 2 times than that in TEM images. In generally, diameters measured by DLS were little larger than that of TEM images. Was it possible that the large diameter was induced by additional microparticles (100-1000 nm) in author's samples.

Response: Thank you very much for your insightful comments. After a careful examination of the samples, we found that the large size of dynamic light scattering (DLS) measurement was might attributed to the incomplete dispersion in PBS after ultracentrifugation. The obtained EVs were collected by gently pipetting the PBS medium against the pellet at the bottom of the ultratube. The EVs characterization was thus updated with more in-depth exploration on their stabilities in the revised **Figures S10** and **S11**. DLS characterization revealed that the native EVs had an average diameter of ~132 nm in our revised **Figure S10A**. Note that the EVs size of transmission electron microscopy (TEM) image might not always represent the actual size distribution in the full population. As shown in our revised **Figure S10B**, a typical cup-like morphology of the isolated EVs was observed by TEM. No obvious aggregation and size distribution change was observed for EVs with 3 cycles of freeze-thaw operations as determined by DLS and TEM (revised **Figure S11**), indicating the sufficient biostability of the EVs nanocarrier. After the RNAi prodrugs were transfected into EVs by electroporation, the loaded EVs displayed a slight increasement in average size (146 nm) yet without noticeable morphologic changes (revised **Figures 2B** and **2C**). These updated characterizations of EVs were clarified in the revised manuscript and supporting information. We hope that the present revision is now acceptable.

Figure S10. Characterization of EVs derived from MDA-MB-231 cells. (A) Size distribution of EVs. (B) TEM characterization of EVs. (C) Western blot analysis of EVs and the corresponding cell lysates. (D) qRT-PCR analysis the content of miR-21 in EVs vs. its expression in MDA-MB-231 cells with miR-21-inhibitor pre-treatment (normalized to U6 RNA). Results are presented as means \pm standard deviation (SD) (n=5).

Figure S11. Characterization of EVs that were treated with multiple freeze-thaw cycles. Size distribution of EVs after one freeze-thaw cycle (A), two freeze-thaw cycles (B) and three freeze-thaw cycles (C). (D) Representative TEM image of EVs after three freeze-thaw cycles.

Figure 2. Characterization of the RNAi prodrugs-packaged EVs. (A) Schematic illustration of the RNAi prodrugs-encapsulated EVs. (B) Hydrodynamic diameter, (C) TEM image, and (D) flow cytometric analysis of RNAi prodrugs-loaded EVs. (E) Diagnostic EVs-sustained HCR-amplified FRET analysis of intracellular miR-21 (in the form of F_A/F_D) and corresponding FRET signal distributions of each pixel in different cell lines: (a) MDA-MB-231, (b) MCF-7, (c) HeLa and (d) MRC-5 cells. All scale bars correspond to 20 μm .

2) The loading efficiency was 0.14 μg of RNAi prodrug per μg of protein. Could the author compare it with presented carriers in manuscript?

Response: Thank you very much for your insightful investigation, which we appreciated very much. According to your suggestion, a brief summary of the present siRNA delivery system was supplemented for comparing the present EVs carrier with other materials in the newly supplemented **Table S3**. Our EVs revealed a comparable and even higher loading efficiency than some of the existing siRNA delivery carriers.

Table S3. Summary of different nanocarriers for siRNA delivery

siRNA delivery Systems	Loading efficiency	Ref.
PLGA nanoparticles	50 pmol per mg of PLGA	1
Mesoporous silica-coated polypyrrole nanoparticles	500 fmol per μg of nanoparticles	2
Gold nanoclusters	226 μmol per g gold nanoclusters	3
Gold nanorods	70 pmol per μg of gold nanorods	4
EVs	1.3 pmol per μg of EVs	Present study

Reference

1. Woodrow, K. A., Cu, Y., Booth, C. J., Saucier-Sawyer, J. K., Wood, M. J., Saltzman, W. M. Intravaginal Gene Silencing Using Biodegradable Polymer Nanoparticles Densely Loaded With Small-Interfering RNA. *Nat. Mater.* **8**, 526-533 (2009).
2. Guo, R., Tian, Y., Yang, Y., Jiang, Q., Wang, Y., Yang, W. A Yolk-Shell Nanoplatform for Gene-Silencing-Enhanced Photolytic Ablation of Cancer. *Adv. Funct. Mater.* **28**, 1706398 (2018).
3. Lei, Y., Tang, L., Xie, Y., Xianyu, Y., Zhang, L., Wang, P., Hamada, Y., Jiang, K., Zheng, W., Jiang, X. Gold Nanoclusters Assisted Delivery of NGF siRNA for Effective Treatment of Pancreatic Cancer. *Nat. Commun.* **8**, 15130 (2017).
4. Bonoiu, A. C., Mahajan, S. D., Ding, H., Roy, I., Yong, K.-T., Kumar, R., Hu, R., Bergey, E. J., Schwartz, S. A., Prasad, P. N. Nanotechnology Approach for Drug Addiction Therapy: Gene Silencing Using Delivery of Gold Nanorod-siRNA Nanoplex in Dopaminergic Neurons. *Proc. Natl. Acad. Sci. U.S.A.* **106**, 5546-5550 (2009).

3) In figure 3D, cell cytotoxicity was evaluated. The results of group g and h were similar, and it is hard to declare more cytotoxicity was observed in group h.

Response: Thank you very much for your insightful comments. According to your suggestion, the cell cytotoxicity of different systems was re-examined by using

live/dead cell analysis (revised **Figure 3E**), quantitative MTT assay (revised **Figure S29**), Transwell migration (newly supplemented **Figures 3C** and **S30A**) and Matrigel invasion assays (revised **Figures 3D** and **S30B**). The cell viability of liposome- or EVs-treated MDA-MB-231 cells was investigated by using live/dead cell analysis, where live and dead cells were stained with calcein AM and propidium iodide (PI), respectively. As expected, the therapeutic efficacy of photoirradiated bis-RNAi approach is slightly higher in EVs than in liposomes (group i vs group h of the revised **Figure 3E**), as evidenced by the green fluorescence of MDA-MB-231 cells. Noteworthy, the live/dead cell visualization provides the intuitive identification of the varied therapeutic efficiency of different treatment groups. The slightly higher therapeutic performance of EVs, as compared with that of liposome, was initially demonstrated by the more precise and quantitative MTT assay (group i vs group h in the revised **Figure S29**). In addition, the Transwell migration and Matrigel invasion assays revealed that these bis-RNAi-EVs-treated MDA-MB-231 cells displayed a comparably higher decreased motility and invasiveness than liposomes (newly supplemented **Figures 3C**, **S30A**, and revised **Figures 3D**, **S30B**). Taking together, these results suggest that the therapeutic efficiency of bis-RNAi strategy is higher in EVs than in liposome, which validates the cell-specific homotypic targeting of EVs from donor to recipient cells. These updated experimental demonstrations were supplemented in the revised manuscript and supporting information with appropriate descriptions. We hope that the present revision is acceptable.

Figure 3. Cell cytotoxicity evaluation of RNAi prodrugs-packaged EVs in hypoxic condition. (A) Cell viability of MRC-5, HeLa, MCF-7 and MDA-MB-231 cells after their incubation with functional EVs encapsulating with different

concentrations of RNAi prodrugs (25, 50, 75 and 150 nM) for 48 h. Results are presented as means \pm standard deviation (SD) (n=5). **P < 0.01, ***P < 0.001, ****P < 0.0001 (two-way ANOVA with Bonferroni's multiple comparisons test). ns, not statistically significant. (B) Combination index (CI) values of RNAi prodrugs-packed EVs with and without photoirradiation. (C) Transwell migration assay, (D) Matrigel invasion assay, and (E) live/dead cells analysis of MDA-MB-231 cells with different treatments. (a): saline; (b): EVs-sustained HCR-scrambled-siRNAs; (c): EVs-sustained HCR-stimulated twist; (d): EVs-sustained HCR-stimulated HIF-1 α ; (e): EVs-sustained HCR-scrambled-siRNA with photoirradiation; (f): EVs-sustained direct twist/HIF-1 α ; (g): EVs-sustained HCR-stimulated twist/HIF-1 α ; (h): liposome-sustained HCR-stimulated twist/HIF-1 α with photoirradiation; (i): EVs-sustained HCR-stimulated twist/HIF-1 α with photoirradiation. All photoirradiation was carried out with 808 nm laser (0.7 W/cm²) for 5 min. Results are presented as means \pm standard deviation (SD) (n=3). *P < 0.05, **P < 0.01, ***P < 0.001, ****P < 0.0001 (one-way ANOVA with a Tukey post hoc test for (C) and (D)).ns, not statistically significant.

4) For in vivo biodistribution, tumors of 50 mm³ were used. It's too small. And also, in the evaluation of hypoxia-alleviating efficiency, when tumors were only 50 mm³, the hypoxia microenvironment was negligible, which might lead to no effects of the RNAi system.

Response: Thank you very much for your insightful comments. We agree that the 50 mm³ tumors were too small to evaluate the hypoxia-alleviating efficiency of our siRNA interfering system (revised **Figures 4** and **S31**). According to your suggestion, the larger 300 mm³ tumor was adapted for exploring the therapeutic effects of the RNAi system. Here the ~300 mm³ tumor-bearing mice were respectively received tail-vein injections of free ICG-RNAi prodrugs, ICG-RNAi prodrugs-loaded liposome and ICG-RNAi prodrugs-packaged EVs. And these different treated mice were examined using an *in vivo* optical imaging system at various time points thereafter. As shown in revised **Figure 4B**, the intravenously injected free RNAi prodrugs were

cleared out without obvious accumulation in tumor site. As compared with the liposome group, the EVs could progressively accumulate in tumor tissue and reached to a maximal accumulation after 12 h. In addition, the endothelial cells of tumor vessels was stained through CD31 biomarker to evaluate the penetration capability of EVs in deep tumor tissue. As compared with the free RNAi prodrugs and liposome-delivered RNAi prodrugs, the EVs-delivered RNAi prodrugs showed a much weaker co-localization with tumor vessels (revised **Figure 4C**), implying that the EVs carriers can effectively extravasate from tumor vessels for penetrating into tumor tissue. We also evaluated their time-dependent distribution in main organs and tumor tissue. After normalizing the radiation efficiency by organ weight, the EVs displayed a higher ratio of tumor to liver or kidney (revised **Figure S31**). These *ex vivo* results were consistent with the whole-body imaging observations, thus demonstrating the favorable tumor accumulation of EVs. Due to the limited space, we transferred the *ex vivo* data into the revised supporting information. These as-suggested *in vivo* bio-distribution experiments have been supplemented in the revised manuscript and supporting information with appropriate discussions.

Figure 4. *In vivo* biodistribution and gene silencing efficiency of RNAi prodrugs-packaged EVs in MDA-MB-231 tumor-bearing nude mice. (A) Construction of Trojan EVs-sustained programmable cascade activation of multi-antenna gene regulation for anti-cancer theranostic application. (B) *In vivo* distribution of free RNAi prodrugs, liposome-packaged RNAi prodrugs or EVs-packaged RNAi prodrugs in tumor-bearing mice under varied administration durations. (C) Colocalization of RNAi prodrugs (red) with CD31-labeled endothelial cells (green) in tumor sections after intravenous injection of free RNAi prodrugs, liposome-packaged RNAi prodrugs or EVs-packaged RNAi prodrugs for 12 h. Scale bars, 100 μm. (D, E) qRT-PCR analysis and (F) western blot analysis of twist/HIF-1α mRNAs and proteins in mice treated with (G1) saline, (G2) EVs-sustained

HCR-scrambled-siRNAs, (G3) EVs-sustained HCR-assembled twist, (G4) EVs-sustained direct twist/HIF-1 α , or (G5) EVs-sustained HCR-assembled twist/HIF-1 α on day five. Results are presented as means \pm standard deviation (SD) (n=5). *p<0.05, **p<0.01, ****p<0.0001 (one-way ANOVA with a Tukey post hoc test for (D) and (E)). (G) Western blot analysis of PDCD4, PTEN and Caspase-3 proteins in these differently treated mice as described above.

Figure S31. Tissue distribution analysis. (A) *Ex vivo* representative fluorescence images of major organs after administration of free RNAi prodrugs, RNAi prodrugs-loaded liposome or RNAi prodrugs-packaged EVs to tumor-bearing mice at the indicated time points. H, heart; Li, Liver; S, spleen; Lu, lung; K, kidney; T, tumor. Quantitative analysis of the biodistribution in tumor to kidney (B) and tumor to liver ratio (C) at the indicated time points. Results are presented as means \pm standard deviation (SD) (n=3).

5) Why intratumoral injection was administrated to evaluate the extent of gene

silencing? While intravenous injection was applied in other experiments.

Response: Thank you very much for your instructive comments. The intratumoral injection was used to provide the robust identification of the varied therapeutic efficiency of different treatments. Under the constant administration of RNAi prodrugs, then the varied therapeutic performance of these different RNAi nanoplateforms could be reliably compared (revised **Figure S33**). Hence, we firstly evaluated the therapeutic performance of our RNAi prodrugs-packaged EVs through straightforward intratumoral injection, the extent of gene silencing was examined by qRT-PCR and western blot after their five days of post-injection. For the control group (saline administration, G1), the tumors observed highly expression level of twist and HIF-1 α mRNAs (revised **Figures S33A** and **S33B**), suggesting the extreme hypoxia environment of intact tumors. The direct twist/HIF-1 α siRNAs (G4) could modestly ameliorate the hypoxic extent of tumors as observed by the decreased expression of both twist and HIF-1 α genes in the administrated tumors. Interestingly, the tumors twist/HIF-1 α genes and proteins were both obviously weaker in the HCR-twist/HIF-1 α siRNAs-administrated (G5) mice than that of other groups (revised **Figure S33C**), suggesting the improved therapeutic performance of our simultaneous multi-antenna hypoxia-gene-silencing strategy. Notably, the gene silencing efficiency of the miRNA-activated HCR-promoted bis-RNAi approach is higher than that of the straightforward siRNAs protocol, implying a prolonged gene silencing procedure *via* the tumor-specific cascade activation of bis-RNAi. In our nanosystem, the RNAi prodrugs could recognize and hybridize with endogenous miR-21 to form dsDNA nanowires, resulting in the upregulation of downstream target genes, such as PDCD4, PTEN and Caspase-3. Indeed, we observed a significant increasement of PDCD4 and PTEN expressions with a concomitant increasement in apoptotic Caspase-3 expression in the specific HCR system (revised **Figure S33D**), confirming the intelligent immediate miR-21-diagnosis-guided successive activation of bis-RNAi nanoplateform *in vivo*.

After confirming the effective tumor hypoxia-alleviating by intratumoral injection into tumors, the miR-21-reponsive bis-RNAi nanoplateform was then explored in

MDA-MB-231 tumor-bearing mice *via* intravenous administration. As expected, the miR-21-assembled siRNAs observed significant downregulation of twist/HIF-1 α mRNAs (revised **Figures 4D** and **4E**, respectively) and proteins (revised **Figure 4F**). On the contrary, the miR-21-assembled scrambled-siRNAs (negative control) showed little effect on these mRNAs and proteins, demonstrating the effective gene silencing operation of our RNAi prodrugs. Interestingly, the gene silencing efficiency of miRNA-initiated HCR-promoted RNAi strategy is significantly higher than that of the straightforward siRNAs protocol, which is might attributed to the tumor-specific cascade activation of bis-siRNAs with prolonged gene silencing duration. Meanwhile, a remarkable up-regulation of PDCD4, PTEN and Caspase-3 mRNAs and proteins were observed in the HCR-involved gene regulation systems (revised **Figures 4G** and **S34**), demonstrating the specific miR-21-initiated RNAi platform. These results convincingly demonstrated the specific and effective activation of our RNAi prodrugs-packaged EVs *in vivo*. After confirming the feasibility of miR-21-responsive bis-RNAi *in vivo*, we attempted to evaluate their antitumor effect in the MDA-MB-231 mice model through intravenous injection. These gene silencing efficiency of RNAi prodrugs-packaged EVs and the accompanying discussions have been supplemented in the revised manuscript and supporting information.

Figure S33. The miR-21-activated gene silencing efficiency in tumors by intratumoral injection. (A,B) qRT-PCR analysis of twist and HIF-1 α mRNAs and (C) western blot analysis of twist, HIF-1 α protein expressions in MDA-MB-231 tumor-bearing mice treated with (G1) saline, (G2) EVs-sustained HCR-scrambled-siRNAs, (G3) EVs-sustained HCR-stimulated twist, (G4) EVs-sustained direct twist/HIF-1 α , or (G5) EVs-sustained HCR-stimulated twist/HIF-1 α on day five. Results are presented as means \pm standard deviation (SD) (n=5). **p<0.01, ****p<0.0001 (one-way ANOVA with a Tukey post hoc test for (A) and (B)). (D) Western blot analysis of the up-regulated PDCD4, PTEN and Caspase-3 proteins in MDA-MB-231 tumor-bearing mice intratumoral injected with miR-21-initiated HCR-amplified RNAi strategy.

Figure 4. *In vivo* biodistribution and gene silencing efficiency of RNAi prodrugs-packaged EVs in MDA-MB-231 tumor-bearing nude mice. (A) Construction of Trojan EVs-sustained programmable cascade activation of multi-antenna gene regulation for anti-cancer theranostic application. (B) *In vivo* distribution of free RNAi prodrugs, liposome-packaged RNAi prodrugs or EVs-packaged RNAi prodrugs in tumor-bearing mice under varied administration durations. (C) Colocalization of RNAi prodrugs (red) with CD31-labeled endothelial cells (green) in tumor sections after intravenous injection of free RNAi prodrugs, liposome-packaged RNAi prodrugs or EVs-packaged RNAi prodrugs for 12 h. Scale bars, 100 μ m. (D, E) qRT-PCR analysis and (F) western blot analysis of twist/HIF-1 α mRNAs and proteins in mice treated with (G1) saline, (G2) EVs-sustained

HCR-scrambled-siRNAs, (G3) EVs-sustained HCR-assembled twist, (G4) EVs-sustained direct twist/HIF-1 α , or (G5) EVs-sustained HCR-assembled twist/HIF-1 α on day five. Results are presented as means \pm standard deviation (SD) (n=5). *p<0.05, **p<0.01, ****p<0.0001 (one-way ANOVA with a Tukey post hoc test for (D) and (E)). (G) Western blot analysis of PDCD4, PTEN and Caspase-3 proteins in these differently treated mice as described above.

Figure S34. The miR-21-responsive bis-RNAi in tumors by intravenous administration. qRT-PCR analysis of the expression of (A) PDCD4, (B) PTEN, (C) Caspase-3 and (D) miR-21 in mice tumor after intravenous injected of (G1) saline, (G2) EVs-sustained HCR-scrambled-siRNAs, (G3) EVs-sustained HCR-stimulated twist, (G4) EVs-sustained direct twist/HIF-1 α , or (G5) EVs-sustained HCR-stimulated twist/HIF-1 α on day five. Results are presented as means \pm standard deviation (SD) (n=5). **p<0.01, ****p<0.0001 (one-way ANOVA with Tukey post hoc test for (A), (B) and (C)). ns, not statistically significant.

6) In figure 5, could the authors provide some evidences for the therapeutic protocol, the laser was too frequency.

Response: Thank you very much for your instructive comments. We are sorry for not supplementing an appropriate demonstration of our therapeutic procedure. Accordingly, the therapeutic protocol was provided by adopting the time-dependent tumor-specific accumulation (revised **Figures 4B** and newly supplemented **S31**) and therapeutic evaluation (**Figure S35**). The $\sim 300 \text{ mm}^3$ -sized tumor-bearing mice were respectively received intravenously injection of saline, RNAi prodrugs-packaged EVs, and RNAi prodrugs-packaged EVs with photoirradiation. The photoirradiation was performed after 12 h injection considering the optimal tumor-specific accumulation in tumor tissues and was repeated at 30 h to guarantee an efficient photothermal therapy (revised **Figure 4B**). Additionally, the anti-tumor effect of our RNAi system was evaluated by TUNEL (deoxynucleotidyl-transferase-mediated nick end labelling) analysis of tumor tissues (newly supplemented **Figure S35C**). The RNAi-treated tumor revealed a significantly increased TUNEL signal with prolonged therapeutic time, demonstrating the effective antitumor ability of our designed RNAi system. Under photoirradiation stimulation, our RNAi-involved gene silencing system showed an enhanced cell apoptosis, confirming the enhanced therapeutic performance of the combined bis-RNAi and PTT. These results indicate that the proposed therapeutic protocol is feasible for cancer therapy. Noteworthy, the mice were photoirradiated two times in our revised therapeutic protocol owing to the more developed tumor microenvironment with big tumor volume ($\sim 300 \text{ mm}^3$).

Figure S35. Validation of the therapeutic performance of the first therapeutic procedure *in vivo*. (A) Schematic illustration the Trojan EVs-sustained cascade activation of multi-antenna gene silencing and auxiliary PTT. (B) Therapeutic protocol of the administration of RNAi prodrugs in the subcutaneous MDA-MB-231 tumor model. (C) Time-dependent TUNEL analysis of the corresponding MDA-MB-231 tumor tissues.

7) In figure 5, the initial tumor size was too small for tumor inhibition. According to the anti-tumor results (Fig5B), contribution of siRNA to tumor inhibition was weak, photothermal therapy induced more obvious tumor eradication. Incomplete microenvironment of small tumors might be responsible. Could the author provide anti-tumor results in big tumors (about 300 mm³) which possess more developed

tumor microenvironment?

Response: Thank you very much for your insightful comments, which we appreciated very much. We agree that the initial 50 mm³-sized tumors were not big enough to accommodate a moderate tumor microenvironment. According to your suggestions, the larger 300 mm³-sized tumor was adapted for exploring the therapeutic performance of our designed RNAi system (revised **Figures 5** and **S36**). When the tumor volumes reached ~300 mm³, these mice were randomly divided into eight groups and treated, respectively, with saline (a), free RNAi prodrug with photoirradiation (b), EVs-sustained HCR-stimulated HIF-1 α (c), EVs-sustained HCR-scrambled-siRNAs with photoirradiation (d) liposome-sustained HCR-stimulated twist/HIF-1 α (e), EVs-sustained direct twist/HIF-1 α (f), EVs-sustained HCR-stimulated twist/HIF-1 α (g) and EVs-sustained HCR-stimulated twist/HIF-1 α with photoirradiation (h) with tail vein injection. The intravenous injections were carried out five times with an interval of 6 days according to the updated therapeutic protocol (revised **Figure 5A**). As compared with saline (group a) and photo-irradiated free RNAi prodrugs (group b), the bis-RNAi/liposome (group e) could halt moderately the tumor progression (revised **Figure 5B**). And the bis-RNAi/EVs platform (group g) exhibited a stronger inhibitory effect than the bis-RNAi/liposome group, which was attributed to the tumor-specific accumulation of EVs with homotypic targeting property. Also, the bis-RNAi/EVs group was observed with a more obvious tumor suppression than the HIF-1 α /EVs system (group c), indicating an enhanced therapeutic performance of our multi-antenna gene regulation system. Especially, the bis-RNAi/EVs strategy was observed with a more efficient inhibition of tumor progression as compared with the straightforward bis-siRNAs administration (group f), which is might attributed to the improved stability of the HCR-involved RNAi therapeutic agents. The photo-irradiated HCR-assembled scrambled siRNA (photoirradiation control, group d) could moderately suppress the growth of tumor, while the photo-irradiated bis-RNAi/EVs (group h) showed a significantly anticancer activity, demonstrating the enhanced therapeutic performance of the combined bis-RNAi and PTT. The therapeutic outcome was also confirmed by

photographs and weights of tumor harvested at day 27 of post-injection (revised **Figures 5C** and **5D**). The therapeutic performance of these different systems was furtherly validated by TUNEL staining (revised **Figure 5E**), which was in good agreement with these aforementioned observations. Meanwhile, no obvious inflammation or disorganization was observed in main organs of mice after our RNAi treatment (revised **Figure S36**), indicating a satisfied systemic biocompatibility of EVs. These as-suggested anticancer evaluation on bigger sized tumor tissues have been supplemented with the accompanying discussions in the revised manuscript and supporting information.

Figure 5. *In vivo* combined bis-gene therapy and PTT for MDA-MB-231 tumor eradication. (A) Schematic therapeutic protocol on tumor-bearing mice. (B) Time-dependent tumor growth profiles of mice that were respectively treated with (a) saline, (b) free RNAi prodrugs with photoirradiation, (c) EVs-sustained HCR-stimulated HIF-1 α , (d) EVs-sustained HCR-scrambled-siRNAs with photoirradiation, (e) liposome-sustained HCR-stimulated twist/HIF-1 α , (f) EVs-sustained direct twist/HIF-1 α , (g) EVs-sustained HCR-stimulated twist/HIF-1 α and (h) EVs-sustained HCR-stimulated twist/HIF-1 α with photoirradiation. All

photoirradiation was carried out with 808 nm laser (0.7 W/cm^2) for 5 min. Results are presented as means \pm standard deviation (SD) (n=6). * $p < 0.05$, *** $p < 0.001$, **** $p < 0.0001$ (two-way ANOVA with Bonferroni's multiple comparisons test). (C) Representative tumor images and (D) the average tumor weights of these differently treated mice. (D) TUNEL analysis of differently treated tumors.

Figure S36. Representative H&E-stained image of main organs after these mice were sacrificed at 27 day post intravenous injection with different formulas: (a) saline; (b) free RNAi prodrug with photoirradiation; (c) EVs-sustained HCR-stimulated HIF-1 α ; (d) EVs-sustained HCR-scrambled-siRNAs with photoirradiation; (e) liposome-sustained HCR-stimulated twist/HIF-1 α ; (f) EVs-sustained direct

twist/HIF-1 α ; (g) EVs-sustained HCR-stimulated twist/HIF-1 α and (h) EVs-sustained HCR-stimulated twist/HIF-1 α with photoirradiation. All photoirradiation was carried out with 808 nm laser (0.7 W/cm²) for 5 min.

8) In figure S18, group h was missed.

Response: Thank you very much for your instructive comments. We are sorry for our careless mistakes. The cell viability of the EVs-sustained HCR-stimulated twist/HIF-1 α with 808 nm photoirradiation (revised group i instead of group h) was provided in the revised **Figure S29**. The viability of bare EVs under 808 nm photoirradiation (data a) was above 85% over the entire concentration range, indicating the high biocompatibility of EVs under the moderate photoirradiation condition. As compared with the single-RNAi-involved gene silencing system at an even high dosage (33% cell viability lose, data c; 36% cell viability lose, data d), the combined bis-RNAi-involved system offered the enhanced therapeutic performance over the entire concentration range (56% cell viability lose, 150 nM, data g). Obviously, the miR-21-responsive RNAi prodrug leads to a more efficient anti-cancerous efficiency than the direct siRNA approach (data g vs f), suggesting a prolonged gene silencing effect. The photo-irradiation of EVs-sustained HCR-scrambled-siRNAs was observed with moderate cell antiproliferation (data e), indicating an effective PTT operation for disease treatment. Under a high dosage of RNAi prodrugs (150 nM) and photoirradiation stimulation, the proliferation of MDA-MB-231 cells was inhibited by 72% (data i), suggesting a cooperatively enhanced therapeutic performance of the combined gene silencing and PTT. Notably, the anti-tumor efficiency of bis-siRNA approach is higher in EVs than in liposome (data i vs h), which further validates the cell-specific homotypic targeting EVs from donor to recipient cells. These control cytotoxicity experiments have been supplemented with and appropriate descriptions in the revised manuscript and supporting information.

Figure S29. Cell viability of the MDA-MB-231 cells with different treatments for 48 h in hypoxic condition. (a): bare EVs with photoirradiation; (b): EVs-sustained HCR-scrambled-siRNAs; (c): EVs-sustained HCR-stimulated twist; (d): EVs-sustained HCR-stimulated HIF-1 α ; (e): EVs-sustained HCR-scrambled-siRNA with photoirradiation; (f): EVs-sustained direct twist/HIF-1 α ; (g): EVs-sustained HCR-stimulated twist/HIF-1 α ; (h): liposome-sustained HCR-stimulated twist/HIF-1 α with photoirradiation; (i): EVs-sustained HCR-stimulated twist/HIF-1 α with photoirradiation. All photoirradiation was carried out with 808 nm laser (0.7 W/cm²) for 5 min. Results are presented as means \pm standard deviation (SD) (n=5).

Reviewer #2:

The manuscript by Gong and co-workers employed the extracellular vesicles to load the RNAi prodrug for gene therapy. The RNAi drug (siRNA) can be generated from the RNAi prodrug through HCR amplification initiated by intracellular miRNA. After the combination with PTT, the drug delivery system demonstrated efficient tumor therapy. Generally, it is an interesting work. There are some major concerns that should be addressed.

1. As shown in Figure 1, the siRNA was divided into two parts (guide strand and passenger strand). After the endocytosis, the siRNA will be recovered from the RNA-DNA hybrid (RNAi prodrug) through HCR amplification. However, as we know, the intracellular RNase H is responsible for the digestion of the RNA strand in

the RNA-DNA hybrid, which is the principle of antisense therapy. The authors should investigate how much the siRNA strands are still retained in the intracellular environment. Will the PTT treatment damage the siRNA?

Response: Thank you very much for your insightful comments. The content of bis-siRNAs in MDA-MB-231 cells was evaluated by using qRT-PCR (newly supplemented **Figure S25**), while the gene silencing activity of photoirradiated RNAi prodrug-packaged EVs was explored by using western blot (newly supplemented **Figure S28**). Indeed, the RNase H could in principle digest RNA strand of RNA/DNA hybrid in intracellular environment, and this issue has been already brought into our serious consideration before the experimental design. After a careful examination of the related RNA interfering study, we find that this issue was not a big obstacle even this effect could not expelled. Here RNase H, as an endonuclease, requires an moderate high concentration of divalent cations (e.g., Mg^{2+}) for cleaving the RNA substrate [Cerritelli, S. M., Crouch, R. J. Ribonuclease H: the enzymes in eukaryotes. *FEBS J.* 276, 1494-1505 (2009)]. However, the concentration of intracellular available Mg^{2+} ions is rather low (~ 1 mM) to realize the efficient RNase H-mediated biocatalysis [Wolf, F. I., Cittadini, A. Chemistry and biochemistry of magnesium. *Mol. Aspects Med.* 24, 3-9 (2003); Tomita, A., Zhang, M., Jin, F., Zhuang, W., Takeda, H., Maruyama, T., Osawa, M., Hashimoto, K. I., Kawasaki, H., Ito, K., et al. ATP-dependent modulation of MgtE in Mg^{2+} homeostasis. *Nat. Commun.* 8, 148 (2017)], thus resulting in low degradation of exogenous RNA strand in cellular environment. Hence, our designed RNAi prodrugs could substantially avoid the RNase H-mediated degradation after endocytosis into cells. One should note that this undesired RNase H digestion could be fully expelled by introducing a specific chemical modification on the DNA scaffolds, e.g., 2-methoxyethoxy (MOE) nucleotides [Lima, W. F., Rose, J. B., Nichols, J. G., Wu, H., Migawa, M. T., Wyrzykiewicz, T. K., Siwkowski, A. M., Crooke, S.T. Human RNase H1 discriminates between subtle variations in the structure of the heteroduplex substrate. *Mol. Pharmacol.* 71, 83-91 (2007)], which we believe could contribute the most efficient RNAi platform. Yet this item is out of the range of the present study since we

are more focus on the proof of concept demonstration of the tumorous biomarker-activated RNAi platform. Meanwhile, the relative content of HIF-1 α and twist siRNAs in MDA-MB-231 cells were evaluated by using the Bulge-LoopTM miRNA qRT-PCR Starter Kit (RIBOBIO, China). As compared with the saline control, the content of both HIF-1 α and twist siRNAs increased through the miR-21-initiated HCR system (sample c of newly supplemented **Figure S25**). As an important control, the miR-21-inhibitor-pretreated cells had insignificant increasement of HIF-1 α and twist siRNAs (sample b of newly supplemented **Figure S25**), thus demonstrating the specific miR-21-initiated HCR-assembled bis-siRNAs.

The miR-21-responsive bis-RNAi platform was further confirmed by western blot (newly supplemented **Figure S28**). Under hypoxia condition, the expression of twist and HIF-1 α proteins was obviously down-regulated in our RNAi-treated MDA-MB-231 cells (sample d of newly supplemented **Figure S28**), while these protein expressions were kept nearly constant in a disabled HCR-involved RNAi system (merely antisense RNA/DNA hybrids-treated cells, sample b of newly supplemented **Figure S28**), indicating that the miR-21-initiated HCR-assembled bis-RNAs could efficiently cleave target mRNAs for gene silencing therapy. The western blot analysis also indicated that the activity of the two different siRNAs would not be affected by the PTT administration (sample c of the newly supplemented **Figure S28**), suggesting the generated photothermal stimulus had no effect on the therapeutic performance of the bis-siRNAs system. These newly supplemented qRT-PCR analysis of bis-siRNAs through miR-21-responsive HCR and gene silencing activity after photoirradiation experiments have been supplemented with appropriate discussions in the revised manuscript and supporting information.

Figure S25. qRT-PCR quantification of the relative expression of HIF-1 α and twist siRNAs after different treatments. a: saline; b; miR-21 inhibitor pre-treated EVs-loaded RNAi prodrugs; c, EVs-loaded RNAi prodrugs. Results are presented as means \pm standard deviation (SD) (n=3).

Figure S28. Western blot analysis of the varied expressions of twist and HIF-1 α proteins in MDA-MB-231 cells with different treatments. (a): saline; (b): EVs-encapsulated antisense RNA/DNA hybrids; (c): EVs-encapsulated RNAi prodrugs with photoirradiation (0.7 W/cm², 5 min); (d): EVs-encapsulated RNAi prodrugs. Under hypoxia condition, MDA-MB-231 cells were, respectively, treated with different reagents for 36 h.

2. As shown in Figure 4A, the intracellular trafficking pathway of the delivery system assisted by extracellular vesicles should be studied in detail. Where does the HCR occur, in or outside the EVs? If in the EVs, how does the miRNA enter? If outside the EVs, how to protect the RNA strands from RNase H?

Response: Thank you very much for your insightful comments. This is a very interesting question that needs to be extensively explored. In fact, the miR-21-responsive HCR was mainly occurred in cytoplasm, as evidenced by the varied content of endogenous miR-21 from EVs and MDA-MB-231 cells (qRT-PCR analysis, newly supplemented **Figure S10D**), the robust HCR without RNAi leakage in EVs (newly supplemented **Figure S14**), the specific endocytosis pathway assay (newly supplemented **Figure S17**), and the endosomal co-localization assay (newly supplemented **Figure S18**). Note that the EVs were produced by a specifically miR-21 inhibitor-pretreated MDA-MB-231 cells to fully expel the undesired miR-21-motivated HCR reaction in EVs before these functional EVs could internalize into the target MDA-MB-231 cells. Accordingly, a small molecule inhibitor was synthesized for specifically and efficiently inhibit the intracellular expression of miR-21 [K. Gumireddy, D.D. Young, X. Xiong, J.B. Hogenesch, Q. Huang, A. Deiters, Small-molecule inhibitors of microRNA miR-21 function. *Angew. Chem. Int. Ed.* **47**, 7482-7484 (2008)]. ¹H NMR spectra (newly supplemented **Figure S7**) and ¹³C NMR spectra (newly supplemented **Figure S8**) demonstrated the successful synthesis of the small molecule (an efficient miR-21 inhibitor). These MDA-MB-231 cells were pre-treated with the miR-21 inhibitor for 48 h, followed by washing thoroughly with PBS and then incubating in serum-free medium for another 48 h for releasing the miR-21-eliminated EVs. Only these miR-21-expelled EVs were then used as the versatile nanocarriers for our RNAi prodrugs.

Firstly, the content of miR-21 in EVs is substantially lower than the intact donor cells (without small molecule inhibitor pretreatment) according to the qRT-PCR analysis (newly supplemented **Figure S10D**), suggesting that the small molecule is an efficient inhibitor to inhibit the intracellular expression of miR-21. Therefore, these

miR-21-cleared EVs were used as versatile nanocarriers for all the subsequent experiments. Secondly, to demonstrate the specific miR-21-stimulated RNAi strategy, these RNAi prodrugs should be robust enough to prevent the undesired RNAi leakage or activation in live cells. Accordingly, the flow cytometry analysis was carried out to explore the stability of these RNAi prodrugs-packaged EVs. It demonstrated that our designed RNAi prodrugs are encoded with sufficient stability in EVs (newly supplemented **Figure S14**), suggesting that the HCR leakage does not occur in EVs.

Thirdly, the endocytosis inhibition experiments were carried out to clarify the uptake mechanism of our RNAi prodrugs-packaged EVs (newly supplemented **Figure S17**). The MDA-MB-231 cells were pretreated with six different endocytosis inhibitors, including sodium azide (NaN_3) inhibitor of ATP expression, chlorpromazine inhibitor of clathrin-mediated endocytosis, wortmannin inhibitor of micropinocytosis, nystatin inhibitor of lipid raft-caveolae endocytosis, and filipin inhibitor of caveolae endocytosis. These differently pretreated cells were then investigated for the subsequent internalization process of our functional EVs. The pretreatment of NaN_3 could abolish the uptake of EVs *via* the downregulation of ATP expression, indicating that the internalization of EVs was an energy-dependent endocytosis process. In addition, chlorpromazine could reduce the uptake of EVs by 34.5% while filipin and nystatin could reduce the uptake of EVs by 28.4% and 33.7%, respectively, indicating that both of clathrin-mediated endocytosis and caveolae-mediated endocytosis were involved in the EVs internalization process. In contrast, a strong internalization was still observed for wortmannin-pretreated cells, suggesting that micropinocytosis process was weakly involved in the internalization of the EVs. Collectively, these results demonstrated that the internalization of EVs was predominantly based on the clathrin-mediated endocytosis and caveolae-dependent endocytosis.

Finally, the internalization pathway of RNAi prodrugs-packaged EVs into MDA-MB-231 cells was extensively investigated by selectively staining lysosomes with LysoTracker Green. As shown in the revised supplemented **Figure S18A**, most of the RNAi prodrugs were escaped from lysosomes even with 1 h incubation with a Pearson correlation coefficient of 0.26. After a prolonged incubation time, the

fluorescence of RNAi prodrugs was enhanced by escaping from lysosomes. The endosomal escape of RNAi prodrugs might be attributed to the back fusion with endosomal membrane after endocytosis [Zheng, Z., Li, Z., Xu, C., Guo, B., Guo, P. Folate-displaying exosome mediated cytosolic delivery of siRNA avoiding endosome trapping. *J. Control. Release* 311–312, 43–49 (2019)]. These results further suggest that these RNAi prodrugs-packaged EVs could be internalized by cells and then was mainly localized in the cytosol compartment. Flow cytometry analysis further revealed an increased fluorescence intensity with elevated incubation time from 1 to 5 h (revised **Figures S18B** and **S18C**), indicating the high uptake efficiency of EVs in homologous MDA-MB-231 cells. Considering the main distribution of miRNA in cytoplasm compartment, the miR-21-initiated HCR was thus believe to occur in the cytoplasm compartment. One should note that the undesired by RNase H-mediated RNA degradation might also contribute to the fluorescence leakage. These newly supplemented qRT-PCR analysis of endogenous miR-21 from EVs and MDA-MB-231 cells, HCR robustness study in EVs, endocytosis demonstration, and endosomal escape experiments have been supplemented with appropriate discussions in the revised manuscript and supporting information.

Figure S6. Synthesis of the miR-21 inhibitor. 4-Phenylazobenzoic acid (30 mg, 0.133 mmol) was dissolved in DCM (1 mL), followed by the addition of 1-ethyl-3-(3'-dimethylaminopropyl) carbodiimide (42 mg, 0.22 mmol) and hydroxybenzotriazole (21 mg, 0.15 mmol). Propargylamine (15 mg, 0.27 mmol) was added, and the mixture was stirring for 12 h at room temperature. The reaction was quenched with water (5mL) and extracted with DCM (3 X 5mL). The organic layer was dried with sodium sulfate, concentrated and purified by silica gel chromatography (2:1 hexane/ethyl acetate) to yield an orange solid (27 mg, 0.10 mmol, 75%). ¹H NMR (400 MHz, CDCl₃) δ 7.99-7.93 (m, 6H), 7.54-7.52 (m, 3H), 6.35 (s, 1H), 4.30

(dd, $J = 4$ Hz, 2H), 2.32 (t, 1H); ^{13}C NMR (100 MHz, CDCl_3) δ 166.51, 154.61, 152.67, 135.52, 131.83, 129.34, 128.19, 123.27, 79.42, 72.30, 30.09.

Figure S7. ^1H NMR spectra of synthesized miR-21 inhibitor in CDCl_3 .

Figure S8. ^{13}C NMR spectra of miR-21 inhibitor in CDCl_3 .

Figure S10. Characterization of the isolated EVs derived from MDA-MB-231 cells. (A) Size distribution of EVs. (B) TEM characterization of EVs. (C) Western blot analysis of EVs and the corresponding cell lysates. (D) qRT-PCR analysis the content of miR-21 in EVs vs. its expression in MDA-MB-231 cells with miR-21-inhibitor pre-treatment (normalized to U6 RNA). Results are presented as means \pm standard deviation (SD) (n=5).

Figure S14. Stability of RNAi prodrugs in EVs. (A) Flow cytometric analysis and (B) the corresponding mean fluorescence intensity (MFI) of RNAi prodrug in EVs. Here, H_1 is functionalized at its 3'-end with BHQ1 while s_1 is modified at its 5'-end with Cy3. Results are presented as means \pm standard deviation (SD) (n=3).

Figure S17. The exploration of the uptake mechanism of RNAi prodrugs-packaged EVs. Confocal microscopy images (A) and quantification by analytical flow cytometry (B, C) of the internalized RNAi prodrugs in the presence of several inhibitors. Scale bar: 20 μ m. Results are presented as means \pm standard deviation (SD) (n=3).

Figure S18. Uptake of RNAi prodrugs by MDA-MB-231 cells over time. (A) Lysosomal co-localization in cells incubated RNAi prodrugs-packaged EVs for varied durations. (B) Flow cytometric analysis and (C) the corresponding mean fluorescence intensity (MFI) in cells incubated with RNAi prodrugs-packaged EVs for varied durations. Results are presented as means \pm standard deviation (SD) (n=3).

3. The size of EVs and the amount of siRNA in each EV are distributed in a certain range. The authors should provide a detailed protocol for the quantification of the siRNA in each injection.

Response: Thank you very much for your valuable comments. We are sorry for not providing the detailed calculation procedure of the loading efficiency of our RNAi prodrugs. The concentration of RNAi prodrugs in each injection was calculated by UV-vis spectroscopy. Here the electroporation of EVs were performed by using the Bio-Rad Gene Pulser Xcell™ Electroporation System. For optimization, 100 µg EVs and 20 µg RNAi prodrugs were mixed in 400 µL electroporation buffer (Bio-Rad) under the circumstance of 250 voltages, 350 µF capacitance in 0.4 cm cuvettes (Bio-Rad). After electroporation, the RNAi prodrugs-loaded EVs were pelleted *via* ultracentrifugation as described in the experimental section, and these unloaded RNAi prodrugs were collected from supernatant. For *in vivo* experiments, electroporation was performed in 400 µL electroporation buffer and pooled together for ultracentrifugation. After ultracentrifugation, EVs were fully suspended in cold sterile PBS, kept on ice and conservatively injected into the mice immediately (100 µL per mice). The concentration of free RNAi prodrugs and total input RNAi prodrugs were calculated by UV-vis spectroscopy (Varian Inc.). The RNAi prodrugs loading efficiency was calculated through the equation below:

$$\text{RNAi loading efficiency} = \frac{\text{Input prodrugs} - \text{free prodrugs}}{\text{Input prodrugs}}$$

3. In Figure 4B, it is confused to find that much lower fluorescence intensity was observed in the group of EVs in the whole body than the other two controls at 2 h. This low fluorescence may indicate that the nanocarriers have been almost metabolized during the short 2 hours. At the same time, the fluorescence intensity of EVs in the tumor region is also very low at 2 h.

Response: Thank you very much for your insightful comments. We admit that the functional RNAi prodrugs-packaged EVs could be partially metabolized in mice during the initial 2 h of intravenous injection. Although the fluorescence intensity of

EVs group is lower, the tumor-specific accumulation is more obvious than the two controls with prolonged incubation time. It is attributed to the longer retention of EVs in tumor tissue, as demonstrated by the efficient tumor-specific accumulation (revised **Figure 4B**, and newly supplemented **Figure S31**) and tumor-penetration capability (newly supplemented **Figure 4C**). Firstly, the ~ 300 mm³-sized tumor-bearing mice were received tail-vein injections of free ICG-RNAi prodrugs, ICG-RNAi prodrugs-loaded liposome and ICG-RNAi prodrugs-packaged EVs, respectively. Then these intravenously injected mice were examined by using an *in vivo* optical imaging system at various time points thereafter. As shown in revised **Figure 4B**, the intravenously injected free RNAi prodrugs were cleared out without obvious accumulation in tumor site. As compared with the liposome group, the EVs could progressively accumulate in tumor tissue and reached to a maximal accumulation after 12 h. In addition, the endothelial cells of tumor vessels was stained through CD31 biomarker to evaluate the penetration capability of EVs in deep tumor tissue. As compared with the free RNAi prodrugs and liposome-delivered RNAi prodrugs, the EVs-delivered RNAi prodrugs showed a much weaker co-localization with tumor vessels (newly supplemented **Figure 4C**), implying that the EVs carriers can effectively extravasate from tumor vessels for penetrating into tumor tissue. We also evaluated their time-dependent distribution in main organs and tumor tissue. After normalizing the radiation efficiency by organ weight, the EVs displayed a higher ratio of tumor to liver or kidney (newly supplemented **Figure S31**). These *ex vivo* results were consistent with the whole-body imaging observations, thus demonstrating the favorable tumor accumulation of EVs.

Generally, RES (liver, spleen and so on) is the most common pathway to remove drugs and xenobiotics with size above 6 nm from the blood stream. Increasing evidence has shown that nanoparticles can end up in the spleen if their sizes are comparable to the inter-endothelial cell slits of spleen (200-500 nm) or accumulate in the liver if their sizes are close to the vascular fenestration of liver (50–100 nm) [Blanco, E., Shen, H., Ferrari, M. Principles of nanoparticle design for overcoming biological barriers to drug delivery. *Nat. Biotechnol.* **33**, 941-951(2015); Du, B.,

Jiang, X., Das, A., Zhou, Q., Yu, M., Jin, R., Zheng, J. Glomerular barrier behaves as an atomically precise bandpass filter in a sub-nanometre regime. *Nat. Nanotech.* **12**, 1096-1102 (2017)]. Additionally, size-dependent renal clearance is readily observed for engineered nanoparticles with sizes below 6 nm by crossing a unique multiple-layer structure of glomeruli [Haraldsson, B., Nyström, J., Deen, W. M. Properties of the glomerular barrier and mechanisms of proteinuria. *Physiol. Rev.* **88**, 451-487 (2008); Du, B., Yu, M., Zheng, J. Transport and interactions of nanoparticles in the kidneys. *Nat. Rev. Mater.* **3**, 358-374 (2018)]. The long-term tumor tissue biodistribution and the accompanying discussions have been supplemented in the revised manuscript and supporting information.

Figure 4. *In vivo* biodistribution and gene silencing efficiency of RNAi

prodrugs-packaged EVs in MDA-MB-231 tumor-bearing nude mice. (A) Construction of Trojan EVs-sustained programmable cascade activation of multi-antenna gene regulation for anti-cancer theranostic application. (B) *In vivo* distribution of free RNAi prodrugs, liposome-packaged RNAi prodrugs or EVs-packaged RNAi prodrugs in tumor-bearing mice under varied administration durations. (C) Colocalization of RNAi prodrugs (red) with CD31-labeled endothelial cells (green) in tumor sections after intravenous injection of free RNAi prodrugs, liposome-packaged RNAi prodrugs or EVs-packaged RNAi prodrugs for 12 h. Scale bars, 100 μ m. (D, E) qRT-PCR analysis and (F) western blot analysis of twist/HIF-1 α mRNAs and proteins in mice treated with (G1) saline, (G2) EVs-sustained HCR-scrambled-siRNAs, (G3) EVs-sustained HCR-assembled twist, (G4) EVs-sustained direct twist/HIF-1 α , or (G5) EVs-sustained HCR-assembled twist/HIF-1 α on day five. Results are presented as means \pm standard deviation (SD) (n=5). *p<0.05, **p<0.01, ****p<0.0001 (one-way ANOVA with a Tukey post hoc test for (D) and (E)). (G) Western blot analysis of PDCD4, PTEN and Caspase-3 proteins in these differently treated mice as described above.

Figure S31. Tissue distribution analyses. (A) *Ex vivo* representative fluorescence images of major organs after administration of free RNAi prodrug, RNAi prodrug-loaded liposome or RNAi prodrug-packaged EVs to tumor-bearing mice at the indicated time points. H, heart; Li, Liver; S, spleen; Lu, lung; K, kidney; T, tumor. Quantitative analysis of the biodistribution in tumor to kidney (B) and tumor to liver ratio (C) at the indicated time points. Results are presented as means \pm standard deviation (SD) (n=3).

4. In Figure 4D, a noticeable higher MFI was found in the kidney for EVs compared with the control groups. The authors should discuss the possible side effects.

Response: Thank you very much for your insightful and valuable comments. Although the fluorescence signal is still observed in kidney after intravenous injection of RNAi prodrugs-packaged EVs for 24 h, we want to emphasize that the EVs were encoded with favorable biocompatibility and biosafety as demonstrated by hematology analysis (revised **Figure S16A**), blood biochemistry test (revised **Figure S16B**), and H&E examination of major organs (revised **Figure S36**). It is unclear that the EVs could bring any obvious side effect to the present system which we may need more exploration in the subsequent studies. Firstly, to evaluate the biosafety of EVs *in vivo*, the healthy BALB/C nude mice (female, 6-8 weeks) were intravenously injected with bare EVs at a dosage of 5 mg every week for a total of 5 times (n=3 mice). After 6 weeks, these mice were sacrificed and the blood were collected for hematology and blood biochemistry analysis. As anticipated, no significant variation was observed for liver and kidney functions as revealed by their corresponding unchanged indicators (revised **Figure S16**), suggesting the favorable biosafety of our EVs nanocarriers for bio-applications. Moreover, the major organs, including heart, liver, spleen, lung and kidney, of each treatment group were observed without obvious inflammation or disorganization after 27 days therapy according to the H&E examination (revised **Figure S36**). These results clearly demonstrated the long-term histo-compatibility of the EVs nanovesicles. These biocompatibility experiments have been supplemented

with appropriate discussions in the revised manuscript and supporting information.

Figure S16. Biosafety evaluation of the EVs nanovesicles *in vivo*. (A) Whole blood cell analysis of mice after 6 weeks post intravenous injection EVs. RBC, red blood cell; WBC, white blood cell; HGB, hemoglobin; HCT, hematocrit; MCH, mean corpuscular hemoglobin; MCV, mean corpuscular volume; MCHC, mean corpuscular hemoglobin concentration, PLT, platelet count. (B) Hepatic and renal functions analysis of the intravenously injected mice with EVs after 6 weeks. TP, total protein; ALB, albumin; ALT, alanine aminotransferase; AST, aspartate aminotransferase; BUN, blood urea nitrogen; CREA, creatinine; GLOB, globulin; TBIL, total bilirubin. Results are presented as means \pm standard deviation (SD) (n=3).

Figure S36. Representative H&E-stained image of main organs after these mice were sacrificed at 27 day post intravenous injection with different formulas: (a) saline; (b) free RNAi prodrugs with photoirradiation; (c) EVs-sustained HCR-stimulated HIF-1 α ; (d) EVs-sustained HCR-scrambled-siRNAs with photoirradiation; (e) liposome-sustained HCR-stimulated twist/HIF-1 α ; (f) EVs-sustained direct twist/HIF-1 α ; (g) EVs-sustained HCR-stimulated twist/HIF-1 α and (h) EVs-sustained HCR-stimulated twist/HIF-1 α with photoirradiation. All photoirradiation was carried out with 808 nm laser (0.7 W/cm²) for 5 min.

5. In Figure S18, group h is missing.

Response: Thank you very much for your instructive comments. We are sorry for our careless mistakes. The cell viability of the EVs-sustained HCR-stimulated twist/HIF-1 α with 808 nm photoirradiation (revised group i instead of group h) was provided in the revised **Figure S29**. The viability of bare EVs under 808 nm photoirradiation (data a) was above 85% over the entire concentration range, indicating the high biocompatibility of EVs under the moderate photoirradiation condition. As compared with the single-RNAi-involved gene silencing system at an even high dosage (33% cell viability lose, data c; 36% cell viability lose, data d), the combined bis-RNAi-involved system offered the enhanced therapeutic performance over the entire concentration range (56% cell viability lose, 150 nM, data g). Obviously, the miR-21-responsive RNAi prodrug leads to a more efficient anti-cancerous efficiency than the direct siRNA approach (data g vs f), suggesting a prolonged gene silencing effect. The photo-irradiation of EVs-sustained HCR-scrambled-siRNAs was observed with moderate cell antiproliferation (data e), indicating an effective PTT operation for disease treatment. Under a high dosage of RNAi prodrugs (150 nM) and photoirradiation stimulation, the proliferation of MDA-MB-231 cells was inhibited by 72% (data i), suggesting a cooperatively enhanced therapeutic performance of the combined gene silencing and PTT. Notably, the anti-tumor efficiency of bis-siRNA approach is higher in EVs than in liposome (data i vs h), which further validates the cell-specific homotypic targeting EVs from donor to recipient cells. These control cytotoxicity experiments have been supplemented with and appropriate descriptions in the revised manuscript and supporting information.

Figure S29. Cell viability of the MDA-MB-231 cells with different treatments for 48 h in hypoxic condition. (a): bare EVs with photoirradiation; (b): EVs-sustained HCR-scrambled-siRNAs; (c): EVs-sustained HCR-stimulated twist; (d): EVs-sustained HCR-stimulated HIF-1 α ; (e): EVs-sustained HCR-scrambled-siRNA with photoirradiation; (f): EVs-sustained direct twist/HIF-1 α ; (g): EVs-sustained HCR-stimulated twist/HIF-1 α ; (h): liposome-sustained HCR-stimulated twist/HIF-1 α with photoirradiation; (i): EVs-sustained HCR-stimulated twist/HIF-1 α with photoirradiation. All photoirradiation was carried out with 808 nm laser (0.7 W/cm²) for 5 min. Results are presented as means \pm standard deviation (SD) (n=5).

Reviewer #3:

1. Although the manuscript is novel and interesting, it is very difficult to read. English needs to be significantly improved, and the text needs to be rearranged so that the message is clear and concise.

Response: Thank you very much for your encouraging comments. We are sorry for not supplementing a clear and concise arrangement of our manuscript. Accordingly to your suggestion, we have gone through the manuscript and correct the ambiguous and tedious descriptions of our revised manuscript. To emphasize the advantage of our miRNA-stimulated RNAi theranostic system, the revised manuscript was supplemented with more experimental demonstrations with appropriate descriptions. In addition, to guarantee a more developed hypoxia tumor microenvironment, the

larger 300 mm³-sized tumor-bearing mice were introduced to execute the *in vivo* theranostic experiments.

2. A discussion of the results obtained in the context of previous literature is missing.

Response: Thank you very much for your valuable comments. We are sorry for not providing the comprehensive comparison of our system with previous literatures. Accordingly, these more relevant context of previous studies were introduced into the newly supplemented Discussion Section of our manuscript.

3. The authors should provide more background on the use of HCR amplification strategies towards therapeutics. Specifically, the authors should mention whether this kind of strategy was previously tested using other nanocarriers, advantages/disadvantages of their system compared to others (if applicable), whether this was tested in other disease contexts or not.

Response: Thank you very much for your insightful comments, which we appreciated very much. The fact is HCR has been initially explored for amplified intracellular imaging with the assistance of liposomes, upconversion nanoparticles or AuNPs nanocarriers [Cheglakov, Z., Cronin, T. M., He, C., Weizmann, Y. *J. Am. Chem. Soc.* **137**, 6116-6119 (2015); Koos, B., Cane, G., Grannas, K., Lof, L., Arngarden, L., Heldin, J., Claesson, C., Klaesson, A., Hirvonen, M. K., de Oliveira, F. M. S., Talibov, V. O., Pham, N. T., Auer, M., Danielson, H., Haybaeck, J., Kamali-Moghaddam, M., Soderberg, O. Proximity-Dependent Initiation of Hybridization Chain Reaction. *Nat. Commun.* **6**, 7294, (2015); Chu, H., Zhao, J., Mi, Y., Zhao, Y., Li, L. Near-Infrared Light-Initiated Hybridization Chain Reaction for Spatially and Temporally Resolved Signal Amplification. *Angew. Chem., Int. Ed.* **58**, 14877-14881 (2019); Wu, Z., Liu, G. Q., Yang, X. L., Jiang, J. H. Electrostatic Nucleic Acid Nanoassembly Enables Hybridization Chain Reaction in Living Cells for Ultrasensitive mRNA Imaging. *J. Am. Chem. Soc.* **137**, 6829-6836 (2015)], while its therapeutic utilization is still in its infancy.

By far, only the aptamer-tethered and pH-responsive HCR nanostructures were

utilized for intracellular imaging and the as-guided drug delivery [Zhu, G. Z., Zheng, J., Song, E. Q., Donovan, M., Zhang, K. J., Liu, C., Tan, W. H. Self-Assembled, Aptamer-Tethered DNA Nanotrains for Targeted Transport of Molecular Drugs in Cancer Theranostics. *Proc. Natl. Acad. Sci. U.S.A.* **110**, 7998-8003 (2013); Ma, W., Chen, B., Zou, S., Jia, R., Cheng, H., Huang, J., Wang, H., He, X., Wang, K. I-motif-based in situ bipedal hybridization chain reaction for specific activatable imaging and enhanced delivery of antisense oligonucleotides. *Anal. Chem.* **91**, 12538-12545 (2019)]. A triggered gene silencing system was recently realized on the long linear DNA scaffold [Ren, K., Zhang, Y., Zhang, X., Liu, Y., Yang, M., Ju, H. In situ siRNA assembly in living cells for gene therapy with microRNA triggered cascade reactions templated by nucleic acids. *ACS Nano* **12**, 10797-10806 (2018)], yet was delivered by a significant amount of synthetic polycationic nanocarriers that may cause undesirable immunogenicity or cytotoxicity. While the naturally secreted EVs have recently emerged as attractive nanocarriers due to their high biocompatibility and stability during systemic administration. Especially, the EVs displayed the cell-specific recognition and uptake capabilities by virtue of their intrinsic membrane proteins [Specificities of secretion and uptake of exosomes and other extracellular vesicles for cell-to-cell communication. *Nat. Cell Biol.* **21**, 9-17 (2019); The biology, function, and biomedical applications of exosomes. *Science* **367**, eaau6977 (2020)]. By using EVs as Trojan horses, we hijacked this robust nanovesicles to protect and deliver RNAi cargo into a specific cancer cells, where the miRNA-initiated HCR-promoted RNAi therapy was realized for achieving a satisfactory anti-tumor performance in subcutaneous and metastatic tumor models. The comparison of our EVs-encapsulating miRNA-responsive HCR-promoted RNAi theranostic system and the other theranostic platforms was clarified in the revised manuscript.

2. EVs need to be further characterized. According to the MISEV 2018 guidelines, it is recommended to detect in EVs – 1) the presence of specific proteins (at least 3, including transmembrane/ lipid-bound proteins and cytosolic markers) and 2) absence or significant reduction of negative markers (e.g. calnexin).

Response: Thank you very much for your valuable comments. According to your suggestion, the EVs-specific membrane proteins were characterized by western blot analysis and the corresponding results were presented in the newly supplemented **Figure S10C**. The EVs showed an enrichment of common EV-markers, including TSG101, Flotilin 2 and tetraspanins CD63, along a decrease of organelle-marker (Calnexin). These as-suggested EVs characterization was supplemented with appropriate descriptions in the revised manuscript and supporting information. We hope that the present revision is now acceptable.

Figure S10. Characterization of the isolated EVs derived from MDA-MB-231 cells. (A) Size distribution of EVs. (B) TEM characterization of EVs. (C) Western blot analysis of EVs and the corresponding cell lysates. (D) qRT-PCR analysis the content of miR-21 in EVs vs. its expression in MDA-MB-231 cells with miR-21-inhibitor pre-treatment (normalized to U6 RNA). Results are presented as means \pm standard deviation (SD) (n=5).

3. Electroporation can induce precipitation of siRNAs, and thus of RNA/DNA hybrids as well (please check this reference <https://pubmed.ncbi.nlm.nih.gov/23994516/>). This needs to be taken into consideration throughout the manuscript. An additional control for electroporation, repeating the protocol with RNA/DNA hybrids in the absence of EVs, should also be performed, so that the extent of RNA/DNA aggregate formation can be evaluated.

Response: Thank you very much for your valuable comments. We agree that the possible precipitation of RNA/DNA hybrids need to be evaluated for guaranteeing a more efficient RNAi therapy. Accordingly, the electroporated EVs were ultracentrifuged, which were then evaluated by DLS and an indirect qRT-PCR quantification with the assistance of RNase H as described in the Experimental Section. It showed that only 1% of RNA/DNA precipitate was observed in the electroporated EVs. Firstly, the electroporation-induced RNAi precipitation was elevated by an indirect morphological characterization *via* DLS. Therefore, in the absence of EVs, the RNA/DNA hybrids were electroporated and ultracentrifuged to collect the precipitation pellet. Not surprisingly, partial RNAi prodrugs aggregate when subjecting to the electroporation procedure with a broad size distribution (newly supplemented **Figure S12A**). Secondly, the HIF-1 α RNA/DNA hybrids were introduced as a model system to further quantify the RNA/DNA precipitation degree after EVs electroporation. To estimate the amount of RNA/DNA aggregates outside of EVs, the EVs pellet were isolated to collect the total RNA. Meanwhile, the same amount of EVs pellet was treated with RNase H to degrade the precipitated RNA/DNA outside of EVs to collect the EVs-encapsulated RNA/DNA hybrids. Then the reverse transcription of these differently collected DNA/RNA hybrids was performed by using the Bulge-LoopTM miRNA qRT-PCR Starter Kit (RIBOBIO, China). It showed that only 1% of RNA/DNA precipitate was observed in the electroporated EVs. Hence, the electroporation did not cause obvious aggregation on RNAi prodrugs. In addition, the electroporation did not significantly change the size distribution of EVs (revised **Figure 2B**) by preserving their native morphological structures (revised **Figure 2C**, and newly supplemented **Figure S12B**), further

demonstrating that the electroporation procedure induce no obvious aggregation of RNAi prodrugs. The aggregation evaluation of these electroporated RNA/DNA with and without EVs have been supplemented with appropriate discussions in the revised manuscript and supporting information. We hope that the present revision is now acceptable.

Figure S12. Characterization of RNAi prodrugs. (A) Size distribution of RNAi prodrugs aggregates after electroporation as measured by DLS. (B) Representative TEM image of RNAi prodrugs-packaged EVs.

Figure 2. Characterization of the RNAi prodrugs-packaged EVs. (A) Schematic illustration of the RNAi prodrugs-encapsulated EVs. (B) Hydrodynamic diameter, (C) TEM image, and (D) flow cytometric analysis of RNAi prodrugs-loaded EVs. (E) Diagnostic EVs-sustained HCR-amplified FRET analysis of intracellular miR-21 (in the form of F_A/F_D) and corresponding FRET signal distributions in different cell lines: (a) MDA-MB-231, (b) MCF-7, (c) HeLa and (d) MRC-5 cells. All scale bars correspond to 20 μm .

4. Nude mice lack a thymus, and are unable to produce T cells. Therefore, to be informative, the immunogenicity of the developed system needs to be tested in immunocompetent mice. Furthermore, the levels of cytokines should be evaluated a few hours after injection, as elevations in serum cytokines are transient and typically resolve within 12-24 hours after initial treatment.

Response: Thank you very much for your valuable comments. We are sorry for our careless mistake on this issue. We agree that the immunogenicity of our system should be tested in immunocompetent mice. According to your suggestion, the

systemic immune response was evaluated on healthy C57BL/6 mice after the intravenous injection of EVs with different durations or time-intervals (revised **Figures S15D** and **S15E**). As expected, the EVs treatment had no significant impact on the cytokine expression as evidenced by the constant content of serum cytokines (IL-6 and TNF- α). These results suggest that our EVs nanovesicles were indeed immunologically inert, which was good accordance with previous observations [Yang, Z., Shi, J., Xie, J., Wang, Y., Sun, J., Liu, T., et al. Large-scale generation of functional mRNA-encapsulating exosomes via cellular nanoporation. *Nat. Biomed. Eng.* **4**, 69-83 (2020); Liang, Q., Bie, N., Yong, T., Tang, K., Shi, X., Wei, Z., et al. The softness of tumour-cell-derived microparticles regulates their drug-delivery efficiency. *Nat. Biomed. Eng.* **3**, 729-740 (2019)]. These revised immunogenicity experiments and the corresponding results have been supplemented with appropriate discussions in the revised manuscript and supporting information. We hope that the present revision is now acceptable.

Figure S15. The biocompatibility evaluation of EVs. (A) Cell viability assay of MDA-MB-231, HeLa, MRC-5 and BMDCs cells after their incubation with bare EVs (225 $\mu\text{g/mL}$) for varied durations. (B) Hemolytic analysis of RNAi prodrugs-packaged EVs at varied concentrations. PBS and water were used as negative and positive control, respectively. (C) SEM image of RBC after its treatment with RNAi prodrugs-loaded EVs (500 $\mu\text{g/mL}$) for 6 h. The respective expression level of IL-6 (D) and TNF- α (E) as measured by ELISA kit after the C57BL/6 mice was

administrated with bare EVs for different durations. The cell viability and hemolysis assay results are presented as means \pm standard deviation (SD) (n=5) while the serum cytokines results are presented as means \pm standard deviation (SD) (n=3).

5. A detailed description of the liposomes used in their studies, and how these were conjugated with HCR hairpins, is lacking. Furthermore, if using lipofectamine 3000, at least for in vivo studies, the authors cannot argue that EVs are more efficient, as there are several improved liposomal preparations that perform much better in vivo (e.g. stable nucleic acid lipid particles). This should be added to discussion.

Response: Thank you very much for your comments. We are sorry for not providing the detailed transfection procedure of RNAi prodrugs *via* lipofectamine 3000 (ThermoFisher). Accordingly, the transfection of RNAi prodrugs was supplemented with appropriate descriptions in the Experimental Section of our revised manuscript. In brief, the RNAi prodrugs was prepared in Opti-MEM (30 μ L), and was then mixed with lipofectamine 3000 (3 μ L) dispersed in Opti-MEM (30 μ L) for 5 min. Subsequently, the prepared Opti-MEM transfection mixture was introduced into the plated cells for 3 h at 37°C. For the *in vivo* assay, RNAi prodrugs (0.5 mg) in Opti-MEM (60 μ L) was mixed with lipofectamine 3000 (30 μ L) in Opti-MEM (60 μ L) for 5 min. Then the above Opti-MEM transfection mixture was intravenously injected into mice immediately.

Indeed, liposomes are one of the most widely investigated nanocarriers for siRNA delivery. The diverse surface properties of liposomal nanocarriers, including varied surface charge, programmable ligand modification, and robust PEGylation can significantly improve the efficiency of gene silencing [Sato, Y., Hatakeyama, H., Sakurai, Y., Hyodo, M., Akita, H., Harashima, H. A pH-sensitive cationic lipid facilitates the delivery of liposomal siRNA and gene silencing activity in vitro and in vivo. *J. Control. Release*, **163**, 267-276 (2012); Xia, Y., Tian, J., Chen, X. Effect of surface properties on liposomal siRNA delivery. *Biomaterials* **79**, 56-68 (2016); Müller, K., Kessel, E., Klein, P. M., Höhn, M., Wagner, E. Post-PEGylation of siRNA lipo-oligoamino amide polyplexes using tetra-glutamylated folic acid as ligand

for receptor-targeted delivery. *Mol. Pharmaceutics* **13**, 2332-2345 (2016)]. In our present study, the lipid nanoparticles were prepared by using a commercial transfection reagent, lipofectamine 3000, without any additional chemical modification according to the manufacturer's instructions. This is might be the reason that our EVs exhibited a significantly improved cells-specific and tumor-targeting properties (as result of the homologous active targeting) than the conventional commercial liposomes. According to your suggestion, the obscure comparison on liposomes was re-phrased in our revised manuscript. We hope that the present version is now acceptable.

6. The authors mention throughout the manuscript the specificity, absence of immunogenicity and lack of off-target effects of the developed system. These claims are not fully proven and therefore these should be suggestions rather than certainties.

Response: Thank you very much for your instructive comments. We admit that some of our statement was too assertive to make the conclusion on specificity, immunogenicity and off-target effects. The demonstration of biocompatibility and biosafety requires a more comprehensive evaluation. Here the immunogenicity analysis (revised **Figures S15D** and **S15E**), hematology test (revised **Figure S16A**), blood biochemistry analysis (revised **Figure S16B**), and H&E examination of major organs (revised **Figure S36**) was introduced to demonstrate the favorable biocompatibility and biosafety of our EVs. Meanwhile, the homotypic targeting ability of EVs was also investigated by CLSM and flow cytometry analysis (revised **Figure S19**), suggesting that the preferential tropism of EVs for their donor cells. However, it is still unclear that the EVs could bring any obvious side effect to the present system which requires more extensive explorations in our subsequent studies. According to your suggestion, these arbitrary descriptions have been rephrased in our revised manuscript and supporting information. We hope that the present revision is now acceptable.

7. The authors show that MDA-MB-231 cells exhibit higher uptake for

MDA-MB-231 EVs than other cells e.g. MCF-7. In contrast, it was previously reported higher transfer of EV cargo from MDA-MB-231 EVs to different target cells, including MCF-7

(<https://www.sciencedirect.com/science/article/pii/S0092867415004985?via=ihub>).

This is an important point for discussion?

Response: Thank you very much for your insightful comments. The preferential tropism of EVs for their donor cells was observed and demonstrated by laser confocal scanning microscopy (CLSM) imaging and flow cytometry analysis (revised **Figure S19**). Here, more cells was introduced to determine whether the EVs have preferential tropism for their progenitor cells by comparing their fluorescence intensity in different cell types, including MDA-MB-231, MCF-7, HeLa and MRC-5 cells. These Cy5- RNAi prodrugs-packaged EVs were readily taken up by both tumor and non-tumor cells after 3 h of incubation (revised **Figure S19**), suggesting that EVs are promising carriers for nucleic acid therapeutic agents. However, the homologous MDA-MB-231 cells exhibit the highest uptake efficiency than the other non-homologous cells, which are in good accordance with the previous works [Cheng, G., Li, W., Ha, L., Han, X., Hao, S., Wan, Y., Wang, Z., Dong, F., Zou, X., Mao, Y., Zheng, S. Y. Self-Assembly of Extracellular Vesicle-like Metal-Organic Framework Nanoparticles for Protection and Intracellular Delivery of Biofunctional Proteins. *J. Am. Chem. Soc.* **140**, 7282-7291 (2018); Sancho-Albero, M., Rubio-Ruiz, B., Pe´rezLo´pez, A. M., Sebastia´n, V., Martı´n-Duque, P., Arruebo, M., Santamarı´a, J., Unciti-´Broceta, A. Cancer-derived exosomes loaded with ultrathin palladium nanosheets for targeted bioorthogonal catalysis. *Nat. Catal.* **2**, 864-872 (2019)]. A growing number of studies have suggest that EVs can recognize the specific types of cells through their surface receptors, and these tumor-cell-derived EVs can thus preferentially target certain distant cells and tissues [Vader, P., Mol, E. A., Pasterkamp, G., Schiffelers, R. M. Extracellular vesicles for drug delivery. *Adv. Drug Deliv. Rev.* **106**, 148-156 (2016); Sancho-Albero, M., Rubio-Ruiz, B., Pe´rezLo´pez, A. M., Sebastia´n, V., Martı´n-Duque, P., Arruebo, M., Santamarı´a, J., Unciti-´Broceta, A. Cancer-derived exosomes loaded with ultrathin palladium

nanosheets for targeted bioorthogonal catalysis. *Nat. Catal.* **2**, 864-872 (2019)]. Meanwhile, Zomer and coworkers have showed that MDA-MB-231-produced EVs are uptaken by less malignant tumor cells [Zomer, A., Maynard, C., Verweij, F.J., Kamermans, A., Schafer, R., Beerling, E., Schiffelers, R.M., de Wit, E., Berenguer, J., Ellenbroek, S.I., et al. In vivo imaging reveals extracellular vesicle-mediated phenocopying of metastatic behavior. *Cell* **161**, 1046-1057 (2015)]. The exact mechanism is not clear why the MCF-7 cells showed a higher cellular uptake of native EVs than the corresponding donor cells. We assume that the preserved membrane bioactivity of EVs may be responsible for the higher cellular uptake. Indeed, further flow cytometry analysis showed that the MDA-MB-231 cells could uptake a slightly higher amount of native EVs than the electroporated EVs (newly supplemented **Figure S20**). Thus the preservation of EVs membrane bioactivity is crucial for their extensive applications. This homotypic targeting performance of EVs has been supplemented with appropriate discussions in the revised manuscript and supporting information. We hope that the present explanation is now acceptable.

Figure S19. Homotypic targeting evaluation of EVs. (A) CLSM image of MDA-MB-231, HeLa, MCF-7 and MRC-5 cells that were incubated with RNAi prodrugs-loaded EVs derived from MDA-MB-231 cells for 3 h. The cells nuclei were

stained with Hoechst 33342 (blue). Scale bar: 20 μm . (B) Flow cytometry analysis of the RNAi prodrugs in different cells. (C) Quantitative analysis of the intracellular fluorescence intensity of RNAi prodrugs in different cells. (D) The relative uptake efficiency of different cells as compared to the MDA-MB-231 cells. Results are presented as means \pm standard deviation (SD) (n=3).

Figure S20. Demonstration of the EVs uptake by MDA-MB-231 cells. (A) Flow cytometry analysis and (B) the corresponding mean fluorescence intensity (MFI) of MDA-MB-231 cells that were incubated with RNAi prodrugs-electroporated DiI-labeled EVs (a) or DiI-labeled native EVs (b) for 3 h. Results are presented as means \pm standard deviation (SD) (n=3).

8. In materials and methods the authors mention that all data were analysed using Student's t-test, but in most, if not all cases, there are more than two different groups of treatment. The authors should rectify the statistical analysis throughout the manuscript.

Response: Thank you very much for your comments. We are sorry for our careless mistake. Here the comparison between two groups was investigated by using unpaired two-tailed Student's t-test while the one-way ANOVA or two-way ANOVA was used for comparison of more than two groups. Values with $p < 0.05$ were considered as statistically significant. All of these data analyses were conducted using Graphpad 8.0 software without excluding any samples. Accordingly, the statistics analysis was

supplemented with appropriate descriptions in the figure legend and Experimental Section of our revised manuscript as well as supporting information.

9. The authors use different names for the RNA/DNA hybrids throughout the manuscript (e.g. RNAi prodrugs, HCR hairpins, siRNA/DNA hybrids, siRNA/DNA prodrugs, ...). Can the authors adopt one term while presenting the results and use it throughout the rest of the manuscript, so that the reader can follow the paper more easily?

Response: Thank you very much for your insightful comments. We are sorry for not make a consistent nomination of the RNA/DNA hybrids. According to your suggestion, the RNA/DNA hybrids was specified as RNAi prodrugs in the revised manuscript and supporting information. We hope that the present version is now acceptable.

Minor points:

1. Can the authors add a brief description of the scheme represented in Figure 1 to the legend?

Response: Thank you very much for your comments. According to your suggestions, a brief description of the scheme in **Figure 1** was presented in our revised manuscript.

Figure 1. The stimulus-responsive RNAi theranostic system. The reconstitution of RNAi system from active siRNA (consisting of sense RNA and antisense RNA) into inactive RNA/DNA hybrids for assembling the stimulated RNAi through siRNA re-association. The rationally designed RNA/DNA hybrids pave the way for stimulus-responsive non-amplified RNAi (Part I) and miRNA-stimulated HCR-amplified RNAi (Part II) systems.

2. Why are the isolated EVs so big (> 200 nm), considering that bigger EVs were eliminated by UC at 10,000 g? Any explanation?

Response: Thank you very much for your insightful comments. After a careful examination of the samples, we found that the large size of dynamic light scattering (DLS) measurement was might attributed to the incomplete dispersion in PBS after ultracentrifugation. The obtained EVs were collected by gently pipetting the PBS medium against the pellet at the bottom of the ultratube. The EVs characterization was thus updated with more in-depth exploration on their stabilities in the revised **Figures S10** and **S11**. DLS characterization revealed that the native EVs had an average diameter of ~132 nm in our revised **Figure S10A**. Note that the EVs size of transmission electron microscopy (TEM) image might not always represent the actual

size distribution in the full population. As shown in our revised **Figure S10B**, a typical cup-like morphology of the isolated EVs was observed by TEM. No obvious aggregation and size distribution change was observed for EVs with 3 cycles of freeze-thaw operations as determined by DLS and TEM (revised **Figure S11**), indicating the sufficient biostability of the EVs nanocarrier. After the RNAi prodrugs were transfected into EVs by electroporation, the loaded EVs displayed a slight increasement in average size (146 nm) yet without noticeable morphologic changes (revised **Figures 2B** and **2C**). These updated characterizations of EVs were clarified in the revised manuscript and supporting information. We hope that the present revision is now acceptable.

Figure S10. Characterization of the isolated EVs derived from MDA-MB-231 cells. (A) Size distribution of EVs. (B) TEM characterization of EVs. (C) Western blot analysis of EVs and the corresponding cell lysates. (D) qRT-PCR analysis the content of miR-21 in EVs vs. its expression in MDA-MB-231 cells with miR-21-inhibitor pre-treatment (normalized to U6 RNA). Results are presented as

means \pm standard deviation (SD) (n=5).

Figure S11. Characterization of EVs that were treated with multiple freeze-thaw cycles. Size distribution of EVs after one freeze-thaw cycle (A), two freeze-thaw cycles (B) and three freeze-thaw cycles (C). (D) Representative TEM image of EVs after three freeze-thaw cycles.

Figure 2. Characterization of the RNAi prodrug-packaged EVs. (A) Schematic illustration of the RNAi prodrug-encapsulated EVs. (B) Hydrodynamic diameter, (C) TEM image, and (D) flow cytometric analysis of RNAi prodrugs-loaded EVs. (E) Diagnostic EVs-sustained HCR-amplified FRET analysis of intracellular miR-21 (in the form of F_A/F_D) and corresponding FRET signal distributions in different cell lines: (a) MDA-MB-231, (b) MCF-7, (c) HeLa and (d) MRC-5 cells. All scale bars correspond to 20 μm .

3. The authors suggest that the RNAi prodrug is encapsulated in EVs (Page 10, Fig. S8) due to a tolerance for nuclease degradation. But exposure to DNase/RNases should be tested to prove their point. The explanation for the stability in 10% FBS at 37°C might be due to the higher thermodynamic stability of DNA/RNA hybrids, as the authors previously mentioned, and it does not mean that the RNAi prodrug is encapsulated in EVs.

Response: Thank you very much for your valuable comments. We agree that the DNase/RNases should be introduced to demonstrate the degradation tolerance of the

EVs-encapsulated RNAi prodrugs. The EVs could prevent the extracellular nuclease-mediate degradation of the as-incorporated RNAi prodrugs, as evidenced by the DNase I/RNases-mediated degradation experiments (newly supplemented **Figure S13**). Here, **H₁** was functionalized at its 3'-end with a BHQ-1 while **s₁** was modified at their 5'-end with Cy3. To test the stability of RNAi prodrugs within EVs, the RNAi prodrugs-electroporated EVs was incubated with DNase/RNases-containing Opti-MEM at 37 °C for 24 h. Unsurprisingly, no obvious fluorescence increasement was observed in the electroporated EVs-encapsulated RNAi prodrugs (carve a, newly supplemented **Figure S13**). However, a gradually increasement of fluorescence signal was observed in the non-electroporated mixture of RNAi prodrugs and EVs (carve b, newly supplemented **Figure S13**). These results demonstrate that the electroporation operation appears to maintain the integrity and functionality of EVs and the EVs could protect the encapsulated RNAi prodrugs from undesired digestion in complex biological environment. The stability of RNAi prodrugs within EVs have been supplemented with the accompanying discussions in the revised manuscript and supporting information.

Figure S13. The RNAi prodrugs degradation profile in opti-MEM containing 10 U RNase H and 10 U DNase at 37 °C: (a) RNAi prodrug-packaged EVs; (b) RNAi prodrug with EVs without electroporation. Here, **H₁** is functionalized at its 3'-end with BHQ1 while **s₁** is modified at its 5'-end with Cy3. Results are presented as means ± standard deviation (SD) (n=3).

4. The authors refer in page 10 that the EVs are safe and do not transfer large amounts of oncogenic miRNA. It would be important to evaluate the miR21 levels in EVs relative to the cells (MDA-MB-231).

Response: Thank you very much for your insightful comments. We agree that the expression of miR-21 should be evaluated in both of EVs and donor cells to guarantee an efficient and stimulus-responsive RNAi system. Accordingly, the endogenous miR-21 content of EVs and MDA-MB-231 cells was evaluated *via* qRT-PCR analysis (newly supplemented **Figure S10D**). Note that the EVs were produced by a specific

miR-21 inhibitor-pretreated MDA-MB-231 cells to fully eliminate the undesired miR-21-motivated HCR reaction in EVs before their internalization into target MDA-MB-231 cells. Here a small molecule was synthesized to specifically and efficiently inhibit the expression of miR-21 [K. Gumireddy, D.D. Young, X. Xiong, J.B. Hogenesch, Q. Huang, A. Deiters, Small-molecule inhibitors of microRNA miR-21 function. *Angew. Chem. Int. Ed.* **47**, 7482-7484 (2008)]. ¹H NMR spectra (newly supplemented **Figure S7**) and ¹³C NMR spectra (newly supplemented **Figure S8**) demonstrated the successful synthesis of small molecule (an efficient miR-21 inhibitor). These MDA-MB-231 cells were pre-treated with this small inhibitor for 48 h, followed by washing thoroughly with PBS and then incubating in serum-free medium for another 48 h to generate the miR-21-eliminated EVs. Only these miR-21-expelled EVs were then used as the versatile nanocarriers for our RNAi prodrugs in all the subsequent experiments. The content of miR-21 in EVs is substantially lower than the intact donor cells (without small molecule inhibitor pretreatment) according to the qRT-PCR analysis (newly supplemented **Figure S10D**). This newly supplemented qRT-PCR analysis of endogenous miR-21 from EVs and MDA-MB-231 cells has been supplemented with appropriate discussions in the revised manuscript and supporting information.

Figure S6. Synthesis of the miR-21 inhibitor. 4-Phenylazobenzoic acid (30 mg, 0.133 mmol) was dissolved in DCM (1 mL), followed by the addition of 1-ethyl-3-(3'-dimethylaminopropyl) carbodiimide (42 mg, 0.22 mmol) and hydroxybenzotriazole (21 mg, 0.15 mmol). Propargylamine (15 mg, 0.27 mmol) was added, and the mixture was stirring for 12 h at room temperature. The reaction was quenched with water (5mL) and extracted with DCM (3 X 5mL). The organic layer was dried with sodium sulfate, concentrated and purified by silica gel chromatography (2:1 hexane/ethyl acetate) to yield an orange solid (27 mg, 0.10 mmol, 75%). ¹H

NMR (400 MHz, CDCl₃) δ 7.99-7.93 (m, 6H), 7.54-7.52 (m, 3H), 6.35 (s, 1H), 4.30 (dd, J = 4 Hz, 2H), 2.32 (t, 1H); ¹³C NMR (100 MHz, CDCl₃) δ 166.51, 154.61, 152.67, 135.52, 131.83, 129.34, 128.19, 123.27, 79.42, 77.16, 76.84, 72.30, 30.09.

Figure S7. ¹H NMR spectra of synthesized miR-21 inhibitor in CDCl₃.

Figure S8. ^{13}C NMR spectra of miR-21 inhibitor in CDCl_3 .

Figure S10. Characterization of the isolated EVs derived from MDA-MB-231 cells. (A) Size distribution of EVs. (B) TEM characterization of EVs. (C) Western blot analysis of EVs and the corresponding cell lysates. (D) qRT-PCR analysis the content of miR-21 in EVs vs. its expression in MDA-MB-231 cells with miR-21-inhibitor pre-treatment (normalized to U6 RNA). Results are presented as means \pm standard deviation (SD) (n=5).

5. In Fig. S9D, are the authors sure about the units for TNF- α levels? Shouldn't it be pg/ml?

Response: Thank you very much for your valuable comments. We are sorry for our careless mistake. The units of TNF- α is pg/ml after re-examining the systemic immune response results (revised **Figure S15E**). The cytokine (IL-6 and TNF- α) expressions were evaluated on healthy C57BL/6 mice after the intravenous injection

of EVs for different durations (revised **Figures S15D** and **S15E**). As expected, the EVs treatment had no significant impact on the cytokine expression as evidenced by the constant content of serum cytokines (IL-6 and TNF- α). These results suggest that our EVs nanovesicles are indeed immunologically inert, which is good accordance with previous observations [Yang, Z., Shi, J., Xie, J., Wang, Y., Sun, J., Liu, T., et al. Large-scale generation of functional mRNA-encapsulating exosomes via cellular nanoporation. *Nat. Biomed. Eng.* **4**, 69-83 (2020); Liang, Q., Bie, N., Yong, T., Tang, K., Shi, X., Wei, Z., et al. The softness of tumour-cell-derived microparticles regulates their drug-delivery efficiency. *Nat. Biomed. Eng.* **3**, 729-740 (2019)]. The as-suggested immunogenicity analysis and the accompanying discussions have been supplemented in the revised manuscript and supporting information.

Figure S15. The biocompatibility evaluation of EVs. (A) Cell viability assay of MDA-MB-231, HeLa, MRC-5 and BMDCs cells after their incubation with blank EVs (225 µg/mL) for varied durations. (B) Hemolytic analysis of RNAi prodrugs-packaged EVs at varied concentrations. PBS and water were used as negative and positive control, respectively. (C) SEM image of RBCs after its treatment with RNAi prodrugs-loaded EVs (500 µg/mL) for 6 h. The respective expression level of IL-6 (D) and TNF-α (E) as measured by ELISA kit after the

C57BL/6 mice was administrated with bare EVs for different durations. The cell viability and hemolysis assay results are presented as means \pm standard deviation (SD) (n=5) while the serum cytokines results are presented as means \pm standard deviation (SD) (n=3).

6. In Fig. S12C, the transfection efficiency is relative to what? (considering that efficiency in MDA-MB-231 is less than 100%).

Response: Thank you very much for your valuable comments. We are sorry for our careless mistake. In fact, the relative transfection efficiency is calculated in comparison to the donor MDA-MB-231 cells (revised **Figure S19**). The mean fluorescence intensity (MFI) of flow cytometry assay was obtained by FlowJo (Version 10). The summed intensity of the intact untreated cells and RNAi prodrugs-packaged EVs-treated cells were respectively set as the background and total fluorescence intensity (revised **Figure S19B** and **S19C**), and were respectively served as the denominator and numerator in the calculation of the normalized MFI. Subsequently, the normalized MFI of MDA-MB-231 cells was set as 100% for calculating the relative efficiency of HeLa, MCF-7 and MRC-5 cells. The relative transfection efficiency was acquired by dividing the normalized MFI of non-homologous cells (HeLa, MCF-7 and MRC-5) with that of MDA-MB-23 cells (revised **Figure S19D**). These homotypic targeting performance of our EVs and the accompanying discussions have been supplemented in the revised manuscript and supporting information.

Figure S19. Homotypic targeting evaluation of EVs. (A) CLSM image of MDA-MB-231, HeLa, MCF-7 and MRC-5 cells that were incubated with RNAi prodrugs-loaded EVs derived from MDA-MB-231 cells for 3 h. The cells nuclei were

stained with Hoechst 33342 (blue). Scale bar: 20 μm . (B) Flow cytometry analysis of the RNAi prodrugs in different cells. (C) Quantitative analysis of the intracellular fluorescence intensity of RNAi prodrugs in different cells. (D) The relative uptake efficiency of different cells as compared to the MDA-MB-231 cells. Results are presented as means \pm standard deviation (SD) (n=3).

7. In Fig. S13 it seems the authors used ICG-pDNA for liposomes and Cy5-RNAi prodrug for EVs. How can they compare the signal if they used different dyes and different cargoes? (although they mention something different in Fig. S13A and C). Did they load pDNA or the RNAi prodrug in EVs?

Response: Thank you very much for your valuable comments. We are sorry for our careless mistake. Here the cellular uptake of RNAi prodrugs-packaged EVs and other nanocarriers was evaluated by using the same Cy5 fluorophore. The Cy5-labeled RNAi prodrugs were used to investigate the homotypic targeting ability of our EVs (revised **Figure S21**). As revealed by CLSM (revised **Figure S21A**) and flow cytometry (revised **Figures S21B** and **21C**), the MDA-MB-231 cells exhibited higher cellular uptake of homologous MDA-MB-231-derived EVs, as compared with the heterologous Cal 27-derived EVs and even the commercial transfection reagent, thus convincingly validating the specific homotypic targeting of EVs-camouflaged RNAi prodrugs toward donor cancer cells. These cancer cell-specific uptake experimental demonstrations and the accompanying discussions have been supplemented in the revised manuscript and supporting information.

Figure S21. Demonstration of the homotypic affinity of EVs towards donor tumor cells. (A) CLSM images of MDA-MB-231 cells incubated with MDA-MB-231-derived EVs, Cal 27-derived EVs and liposome that were transfected with an equivalent amount of Cy5-labeled RNAi prodrugs for 3 h. Scale bar: 20 μ m. (B) Flow cytometric analysis and (C) the corresponding mean fluorescence intensity (MFI) of MDA-MB-231 cells incubated with MDA-MB-231-derived EVs, Cal 27-derived EVs and liposome that were transfected with an equivalent amount of Cy5-labeled RNAi prodrugs for 3 h. Results are presented as means \pm standard deviation (SD) (n=3).

8. The expression levels of miR-21 in different cells are relative to what (Fig. S15)?

Response: Thank you very much for your valuable comments. Herein, the relative expression levels of miR-21 was normalized by using the U6 small RNA as the internal control. The varied miR-21 expressions of three different cells, including MDA-MB-231, MCF-7, HeLa and MRC-5, were evaluated by traditional qRT-PCR where the reverse transcription was carried out by using Mir-X miRNA First-Strand Synthesis Kit (TaKaRa, Japan) according to the manufactures instructions.

Figure S23. The relative expressions of miR-21 in MDA-MB-231, MCF-7, HeLa and MRC-5 cells by using qRT-PCR evaluation. Results are presented as means \pm standard deviation (SD) (n=5).

9. In Figure 3A the authors mention in the legend, that cells were incubated with DNA-prodrug packaging EVs, but they must be using RNA/DNA hybrids.

Response: Thank you very much for your valuable comments. We are sorry for providing these as-mentioned and other ambiguous descriptions on RNA/DNA hybrids. Here the content of partition DNA was adapted as the concentration of RNAi prodrugs. Accordingly, the original obscure description “Cell viability of MRC-5, HeLa, MCF-7 and MDA-MB-231 cells after incubation with DNA prodrug-packaging EVs (internal DNA concentrations of 25, 50, 75 and 150 nM) for 48 h” was re-phrased into “Cell viability of MRC-5, HeLa, MCF-7 and MDA-MB-231 cells after their incubation with functional EVs encapsulating with different concentrations of RNAi prodrugs (25, 50, 75 and 150 nM) for 48 h” in the revised manuscript. We hope that the present version is now acceptable.

Figure 3. Cell cytotoxicity evaluation of RNAi prodrugs-packaged EVs in hypoxic condition. (A) Cell viability of MRC-5, HeLa, MCF-7 and MDA-MB-231 cells after their incubation with functional EVs encapsulating with different

concentrations of RNAi prodrugs (25, 50, 75 and 150 nM) for 48 h. Results are presented as means \pm standard deviation (SD) (n=5). **P < 0.01, ***P < 0.001, ****P < 0.0001 (two-way ANOVA with Bonferroni's multiple comparisons test). ns, not statistically significant. (B) Combination index (CI) values of RNAi prodrugs-packed EVs with and without photoirradiation. (C) Transwell migration assay, (D) Matrigel invasion assay, and (E) live/dead cells analysis of MDA-MB-231 cells with different treatments. (a): saline; (b): EVs-sustained HCR-scrambled-siRNAs; (c): EVs-sustained HCR-stimulated twist; (d): EVs-sustained HCR-stimulated HIF-1 α ; (e): EVs-sustained HCR-scrambled-siRNA with photoirradiation; (f): EVs-sustained direct twist/HIF-1 α ; (g): EVs-sustained HCR-stimulated twist/HIF-1 α ; (h): liposome-sustained HCR-stimulated twist/HIF-1 α with photoirradiation; (i): EVs-sustained HCR-stimulated twist/HIF-1 α with photoirradiation. All photoirradiation was carried out with 808 nm laser (0.7 W/cm²) for 5 min. Results are presented as means \pm standard deviation (SD) (n=3). *P < 0.05, **P < 0.01, ***P < 0.001, ****P < 0.0001 (one-way ANOVA with a Tukey post hoc test for (C) and (D)). ns, not statistically significant.

10. In Fig. S18 one group – group h – is missing.

Response: Thank you very much for your instructive comments. We are sorry for our careless mistakes. The cell viability of the EVs-sustained HCR-stimulated twist/HIF-1 α with 808 nm photoirradiation (revised group i instead of group h) was provided in the revised **Figure S29**. The viability of bare EVs under 808 nm photoirradiation (data a) was above 85% over the entire concentration range, indicating the high biocompatibility of EVs under the moderate photoirradiation condition. As compared with the single-RNAi-involved gene silencing system at an even high dosage (33% cell viability lose, data c; 36% cell viability lose, data d), the combined bis-RNAi-involved system offered the enhanced therapeutic performance over the entire concentration range (56% cell viability lose, 150 nM, data g). Obviously, the miR-21-responsive RNAi prodrug leads to a more efficient anti-cancerous efficiency than the direct siRNA approach (data g vs f), suggesting a

prolonged gene silencing effect. The photo-irradiation of EVs-sustained HCR-scrambled-siRNAs was observed with moderate cell antiproliferation (data e), indicating an effective PTT operation for disease treatment. Under a high dosage of RNAi prodrugs (150 nM) and photoirradiation stimulation, the proliferation of MDA-MB-231 cells was inhibited by 72% (data i), suggesting a cooperatively enhanced therapeutic performance of the combined gene silencing and PTT. Notably, the anti-tumor efficiency of bis-siRNA approach is higher in EVs than in liposome (data i vs h), which further validates the cell-specific homotypic targeting EVs from donor to recipient cells. These control cytotoxicity experiments have been supplemented with and appropriate descriptions in the revised manuscript and supporting information.

Figure S29. Cell viability of the MDA-MB-231 cells after different treatments for 48 h in hypoxic condition. (a): bare EVs with photoirradiation; (b): EVs-sustained HCR-scrambled-siRNAs; (c): EVs-sustained HCR-stimulated twist; (d): EVs-sustained HCR-stimulated HIF-1 α ; (e): EVs-sustained HCR-scrambled-siRNA with photoirradiation; (f): EVs-sustained direct twist/HIF-1 α ; (g): EVs-sustained HCR-stimulated twist/HIF-1 α ; (h): liposome-sustained HCR-stimulated twist/HIF-1 α with photoirradiation; (i): EVs-sustained HCR-stimulated twist/HIF-1 α with photoirradiation. All photoirradiation was carried out with 808 nm laser (0.7 W/cm²) for 5 min. Results are presented as means \pm standard deviation (SD) (n=5).

11. The photothermal activity of ICG should be explained in more detail in page 15, before the obtained results are explained.

Response: Thank you very much for your insightful comments. According to your suggestion, the enhanced photothermal conversion capacity of the EVs-encapsulated ICG was explained in the revised manuscript and supporting information. As shown in **Figure S27A**, the temperature increase of EVs-encapsulated ICG-labeled RNAi prodrugs was slightly higher than that of merely ICG-labeled RNAi prodrugs without EVs. This promoted photothermal performance was attributed to the high condensation of encapsulated ICG in EVs, resulting in the lower heat dissipation in EVs upon photoirradiation [Saxena, V.; Sadoqi, M.; Shao, J. Enhanced Photo-Stability, Thermal-Stability and Aqueous-Stability of Indocyanine Green in Polymeric Nanoparticulate Systems. *J. Photochem. Photobiol., B* **74**, 29-38 (2004)]. Thus the ICG-labeled RNAi prodrugs could be utilized as a robust photothermal agent for noninvasive hyperthermia-mediated tumor ablation, as evidenced by their dose-dependent photothermal performance (**Figures S27B** and **27C**). This photothermal evaluation of EVs-encapsulated ICG has been supplemented with appropriate discussions in the revised manuscript and supporting information. We hope this explanation is acceptable.

Figure S27. Photothermal responses of ICG-labeled RNAi prodrugs. (A) Photothermal conversion curve of (a) saline, (b) free RNAi prodrugs, (c) RNAi prodrugs-packaged EVs. (B) Photothermal conversion curve of the different concentrations of RNAi prodrugs-packaged EVs: (a) saline, (b) 2.5 $\mu\text{g/mL}$, (c) 5 $\mu\text{g/mL}$, (d) 15 $\mu\text{g/mL}$. (C) The corresponding thermal images of different concentrations of RNAi prodrugs-packaged EVs in Figure S27B. All photoirradiation was carried out with 808 nm laser (0.7 W/cm^2) for 5 min.

12. How were the RNA/DNA hybrids functionalized with ICG? There is no reference to this anywhere, and this protocol is very important to enable data reproducibility.

Response: Thank you very much for your insightful comments. We are sorry for providing an ambiguous description on the ICG-labeled DNA. Here the ICG-NHS ester (MW: 713.9521) was obtained from Xi'an ruixi Biological Technological Co., Ltd (China). The intact DNA (\mathbf{H}_2 , MW: 18450.0627) and ICG-labeled \mathbf{H}_2 were synthesized from Takara. According to the ESI-MS (**Figure S26**), the DNA (\mathbf{H}_2) was successfully functionalized with ICG. The characterization of ICG-labeled DNA has

been supplemented with appropriate discussions in the revised manuscript and supporting information.

ESI-MS (m/z): [ICG-H₂]⁺ calcd for 19164, found 19167.3.

Figure S26. ESI-MS analysis of ICG-functionalized DNA.

13. It is not clear if the RNA/DNA hybrids are functionalized with ICG in Fig. S18. Legends should reflect this information. An additional control using EVs HCR-scramble siRNAs subjected to photoirradiation should also be added, as it would be important to evaluate the effect of photoirradiation by itself.

Response: Thank you very much for your insightful comments. We agree that the photothermal property of HCR-scramble siRNAs should be evaluated in EVs under photoirradiation. Accordingly, the cell viability of EVs-sustained HCR-scrambled-siRNAs was evaluated under 808 nm laser irradiation (0.7 W/cm²). As shown in the revised **Figure S29**, the photo-irradiation of EVs-sustained HCR-scrambled-siRNAs induced a moderate cell antiproliferation (data e), indicating an effective PTT operation in live cells. The extensive photothermal evaluation of different systems has been supplemented with appropriate discussions in the revised manuscript and supporting information.

Figure S29. Cell viability of the MDA-MB-231 cells after different treatments for 48 h in hypoxic condition. (a): bare EVs with photoirradiation; (b): EVs-sustained HCR-scrambled-siRNAs; (c): EVs-sustained HCR-stimulated twist; (d): EVs-sustained HCR-stimulated HIF-1 α ; (e): EVs-sustained HCR-scrambled-siRNA with photoirradiation; (f): EVs-sustained direct twist/HIF-1 α ; (g): EVs-sustained HCR-stimulated twist/HIF-1 α ; (h): liposome-sustained HCR-stimulated twist/HIF-1 α with photoirradiation; (i): EVs-sustained HCR-stimulated twist/HIF-1 α with photoirradiation. All photoirradiation was carried out with 808 nm laser (0.7 W/cm²) for 5 min. Results are presented as means \pm standard deviation (SD) (n=5).

14. A description how the combination index in Fig. 3B was calculated should be added to Materials and Methods.

Response: Thank you very much for your insightful comments. We are sorry for not providing the detailed calculation procedure of combination index (CI). According to your suggestion, the detailed procedure of CI acquirement was supplemented with appropriate descriptions in the Experimental Section of our revised manuscript. The inherent relevancy among these different therapeutic strategies were assessed by calculating CI values through the Chou-Talalay analysis by using CalcuSyn software [Chou, T.-C. Drug Combination Studies and Their Synergy Quantification Using the Chou-Talalay Method. *Cancer Res.* **70**, 440-446 (2010)]. Firstly, the cell viability of different treatments was evaluated using MTT. By plotting the cell viability vs the

concentration of RNAi prodrugs, the CI parameters can be obtained in CalcuSyn software. The CI value indicates the intrinsic drug interaction: $CI > 1$ indicates an antagonistic effect, $CI = 1$ indicates an additive effect, and $CI < 1$ indicates a synergistic effect. Considering the photoirradiated bare EVs were observed with negligible cytotoxicity, the CI values were plotted against the dosages of RNAi prodrugs. According to the revised **Figure 3B**, the twist and HIF-1 α siRNA showed an obvious synergistic effect ($0 < CI < 0.4$) in the as-investigated concentration range, suggesting that the bis-siRNAs could indeed enhance the gene silencing performance. Also, an obvious synergism ($0 < CI < 0.44$) was acquired in photoirradiated miRNA-activated RNAi prodrugs-treated MDA-MB-231 cells, validating a cooperatively enhanced therapeutic performance of the combined gene silencing and PTT system. The calculation procedure of CI has been supplemented with appropriate discussions in the revised manuscript and supporting information. We hope the present version is acceptable.

Figure 3. Cell cytotoxicity evaluation of RNAi prodrugs-packaged EVs in hypoxic condition. (A) Cell viability of MRC-5, HeLa, MCF-7 and MDA-MB-231 cells after their incubation with functional EVs encapsulating with different

concentrations of RNAi prodrugs (25, 50, 75 and 150 nM) for 48 h. Results are presented as means \pm standard deviation (SD) (n=5). **P < 0.01, ***P < 0.001, ****P < 0.0001 (two-way ANOVA with Bonferroni's multiple comparisons test). ns, not statistically significant. (B) Combination index (CI) values of RNAi prodrugs-packed EVs with and without photoirradiation. (C) Transwell migration assay, (D) Matrigel invasion assay, and (E) live/dead cells analysis of MDA-MB-231 cells with different treatments. (a): saline; (b): EVs-sustained HCR-scrambled-siRNAs; (c): EVs-sustained HCR-stimulated twist; (d): EVs-sustained HCR-stimulated HIF-1 α ; (e): EVs-sustained HCR-scrambled-siRNA with photoirradiation; (f): EVs-sustained direct twist/HIF-1 α ; (g): EVs-sustained HCR-stimulated twist/HIF-1 α ; (h): liposome-sustained HCR-stimulated twist/HIF-1 α with photoirradiation; (i): EVs-sustained HCR-stimulated twist/HIF-1 α with photoirradiation. All photoirradiation was carried out with 808 nm laser (0.7 W/cm²) for 5 min. Results are presented as means \pm standard deviation (SD) (n=3). *P < 0.05, **P < 0.01, ***P < 0.001, ****P < 0.0001 (one-way ANOVA with a Tukey post hoc test for (C) and (D)). ns, not statistically significant.

15. Again, in Fig. S20, S21 and Fig. 3, an additional control using EV HCR scramble-siRNAs with photoirradiation is lacking. The presence of ICG in the respective groups, should be included in the figure legends.

Response: Thank you very much for your insightful comments. According to your suggestion, more control experiments, including saline (a) ; EVs-sustained HCR-scrambled-siRNAs (b); EVs-sustained HCR-stimulated twist (c); EVs-sustained HCR-stimulated HIF-1 α (d); EVs-sustained HCR-scrambled-siRNAs with 808 nm photoirradiation (e); EVs-sustained direct twist/HIF-1 α (f); EVs-sustained HCR-stimulated twist/HIF-1 α (g); liposome-sustained HCR-stimulated twist/HIF-1 α with 808 nm photoirradiation (h); EVs-sustained HCR-stimulated twist/HIF-1 α with 808 nm photoirradiation (i), were carried out to demonstrate the cooperatively enhanced therapeutic performance of the combinational gene silencing and PTT system (revised **Figures S29, 3C, 3D, 3E and S30**). The photo-irradiation of

EVs-sustained HCR-scrambled-siRNAs induced a moderate cell antiproliferation (date e, revised **Figure S29**), indicating an effective PTT operation in live cells. Cell migration and proliferation were further precisely quantified by using Transwell migration, not by the wound closure assay. Transwell migration Migratory (newly supplemented **Figures 3C** and **S30A**) and Matrigel invasion assays (revised **Figures 3D** and **S30B**) found that the synchronized activation of twist and HIF-1 α siRNA (group g) displayed a 69% decrease of motility and a simultaneous 82% decrease of invasiveness in MDA-MB-231 cells. All of these values are much higher than that of individual twist siRNA (group c) or HIF-1 α siRNA (group d). In contrast, the *in vitro* proliferation/invasiveness was not significantly affected by scrambled-siRNAs (group b). In addition, the miR-21-activated bis-RNAi system (group g) led to a 1.3-fold reduction in both motility and invasiveness properties of the MDA-MB-231 cells, indicating the enhanced gene silencing effect of our RNAi approach. The photoirradiated MDA-MB-231 cells displayed ~35% decrease in both motility and invasiveness (group e) while the photoirradiated bis-RNAi/EVs (group i) could abrogate the cell motility and invasiveness, confirming the synergistically enhanced therapeutic performance of the combined bis-gene silencing and PTT. One should note that the photoirradiated bis-RNAi/liposome-treated cells could display 81% and 92% decrease of mobility and invasiveness, respectively, which is lower than the EVs-based system. The higher mobility and invasiveness inhibition of EVs than commercial liposome was attributed to the cell-specific homotypic targeting ability of EVs. The synergistically enhanced therapy of our combined bis-gene therapy and PTT was further confirmed by live/dead cell assay (revised **Figure 3E**). These as-suggested extensive photothermal therapeutic evaluation has been supplemented with appropriate descriptions in the revised manuscript and supporting information. We hope the present version is acceptable.

Figure 3. Cell cytotoxicity evaluation of RNAi prodrugs-packaged EVs in hypoxic condition. (A) Cell viability of MRC-5, HeLa, MCF-7 and MDA-MB-231 cells after their incubation with functional EVs encapsulating with different

concentrations of RNAi prodrugs (25, 50, 75 and 150 nM) for 48 h. Results are presented as means \pm standard deviation (SD) (n=5). **P < 0.01, ***P < 0.001, ****P < 0.0001 (two-way ANOVA with Bonferroni's multiple comparisons test). ns, not statistically significant. (B) Combination index (CI) values of RNAi prodrugs-packed EVs with and without photoirradiation. (C) Transwell migration assay, (D) Matrigel invasion assay, and (E) live/dead cells analysis of MDA-MB-231 cells with different treatments. (a): saline; (b): EVs-sustained HCR-scrambled-siRNAs; (c): EVs-sustained HCR-stimulated twist; (d): EVs-sustained HCR-stimulated HIF-1 α ; (e): EVs-sustained HCR-scrambled-siRNA with photoirradiation; (f): EVs-sustained direct twist/HIF-1 α ; (g): EVs-sustained HCR-stimulated twist/HIF-1 α ; (h): liposome-sustained HCR-stimulated twist/HIF-1 α with photoirradiation; (i): EVs-sustained HCR-stimulated twist/HIF-1 α with photoirradiation. All photoirradiation was carried out with 808 nm laser (0.7 W/cm²) for 5 min. Results are presented as means \pm standard deviation (SD) (n=3). *P < 0.05, **P < 0.01, ***P < 0.001, ****P < 0.0001 (one-way ANOVA with a Tukey post hoc test for (C) and (D)). ns, not statistically significant.

Figure S30. Cell motility and invasiveness analysis of differently treated MDA-MB-231 cells under hypoxia condition. (A) Transwell migration assay (left) and their corresponding quantification analysis (right) from differently treated MDA-MB-231 cells. (B) Transwell Matrigel invasion assay (left) and their corresponding quantification analysis (right) from differently treated MDA-MB-231 cells. (a): saline; (b): EVs-sustained HCR-scrambled-siRNAs; (c): EVs-sustained HCR-stimulated twist; (d): EVs-sustained HCR-stimulated HIF-1 α ; (e): EVs-sustained HCR-scrambled-siRNA with photoirradiation; (f): EVs-sustained direct twist/HIF-1 α ; (g): EVs-sustained HCR-stimulated twist/HIF-1 α ; (h): liposome-sustained HCR-stimulated twist/HIF-1 α with photoirradiation; (i): EVs-sustained HCR-stimulated twist/HIF-1 α with photoirradiation. All photoirradiation was carried out with 808 nm laser (0.7 W/cm²) for 5 min. Results are presented as means \pm standard deviation (SD) (n=3). *p<0.05, **p<0.01, ***p<0.001, ****p<0.0001 (one-way ANOVA with a Tukey post hoc test for (A) and (B)). ns, not statistically significant.

Reviewers' Comments:

Reviewer #1:

Remarks to the Author:

I am satisfied with all the responses which are largely beyond my expectation. The revised manuscript is recommended to publish in Nature Communications.

One minor suggestion:

It would be better that explanations of abbreviations for different groups are added in each figure directly.

Reviewer #2:

Remarks to the Author:

The authors stated that the HCR occurs in the cytoplasm outside EVs after endocytosis. However, as shown in the revised Fig S13, most of the unpacked RNA/DNA hybrids have been degraded by nucleases, indicating that the released RNA/DNA hybrids from EVs will be digested by nuclease in the cytoplasm.

RNase H is a ubiquitous cellular ribonuclease that recognizes a DNA/RNA duplex and degrades the RNA strand of this duplex. RNase H plays a crucial role in the biochemical processes associated with DNA replication, gene expression, and DNA repair where RNA/DNA hybrids can occur. RNase H is fully functional to degrade the RNA strand in the RNA/DNA hybrids in the intracellular environment, where the concentration of Mg⁺ is ~ 1mM. It is the principal mechanism of antisense therapy, which has been studied for several decades. If the authors believe this mechanism is not correct, they should provide experimental evidence. Otherwise, the authors need to provide a clear explanation for their claim: "The RNAi prodrugs could substantially avoid the RNase H-mediated degradation after endocytosis".

Reviewer #3:

Remarks to the Author:

The authors have satisfactorily addressed most of my concerns, but I have a few further comments:

1. The English still needs to be substantially improved to deliver a clear and concise message to the reader.
2. On page 11 the authors claim "Clearly, the homologous MDA-MB-231 cells exhibited the highest uptake of EVs than these non-homologous cells, which were in good accordance with previous reports.³⁹⁻⁴¹". None of the cited references evaluates or supports the authors' claim. Furthermore, other publications show exactly the opposite, so this needs EITHER to be re-written OR the authors must provide additional evidence in support of their claims.
3. In Fig. S15, why do C57BL/6 mice have so much circulating IL6 and TNF- α before injection, at 0h? This is surprising and difficult to explain. The authors should provide an explanation supported by data.

Reviewer #1 (Remarks to the Author):

I am satisfied with all the responses which are largely beyond my expectation. The revised manuscript is recommended to publish in Nature Communications.

One minor suggestion:

It would be better that explanations of abbreviations for different groups are added in each figure directly.

Response: Thank you very much for your encouraging comments. According to your suggestion, the abbreviations of different groups were directly added into each figure of the revised manuscript and supporting information. We hope that the present version is now acceptable.

Reviewer #2 (Remarks to the Author):

The authors stated that the HCR occurs in the cytoplasm outside EVs after endocytosis. However, as shown in the revised Fig S13, most of the unpacked RNA/DNA hybrids have been degraded by nucleases, indicating that the released RNA/DNA hybrids from EVs will be digested by nuclease in the cytoplasm.

RNase H is a ubiquitous cellular ribonuclease that recognizes a DNA/RNA duplex and degrades the RNA strand of this duplex. RNases H plays a crucial role in the biochemical processes associated with DNA replication, gene expression, and DNA repair where RNA/DNA hybrids can occur. RNase H is fully functional to degrade the RNA strand in the RNA/DNA hybrids in the intracellular environment, where the concentration of Mg⁺ is ~ 1mM. It is the principal mechanism of antisense therapy, which has been studied for several decades. If the authors believe this mechanism is not correct, they should provide experimental evidence. Otherwise, the authors need to provide a clear explanation for their claim: “The RNAi prodrugs could substantially avoid the RNase H-mediated degradation after endocytosis”.

Response: Thank you very much for your insightful comments. We admit that the RNase H-mediated RNA degradation could simultaneously occurs with HCR activation after endocytosis. Yet it seems that the miR-21-responsive bis-RNAs dominated the overall process from the time-dependent FRET evaluation (**Figure S22**) and western

blot analysis (**Figure S28**). The intracellular FERT signal reached to a saturated value in 4 h (**Figure S22**), indicating that the miR-21-responsive HCR proceeded in cytoplasm quickly. Meanwhile, under a similar circumstance, a minority of the RNAi prodrugs (41%) was degraded within the 4 h duration in DNase/RNases-containing Opti-MEM at 37 °C (**Figure S13**). This means that the majority of the RNAi prodrugs (59%) was retained for participating the miR-21-activated HCR system. Even the undesired RNase H-mediated degradation of RNA/DNA hybrids could not be prevented, the residual RNAi prodrugs were sufficient to be activated by miRNA for motivating the stimuli-responsive gene silencing principle. Thus, the HCR-motivated RNAi dominated the overall cellular biotransformations by generating active bis-siRNAs from the RNAi prodrugs during the initial 4 h, after which, the RNase H degradation of residual unreacted RNAi prodrugs played a dominating role within cells. One should note that, after the accomplishment of miR-21-motivated HCR (4 h), the functional usage of residual RNAi prodrugs were rather low considering that the residual miR-21 is rather low and even unavailable for continuing the HCR-involved RNAi activation from RNAi prodrugs (most of miRNA has been consumed via the HCR principle during the initial 4 h reaction). The subsequent RNase H-mediated degradation of RNAi prodrugs could eliminate the possible off-target side effect of RNAi prodrugs. The claim “The RNAi prodrugs could substantially avoid the RNase H-mediated degradation after endocytosis” is inappropriate and thus has been corrected in the revised manuscript and supporting information.

One should note that this undesired RNase H digestion could be fully expelled by introducing a specific chemical modification on the DNA scaffolds, e.g., 2-methoxyethoxy (MOE) nucleotides [Lima, W. F., Rose, J. B., Nichols, J. G., Wu, H., Migawa, M. T., Wyrzykiewicz, T. K., Siwkowski, A. M., Crooke, S.T.. Human RNase H1 discriminates between subtle variations in the structure of the heteroduplex substrate. *Mol. Pharmacol.* **71**, 83-91 (2007)], which we believe could contribute the most efficient RNAi platform. Yet this item is out of the range of the present study since we are more focus on the proof of concept demonstration of the tumorous biomarker-activated RNAi platform.

Further western blot analysis was used to demonstrate the miR-21-responsive RNAi principle (**Figure S28**). Under hypoxia condition, the expression of twist and HIF-1 α proteins were obviously down-regulated in our RNAi-treated cells, while these protein expressions were kept nearly constant in a disabled HCR-involved RNAi system (merely antisense RNA/DNA hybrids-treated cells), indicating that the miR-21-initiated HCR-assembled bis-RNAs could efficiently cleave target mRNAs for gene silencing therapy.

The time-dependent FRET evaluation of miR-21-initiated HCR has been supplemented with appropriate discussions, and all the related claims “The RNAi prodrugs could substantially avoid the RNase H-mediated degradation after endocytosis” have been corrected in the revised manuscript and supporting information. We hope that the present revision is now acceptable.

Figure S22. Time-dependent miR-21 imaging in MDA-MB-231 cells. (A) CLSM analysis of the MDA-MB-231 cells incubated with the RNAi prodrug-packaged EVs

system for different time-intervals. All scale bars are 20 μm . (B) FRET signal distributions of each pixel in Figure S22A. (C) Statistical histogram analysis of the relative fluorescence intensity (in the form of F_A/F_D) of the above five cell samples. Results are presented as means standard deviation (SD) (n=10).

Reviewer #3 (Remarks to the Author):

The authors have satisfactorily addressed most of my concerns, but I have a few further comments:

1. The english still needs to be substantially improved to deliver a clear and concise message to the reader.

Response: Thank you very much for your insightful comments. Accordingly to your suggestion, we have gone through the manuscript and try our best to correct all of these ambiguous and tedious descriptions in our revised manuscript and supporting information. We have also send the revised paper to AJE (American Journal Experts) for further grammar and format editing(attached please find the service verification from the company), and hope the extensively revised paper could now deliver a clear and concise message to the reader.

2. On page 11 the authors claim “Clearly, the homologous MDA-MB-231 cells exhibited the highest uptake of EVs than these non-homologous cells, which were in good accordance with previous reports.³⁹⁻⁴¹”. None of the cited references evaluates or supports the authors' claim. Furthermore, other publications show exactly the opposite, so this needs EITHER to be re-written OR the authors must provide additional evidence in support of their claims.

Response: Thank you very much for your instructive comments. The fact is that these references 39-41 are wrongly placed in this “homologous cellular uptake of EVs”, the corrected ones should be 36 and 37. We are sorry for our careless mistakes. Here the confocal laser scanning microscopy (CLSM) imaging results and quantitative flow cytometry analysis showed that the homologous MDA-MB-231 cells exhibited the highest uptake efficiency than the other non-homologous cells (**Figure S19**), which

were in good accordance with the previous works [Cheng, G., Li, W., Ha, L., Han, X., Hao, S., Wan, Y., Wang, Z., Dong, F., Zou, X., Mao, Y., Zheng, S. Y. Self-Assembly of Extracellular Vesicle-like Metal-Organic Framework Nanoparticles for Protection and Intracellular Delivery of Biofunctional Proteins. *J. Am. Chem. Soc.* **140**, 7282-7291 (2018); Sancho-Albero, M., Rubio-Ruiz, B., Pe´rezLo´pez, A. M., Sebastia´n, V., Martı´n-Duque, P., Arruebo, M., Santamarı´a, J., Unciti-´Broceta, A. Cancer-derived exosomes loaded with ultrathin palladium nanosheets for targeted bioorthogonal catalysis. *Nat. Catal.* **2**, 864-872 (2019)]. Interestingly, Zomer and coworkers have showed that the MDA-MB-231-produced EVs could however be uptaken by less malignant tumor cells [Zomer, A., Maynard, C., Verweij, F.J., Kamermans, A., Scha¨fer, R., Beerling, E., Schiffelers, R.M., de Wit, E., Berenguer, J., Ellenbroek, S.I., et al. In vivo imaging reveals extracellular vesicle-mediated phenocopying of metastatic behavior. *Cell* **161**, 1046-1057 (2015)]. This is rather difficult for us to provide the exact underlying mechanism since we do not participate these experiments and could not examine the exact differences on experimental apparatus and executions.

Even the related mechanism is unclear, we still try to supplement some possible explanations. This opposite conclusion is might attributed to the different experimental executions and demonstrations. The preferred heterologous cellular uptaken of Zomer’s observation was demonstrated by transwell experiments, in which MDA-MB-231 cells were placed in the upper well while MCF-7 cells were placed at the bottle well for one week prior to analysis. The diverse information transmission or crosstalk of distant cells is realized by the secretion of soluble factors (for example, growth factors), as well as the diverse EVs trafficking paradigm. Here both of MDA-MB-231 donor cells and MCF-7 acceptor cells were genetically modified by using a standard lentiviral transduction protocol, so that the distant EVs-mediated cell-to-cell communication could be revealed by a newly expressed green fluorescent protein (GFP), an indirect indicator of EVs-delivering mRNA event. Yet the MDA-MB-231-secreted soluble factors and exomeres could not expelled, and might be uptaken by MCF-7 acceptor cells, resulting in a converted cellular behavior. Thus the gene-engineering of MDA-MB-231/MCF-7 cells and the non-EVs-based information/substance exchange may

lead to the preferred heterologous cellular uptake in Zomer's system. In our present study, native MDA-MB-231 cells were used to produce EVs, and the native EVs were then electroporated with multifunctional DNA probes and were further purified *via* ultracentrifugation to prevent the possible contamination of other smaller exosomes. Similarly, the native MDA-MB-231/MCF-7 cells were utilized for studying their different cellular uptake performance for functional EVs. The varied cellular uptake of EVs was revealed directly by the encapsulated fluorophore-labeled DNA probes without the complicated mRNA translation step, and was thus accompanied with lower biological interference. Thus the native cell-secreted EVs, the direct cargo-based EVs quantification and non-EVs contamination contributed to the homologous cellular uptake result. One should note that these different experimental design were in fact reasonable even the conclusions were opposite, this might need more in-depth explorations for the EVs research community. For audience's concern, these related descriptions have been supplemented in the revised manuscript and supporting information. We hope that the present revision is now acceptable.

3. In Fig. S15, why do C57BL/6 mice have so much circulating IL6 and TNF- α before injection, at 0h? This is surprising and difficult to explain. The authors should provide an explanation supported by data.

Response: Thank you very much for your insightful comments. In fact, we were also puzzled by this unexpected results, which pushed us to explore the possible explanations. In fact, this phenomena has also been discovered by other research groups [Gong, N., Ma, X., Ye, X., Zhou, Q., Chen, X., Tan, X., Yao, S., Huo, S., Zhang, T., Chen, S., Teng, X., Hu, X., Yu, J., Gan, Y., Jiang, H., Li, J., Liang, X. J. Carbon-dot-supported atomically dispersed gold as a mitochondrial oxidative stress amplifier for cancer treatment. *Nat. Nanotechnol.* **14**, 379-387 (2019); Li, S. P., Jiang, Q., Liu, S. L., Zhang, Y. L., Tian, Y. H., Song, C., Wang, J., Zou, Y. G., Anderson, G. J., Han, J. Y., Chang, Y., Liu, Y., Zhang, C., Chen, L., Zhou, G. B., Nie, G. J., Yan, H., Ding, B. Q., Zhao, Y. L. *Nat. Biotechnol.* **36**, 258-264 (2018)]. We speculated that the healthy C57BL/6 mice may requires a certain amount of serum cytokine (maintaining in a

normal level) to sustain the self-survival metabolism. In addition, the concentration of serum cytokine may also have subtle variation because of the different animal models and ages. One should note that the immunogenicity analysis was introduced to demonstrate the favorable biocompatibility and biosafety of our EVs. Thus the initial expression level of serum cytokine (prior to injection) has scarcely any effect on the biocompatibility evaluation of our EVs. We hope that the present explanation is now acceptable.

Reviewers' Comments:

Reviewer #2:

Remarks to the Author:

The authors fully addressed my concerns. I recommend the publication of this manuscript.

Reviewer #3:

Remarks to the Author:

I am now satisfied with the author responses. No further comments.